# Parkin-mediated ubiquitylation redistributes MITOL/March5 from mitochondria to peroxisomes

Fumika Koyano[1], Koji Yamano[1], Hidetaka Kosako[2], Yoko Kimura[3], Mayumi Kimura[1], Yukio Fujiki[4], Keiji Tanaka[5] & Noriyuki Matsuda[1,*] (ID)

## Abstract

Ubiquitylation of outer mitochondrial membrane (OMM) proteins is closely related to the onset of familial Parkinson's disease. Typically, a reduction in the mitochondrial membrane potential results in Parkin-mediated ubiquitylation of OMM proteins, which are then targeted for proteasomal and mitophagic degradation. The role of ubiquitylation of OMM proteins with non-degradative fates, however, remains poorly understood. In this study, we find that the mitochondrial E3 ubiquitin ligase MITOL/March5 translocates from depolarized mitochondria to peroxisomes following mitophagy stimulation. This unusual redistribution is mediated by peroxins (peroxisomal biogenesis factors) Pex3/16 and requires the E3 ligase activity of Parkin, which ubiquitylates K268 in the MITOL C-terminus, essential for p97/VCP-dependent mitochondrial extraction of MITOL. These findings imply that ubiquitylation directs peroxisomal translocation of MITOL upon mitophagy stimulation and reveal a novel role for ubiquitin as a sorting signal that allows certain specialized proteins to escape from damaged mitochondria.

**Keywords** March5; peroxin; PINK1- and Parkin-mediated mitophagy; ubiquitin; VCP
**Subject Categories** Autophagy & Cell Death; Post-translational Modifications & Proteolysis

## Introduction

The mitochondria are a well-organized organelle. When the mitochondrial membrane potential is decreased, PINK1 and Parkin, which are causal gene products of inherited Parkinson's disease, play pivotal roles in the selective degradation of damaged mitochondria via the proteasome and autophagy pathway (the following is called mitophagy). PINK1 is a Ser/Thr kinase, and Parkin is an RBR-type E3 (ubiquitin ligase) that catalyzes ubiquitin conjugation to protein substrates from a ubiquitin-adducted E2 (ubiquitin-conjugating enzyme) [1–9]. PINK1 accumulates on damaged mitochondria and subsequently phosphorylates Ser65 of both ubiquitin and ubiquitin-like (Ubl) domain of Parkin [10–15]. Through the interaction with phosphorylated ubiquitin, Parkin also becomes phosphorylated, which triggers auto-activation and subsequent conformational changes [16–18]. Ubiquitylation of various mitochondrial proteins by the activated Parkin then leads to selective elimination of damaged mitochondria [19–31].

MITOL/March5 is a RING-type E3 with four transmembrane domains that is integrated into the outer mitochondrial membrane [32–34]. It has been reported that MITOL regulates mitochondrial morphology via association with and ubiquitylation of the mitochondrial fission factors, Fis1 and Drp1 [34]. Indeed, siRNA-mediated knockdown of MITOL resulted in mitochondrial fragmentation. MITOL is also known to ubiquitylate mitochondrial mitofusin2 (MFN2), which leads to the tethering of mitochondria to the endoplasmic reticulum (ER) [33,35,36].

When we investigated whether a mitochondrial E3(s) cooperate with Parkin to eliminate damaged mitochondria [37], we noticed that MITOL showed ectopic localization during Parkin-mediated mitochondrial ubiquitylation. This finding is of interest as the disconnected localization of MITOL and mitochondria had not been previously reported. In this study, we found that both overexpressed and endogenous MITOL translocated from damaged mitochondria to peroxisomes in response to mitochondrial depolarization. Notably, MITOL translocation was completely dependent on PINK1 and the E3 activity of Parkin with ubiquitylation of MITOL at K268 triggering mitochondrial extraction in a p97/VCP-dependent manner. Furthermore, peroxin-dependent translocation of MITOL was also observed, suggesting that the p97/VCP–peroxin axis regulates ubiquitylated MITOL translocation following mitophagy stimulation.

1   Ubiquitin Project, Tokyo Metropolitan Institute of Medical Science, Tokyo, Japan
2   Division of Cell Signaling, Fujii Memorial Institute of Medical Sciences, Tokushima University, Tokushima, Japan
3   Department of Agriculture Graduate School of Integrated Science and Technology, Shizuoka University, Shizuoka, Japan
4   Medical Institute of Bioregulation, Kyushu University, Higashi-ku, Fukuoka, Japan
5   Laboratory of Protein Metabolism, Tokyo Metropolitan Institute of Medical Science, Tokyo, Japan
    *Corresponding author. Tel: +81 3 5316 3123; Fax: +81 3 5316 3152; E-mail: matsuda-nr@igakuken.or.jp

# Results

## MITOL redistributes to peroxisomes in response to mitochondrial depolarization in a Parkin-dependent manner

To understand the molecular details of MITOL localization when the mitochondrial membrane potential is decreased, C-terminal HA-tagged MITOL and Flag-Parkin were transiently expressed in HeLa cells. Cells treated with or without the uncoupler CCCP (carbonyl cyanide *m*-chlorophenylhydrazone) were stained with anti-HA and anti-Tom20/TOMM20 antibodies (Fig 1A). Without CCCP treatment, MITOL-HA overlapped with the mitochondrial outer membrane protein Tom20, indicating its mitochondrial localization. Interestingly, after 3 h of CCCP treatment, a large population of MITOL clustered in dot-like structures that did not co-localize with Tom20, and we assumed that MITOL had localized in another organelle rather than the cytosol. To clarify MITOL localization after mitophagy stimulation, we first performed co-immunostaining experiments using various organelle markers. The MITOL-HA-positive dot-like structures were not coincident with the immunostaining patterns of Sec61β (ER marker) or LAMP1 (lysosome marker; Fig 1B and C), but did largely co-localize with catalase (peroxisome marker) after 3 h of CCCP treatment (Fig 1D and F). Next, co-localization between MITOL and each marker protein was assessed as a Pearson correlation coefficient. The quantitative analysis supported our aforementioned findings that co-localization between MITOL and Tom20 was significantly decreased following CCCP treatment for 3 h (Fig 1E, columns 1 and 2). Although a small percentage of MITOL was merged with catalase in the absence of CCCP treatment, the co-localization markedly increased in response to CCCP treatment for 3 h (Fig 1E, columns 7 and 8). Peroxisomal membrane protein PMP34 fused with FusionRed at its C-terminus (PMP34-FusionRed) also co-localized with MITOL-GFP in Parkin-expressing HeLa cells after 3 h of CCCP treatment (Fig 1G), and this co-localization was confirmed by quantitative analysis (Fig 1H). In addition, the peroxisomal localization of MITOL following dissipation of the mitochondrial membrane potential was unaffected by C-terminal addition of the 3Flag epitope (Appendix Fig S1A). Cells exposed to the potassium ionophore valinomycin (another type of mitochondrial uncoupler) for 3 h also exhibited MITOL and catalase co-localization,

suggesting that peroxisomal localization of MITOL was not an artifact of CCCP treatment (Appendix Fig S1B). These results indicate that MITOL moves to peroxisomes in response to mitophagy stimulation. Next, we tested the possibility that peroxisomes are the halfway point of a translocation process that leads to another organelle. Following extended CCCP treatment, HeLa cells stably expressing HA-Parkin and 3Flag-MITOL were stained with Pex14 (Appendix Fig S2A), Sec61β (Appendix Fig S2B), and Tom20 (Appendix Fig S2C: because Tom20 was substantially degraded via mitophagy for > 12 h, Tom20-positive cells were selected to be shown). MITOL and Pex14 co-localization peaked within 3 h of CCCP treatment and then dispersed over time. However, even with extended CCCP treatment, we found no evidence of MITOL co-localization with Tom20 or Sec61β.

## PINK1 and the E3 activity of Parkin are required for MITOL redistribution to peroxisomes

Next, we investigated whether Parkin is required for the redistribution of MITOL from impaired mitochondria to peroxisomes. For this purpose, MITOL-HA was transfected into HeLa cells that lack a functional *PARK2* gene by genomic mutation [4,38]. As expected, MITOL-HA moved to peroxisomes in HeLa cells stably expressing GFP-Parkin after 3 h of CCCP treatment and did not merge with Tom20 (Fig 2A, bottom panel). In contrast, MITOL-HA was retained on mitochondria after CCCP treatment in HeLa cells lacking endogenous Parkin expression (Fig 2A, upper panel). Valinomycin-treated cells showed the same phenomena (Appendix Fig S1C), and quantitative analysis confirmed that in the absence of Parkin, MITOL-HA was retained on depolarized mitochondria (Fig 2B). These results indicate that Parkin is required for MITOL relocation from mitochondria to peroxisomes.

Next, we sought to demonstrate that pre-existing mitochondrial MITOL moved to peroxisomes in response to mitochondrial depolarization, rather than direct peroxisomal targeting of newly synthesized MITOL following CCCP treatment. The simplest experiment would suggest the use of cycloheximide (CHX), which blocks protein synthesis. However, we cannot use CHX as Parkin translocation to impaired mitochondria depends on the accumulation of newly synthesized PINK1 on the outer mitochondrial membrane

---

**Figure 1. MITOL redistributes to peroxisomes in response to a loss in the mitochondrial membrane potential.**

A–C   MITOL-HA formed small dot-like structures following CCCP treatment. HeLa cells transiently expressing Flag-Parkin and MITOL-HA were treated with 15 μM CCCP for 3 h, and then subjected to immunocytochemistry. MITOL and Tom20 (mitochondrial marker; A) signals co-localized well without CCCP treatment, whereas the MITOL-positive small dots were not coincident with Tom20, Sec61β (ER marker; B), or LAMP1 (lysosomal marker; C) after CCCP treatment. Scale bars, 10 μm.

D   MITOL-HA co-localized with catalase (peroxisome marker) following CCCP treatment. Higher magnification images of the boxed regions are shown in the small panel. Scale bars, 10 μm. Arrowheads indicate representative examples of MITOL-HA co-localization with catalase.

E   Correlation statistics for the localization of MITOL-HA and Tom20, Sec61β, LAMP1, or catalase. Dots indicate individual Pearson correlation coefficient data points. In the box-plots, the center lines indicate the medians, the box limits indicate the 25th and 75th percentiles as determined in the R software package, and the whiskers extend 1.5 times the interquartile range from the 25th and 75th percentiles. Means and the number of samples are shown on the box and *X*-axis, respectively. Statistical significance was calculated using a one-tailed Welch's *t*-test.

F   The line graph shows a line scan of fluorescence through three MITOL-positive peroxisomes (red bar in D) that clearly indicates co-localization of MITOL (green line) and catalase (magenta line).

G   Peroxisomal membrane protein (PMP) 34-FusionRed also co-localized with MITOL-GFP in Parkin-expressing HeLa cells after 3 h of CCCP treatment. Higher magnification images of the boxed regions are shown in the small panel. Scale bars, 10 μm.

H   Correlation statistics for the localization of MITOL-GFP and PMP34-FusionRed. Dots indicate individual Pearson correlation coefficient data points. In the box-plots, the center lines indicate the medians, box limits indicate the 25th and 75th percentiles as determined in the R software package, and the whiskers extend 1.5 times the interquartile range from the 25th and 75th percentiles. Means and the number of samples are shown on the box and *X*-axis, respectively. Statistical significance was calculated using a one-tailed Welch's *t*-test.

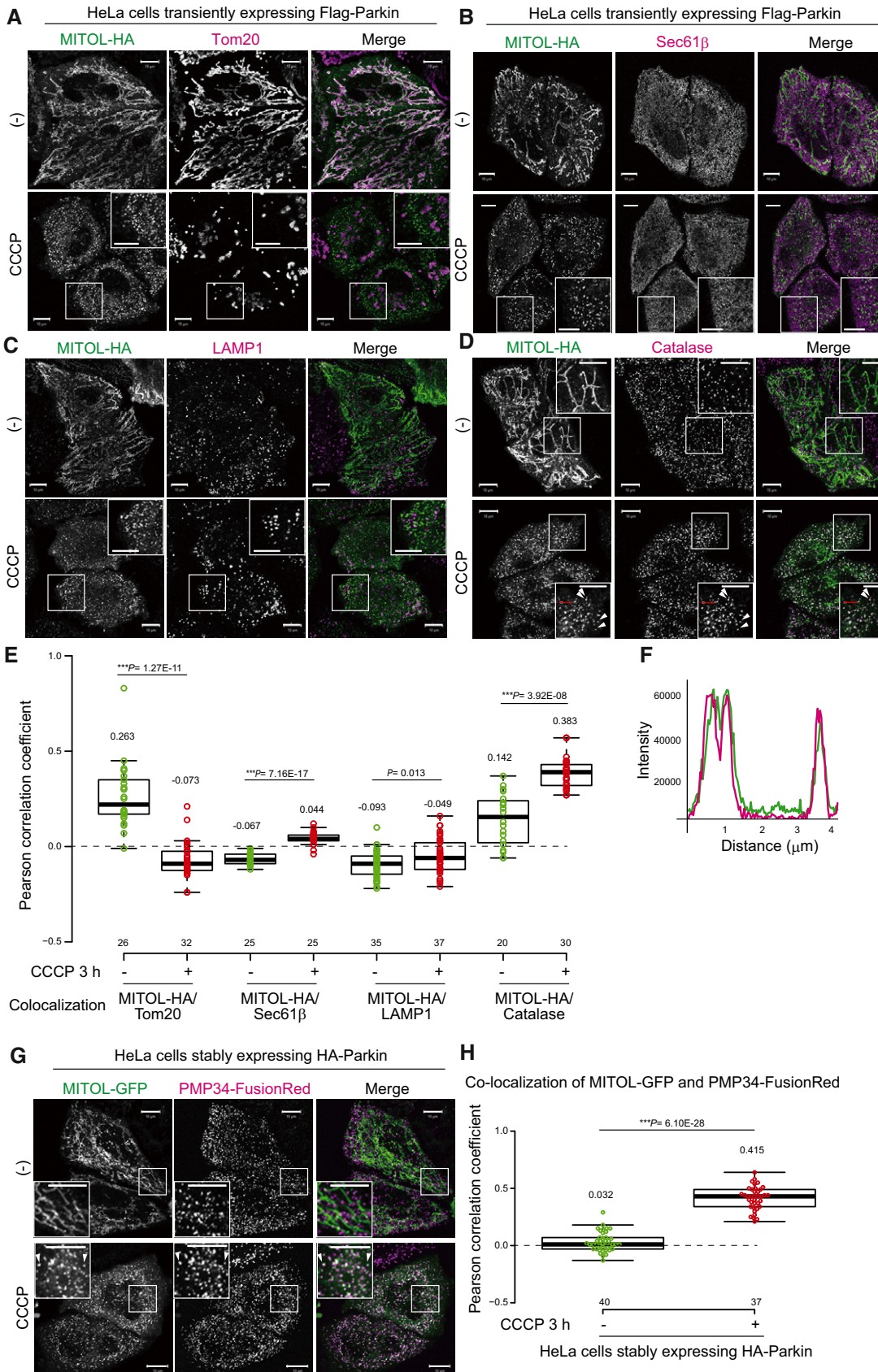

Figure 1.

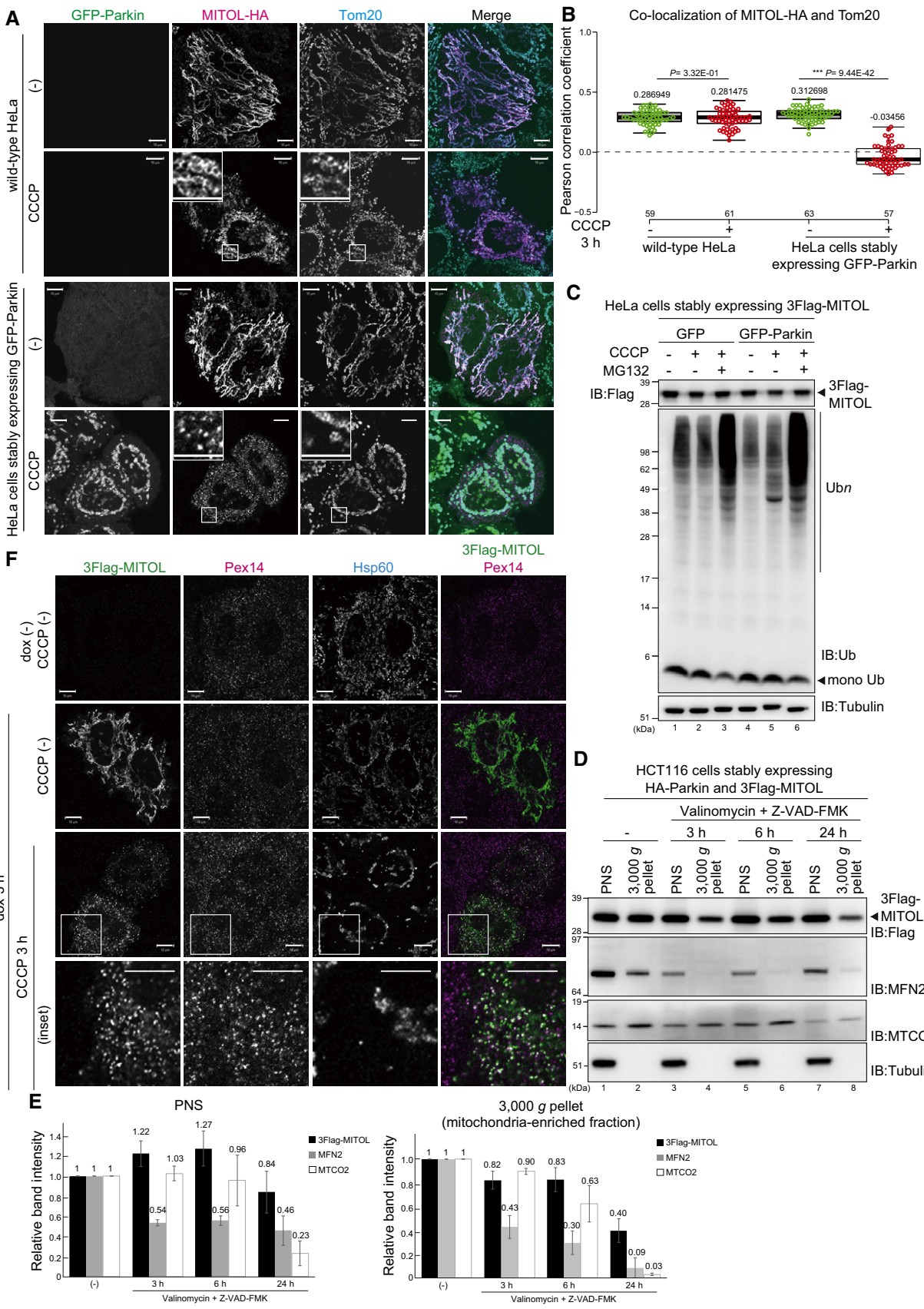

**Figure 2.**

**Figure 2.  Parkin is required for MITOL redistribution to peroxisomes.**

A   MITOL-HA did not move to peroxisomes, but was rather retained on mitochondria even after CCCP treatment in HeLa cells lacking endogenous Parkin. Wild-type HeLa cells or HeLa cells stably expressing GFP-Parkin were transfected with MITOL-HA, treated with 15 μM CCCP for 3 h, and then subjected to immunocytochemistry with anti-HA and anti-Tom20 antibodies. Higher magnification images of the boxed regions are shown in the small panel. Scale bars, 10 μm.

B   Correlation statistics for the localization of MITOL-HA and Tom20 in the absence or presence of GFP-Parkin. Dots indicate individual Pearson correlation coefficient data points. In the box-plots, the center lines indicate the medians, the box limits indicate the 25th and 75th percentiles as determined in the R software package, and the whiskers extend 1.5 times the interquartile range from the 25th and 75th percentiles. Means and the number of samples are shown on the box and *X*-axis, respectively. Statistical significance was calculated using a one-tailed Welch's *t*-test.

C   MITOL was not degraded following mitochondrial depolarization. HeLa cells stably expressing 3Flag-MITOL were transfected with a GFP-Parkin plasmid or the pEGFP-C1 vector, treated with 15 μM CCCP ± 10 μM MG132 for 3 h, and then immunoblotted with anti-Flag, anti-ubiquitin, and anti-tubulin antibodies. Black arrowheads indicate 3Flag-MITOL in the upper panel and mono-ubiquitin in the middle panel, respectively.

D   The total cellular amount of MITOL was not dramatically reduced following extended valinomycin treatment. Immunoblotting combined with fractionation analysis showed that mitofusin2 (MFN2) underwent rapid degradation within 3 h of valinomycin treatment, in particular in the 3,000 *g* pellet (mitochondria-rich fractions). Cytochrome c oxidase subunit 2 (MTCO2, inner mitochondrial protein) was significantly reduced at 24 h 10 μM valinomycin treatment. In contrast to those two proteins, MITOL degradation was minimal. Note that the chemical apoptosis inhibitor Z-VAD-FMK (10 μM) was added to cells along with valinomycin to prevent cell death.

E   Quantification of 3Flag-MITOL, MFN2, and MTCO2 protein levels in the PNS and 3,000 *g* pellet fraction following 10 μM valinomycin + Z-VAD-FMK treatment at the indicated times. Data represent the mean fold change ± s.e.m. relative to untreated samples in three biological replicates.

F   Pre-existing MITOL on mitochondria moves to peroxisomes following CCCP treatment. Following doxycycline treatment for 3 h to induce MITOL expression, cells were washed with fresh medium to stop the synthesis of new MITOL. After treatment with or without CCCP for more than 3 h, cells were immunostained using anti-Flag, anti-Pex14 (peroxisomal membrane protein), and anti-Hsp60 antibodies. Higher magnification images of the boxed regions are shown in the bottom panel. Scale bars, 10 μm.

following CCCP treatment, and thus, CHX treatment would block PINK1 synthesis and consequently impede Parkin translocation/activation [39]. Instead of CHX, we utilized a doxycycline induction/repression system. HeLa cells stably expressing HA-Parkin were transiently transfected with pTRE3G-3Flag-MITOL and pCMV-Tet3G plasmids. Before doxycycline treatment, MITOL expression was repressed and no signal was observed (Fig 2F, top panel). After 3 h of doxycycline treatment, MITOL expression was induced and the protein localized on mitochondria (Fig 2F). Cells were then repeatedly washed to remove doxycycline induction (i.e., no new MITOL synthesis) and treated with CCCP for an additional 3 h. Examination of these cells revealed co-localization of MITOL and the peroxisomal marker Pex14 (Fig 2F), suggesting that previously synthesized MITOL that had been localized on mitochondria had moved to peroxisomes in response to CCCP treatment.

We also wanted to eliminate the trivial possibility that (i) MITOL exists in two distinct organellar states, one that is predominantly localized on mitochondria and a smaller grouping localized on peroxisomes, and (ii) that very rapid degradation (within 3 h) of mitochondria-localized MITOL leaves peroxisome-localized MITOL as the dominant grouping. This possibility could lead to an erroneous conclusion that MITOL translocated to the peroxisomes. If true, then the total amount of MITOL should decrease rapidly; however, the protein level of MITOL was not drastically altered following CCCP or MG132 treatment, suggesting that a large fraction of MITOL is not rapidly degraded during mitophagy stimulation (Fig 2C). Furthermore, we monitored the levels of MITOL over an extended period of valinomycin treatment (6 and 24 h) in conjunction with fractionation analysis. Mitofusin2 (MFN2), which is degraded by the proteasome following mitochondrial depolarization [22], underwent rapid degradation within 3 h of valinomycin treatment (Fig 2D and E), in particular the mitochondria-rich fractions. MTCO2, an inner mitochondrial protein that is degraded via mitophagy following mitochondrial depolarization [40], was significantly decreased following 24 h of valinomycin treatment (Fig 2D and E). In contrast, 3Flag-MITOL persisted throughout the first 6 h and only underwent a slight reduction after 24 h, confirming that the total cellular amount of MITOL is not dramatically decreased. Although

MITOL underwent a modest reduction at 24 h, it was not as rapid as that seen with MFN2 and MTCO2 that are destined for Ub- and mitophagy-mediated degradation (Fig 2D and E). Given that the mitochondria-enriched fraction contains some peroxisomes (as shown in Fig 11), the results are consistent with our hypothesis that mitochondrial MITOL is not degraded but rather migrates to peroxisomes.

MITOL redistribution from impaired mitochondria to peroxisomes occurs downstream of Parkin. Because *ATG5* is indispensable for the progression of autophagy, an *ATG5* knockout cell line was used to investigate whether mitophagy inhibition affects the peroxisomal translocation of MITOL. When MITOL localization in *ATG5* knockout cells stably expressing HA-Parkin and 3Flag-MITOL was analyzed by immunocytochemistry, MITOL clearly overlapped with PMP70 (peroxisomal marker) following CCCP treatment, which was similar to that seen in wild-type cells (Fig EV1). This indicates that inhibition of mitophagy does not affect the redistribution of MITOL to peroxisomes. These results further refute the possibility that MITOL is localized on both peroxisomes and mitochondria under steady-state conditions and that peroxisomal localization only becomes apparent following the rapid degradation of mitochondrial MITOL via CCCP-induced mitophagy.

Collectively, it is unlikely that newly synthesized MITOL is directly targeted to peroxisomes in response to CCCP treatment or that mitochondrial MITOL is rapidly degraded within 3 h of CCCP treatment; rather, the data support translocation of pre-existing mitochondrial MITOL to peroxisomes in response to mitochondrial depolarization.

To further analyze Parkin dependency, we examined whether Parkin E3 activity is required for MITOL translocation to peroxisomes. In the presence of wild-type Parkin, 3Flag-MITOL localized to peroxisomes after CCCP treatment as described above. In contrast, no MITOL translocation was observed in cells expressing a Parkin (C431S) mutant in which the catalytic Cys critical for E3 activity had been mutated (Fig 3A, middle panels), indicating that the redistribution of MITOL from mitochondria to peroxisomes requires Parkin E3 activity. Furthermore, we also confirmed that PINK1 is required for MITOL redistribution. In *PINK1* knockout

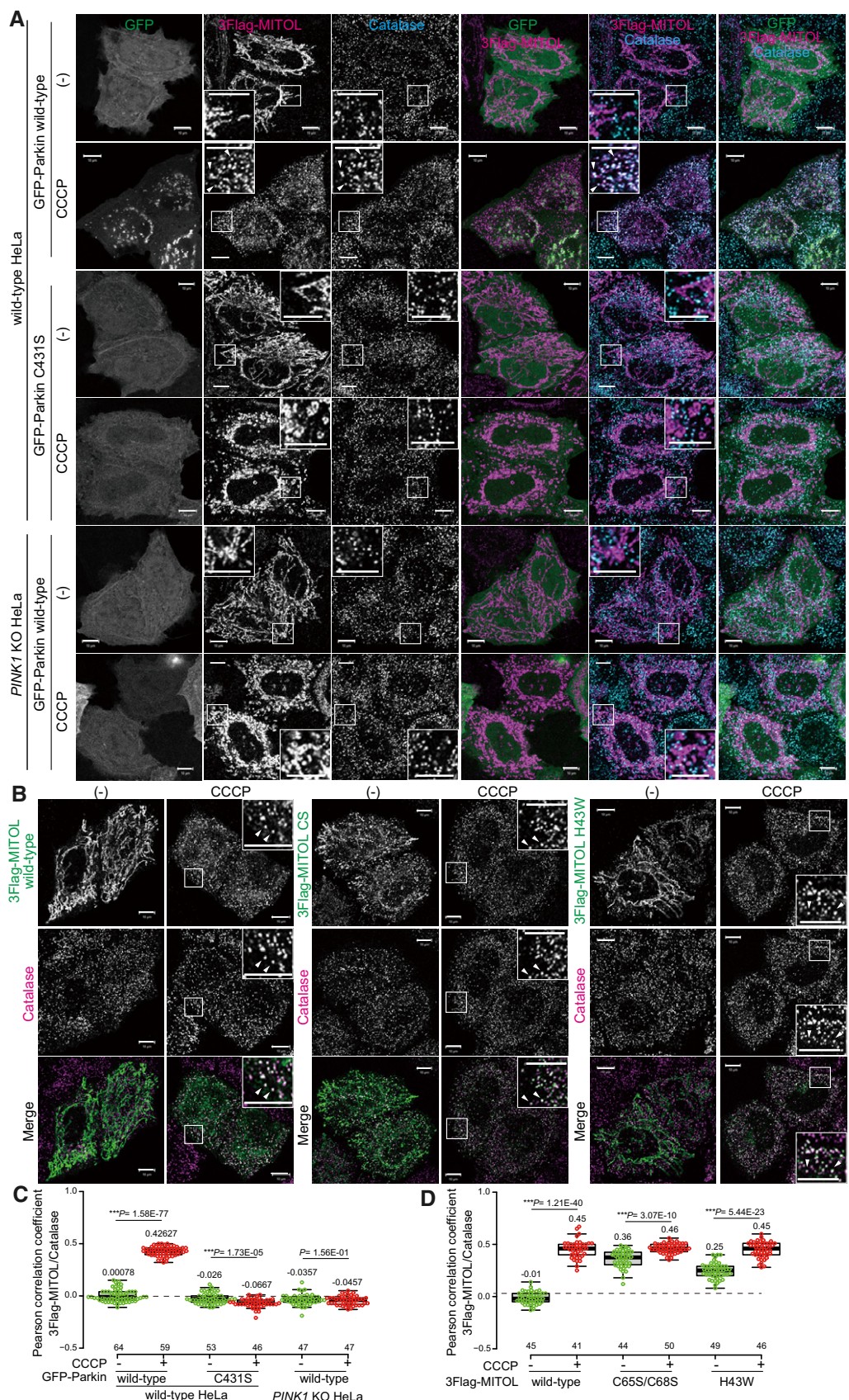

**Figure 3.  Both PINK1 and the E3 activity of Parkin are required for MITOL redistribution to peroxisomes.**

A   Wild-type or *PINK1* knockout (KO) HeLa cells were transfected with GFP-Parkin wild-type or the C431S mutant, treated with 15 μM CCCP, and then immunostained with anti-Flag and anti-catalase antibodies. Wild-type Parkin, but not the catalytically inactive Parkin (C431S) mutant, mediated 3Flag-MITOL translocation to peroxisomes following mitophagy stimulation. In *PINK1* KO HeLa cells, the redistribution of MITOL to peroxisomes was not observed even in the presence of wild-type Parkin when mitochondria were damaged. Higher magnification images of the boxed regions are shown in the small panel. Scale bars, 10 μm. Arrowheads indicate representative examples of MITOL–peroxisome co-localization observed only in the presence of wild-type Parkin.

B   The redistribution of MITOL from mitochondria to peroxisomes does not require its own E3 activity. The E3-inactive MITOL Cys65Ser/Cys68Ser (CS) and H43W mutants were transfected into HeLa cells stably expressing HA-Parkin, treated with 15 μM CCCP, and then subjected to immunocytochemistry with anti-Flag and anti-catalase antibodies. After 3 h of CCCP treatment, both the CS and H43W mutants co-localized with catalase. Higher magnification images of the boxed regions are shown in the small panel. Scale bars, 10 μm. Arrowheads indicate representative examples of 3Flag-MITOL co-localization with catalase.

C   Correlation statistics for the localization of 3Flag-MITOL and catalase in the presence of GFP-Parkin wild-type or an inactive C431S mutant. Dots indicate individual Pearson correlation coefficient data points. In the box-plots, the center lines indicate the medians, the box limits indicate the 25th and 75th percentiles as determined in the R software package, and the whiskers extend 1.5 times the interquartile range from the 25th and 75th percentiles. Means and the number of samples are shown on the box and *X*-axis, respectively. Statistical significance was calculated using a one-tailed Welch's *t*-test. MITOL overlapped with catalase only in the presence of wild-type Parkin in wild-type HeLa cells. In *PINK1* knockout (KO) HeLa cells, wild-type Parkin was unable to induce the peroxisomal translocation of MITOL.

D   Correlation statistics for the localization of 3Flag-MITOL wild-type or inactive mutants with catalase. Dots indicate individual Pearson correlation coefficient data points. In the box-plots, the center lines indicate the medians, the box limits indicate the 25th and 75th percentiles as determined in the R software package, and the whiskers extend 1.5 times the interquartile range from the 25th and 75th percentiles. Means and the number of samples are shown on the box and *X*-axis, respectively. Statistical significance was calculated using a one-tailed Welch's *t*-test.

(KO) HeLa cells [41], the redistribution of MITOL to peroxisomes was not observed when mitochondria were damaged despite the presence of wild-type Parkin (Fig 3A, lower panel). These results were further substantiated by quantitative analysis (Fig 3C). To determine whether endogenous levels of Parkin are sufficient for the transition of MITOL from damaged mitochondria to peroxisomes, we utilized an SH-SY5Y human neuroblastoma cell line that expresses endogenous Parkin. SH-SY5Y cells transiently expressing 3Flag-MITOL were treated with valinomycin + Z-VAD-FMK for 3 or 6 h and then analyzed by immunofluorescence (note that Z-VAD-FMK is an apoptosis inhibitor that was included in the incubation because SH-SY5Y cells are more sensitive to uncoupler treatment and detach easily). As shown in Appendix Fig S3, MITOL once again redistributed to peroxisomes. However, the frequency of MITOL redistribution in SH-SY5Y cells appeared to be lower than that observed following overexpression in HeLa cells. We think this difference arises from the experimental conditions. Typically, only a minor population of mitochondria are damaged, and thus, the physiological levels of Parkin are sufficient to manage them. However, valinomycin or CCCP treatment causes depolarization of all mitochondria, and as a result, the endogenous levels of Parkin are overwhelmed. Given that the endogenous levels of Parkin in SH-SY5Y cell lines are lower than in the overexpression conditions, the endogenous Parkin might be insufficient to assist in relocation of all the MITOL molecules from damaged mitochondria to peroxisomes. However, we do observe peroxisomal localization of MITOL without exogenous Parkin expression. We thus conclude that in addition to mitochondrial membrane dissipation, both PINK1 and the E3 activity of Parkin are essential for the translocation of MITOL to peroxisomes.

**MITOL specifically relocates to peroxisomes**

Since MITOL is an E3, we next examined whether the E3 activity of MITOL itself is required for translocation to peroxisomes. To impair the MITOL E3 activity, we mutated Cys65/Cys68 within the RING domain of MITOL to Ser (CS) or the conserved Zn-binding His43 to Trp (H43W) [34,35]. Despite higher background peroxisomal signals, MITOL CS and H43W mutants localize on mitochondria. However, following 3 h of CCCP treatment, both mutants clearly co-localized with catalase (Fig 3B). Quantitative analysis confirmed the cytochemical results (Fig 3D), indicating that the redistribution of MITOL from mitochondria to peroxisomes does not require its own E3 activity.

Next, to determine whether the Parkin-dependent translocation from depolarized mitochondria to peroxisomes is limited to MITOL, the localization of other mitochondrial proteins was analyzed by immunostaining. The proteins examined included both N-terminal anchored proteins, MitoNEET/CISD1 and Tom70, and C-terminal anchored proteins Fis1 and Miro1/2 (Fig 4A). No peroxisomal translocation following CCCP treatment was observed for either N-terminal anchored or C-terminal anchored proteins (Fig 4B). The matrix protein, Hsp60, likewise did not undergo peroxisomal translocation following CCCP treatment (Fig 4B). It has been reported that Drp1, Fis1, Mitofusin (MFN) 1/2, MiD49, and Mcl1 interact with MITOL as substrates [33,34,42–44]. We consequently also examined the subcellular localization of these proteins following CCCP treatment. Because antibodies that detect endogenous MFN2, MiD49, and Mcl1 were not available, we constructed tagged plasmids. We found that none of the proteins [i.e., Drp1 (Fig EV2A), MFN1 (Fig EV2E), MFN2 (Fig EV2B), Mcl1 (Fig EV2C), and MiD49 (Fig EV2D)] was predominantly localized to peroxisomes following

**Figure 4.  MITOL specifically relocates to peroxisomes.**

A   Schematic topology representation of the mitochondrial membrane proteins analyzed in this study.

B   Peroxisomal translocation of other mitochondrial proteins such as MitoNEET/CISD1, Fis1, Miro1/2, and Tom70 (mitochondrial outer membrane proteins), and Hsp60 (mitochondrial matrix protein) was not observed in CCCP-treated cells. HeLa cells stably expressing HA-Parkin were treated with 15 μM CCCP for 3 h, and then immunostained with anti-MitoNEET/CISD1, anti-Fis1, anti-Miro1/2, anti-Tom70, anti-Hsp60, and anti-catalase antibodies. Scale bars, 10 μm.

C   A multi-spanning outer membrane protein PBR (peripheral benzodiazepine receptor) did not co-localize with catalase, but did with Tom20 during mitophagy. HeLa cells stably expressing HA-Parkin were treated with 15 μM CCCP for 3 h, and then immunostained with anti-Flag, anti-Tom20, and anti-catalase antibodies. Scale bars, 10 μm.

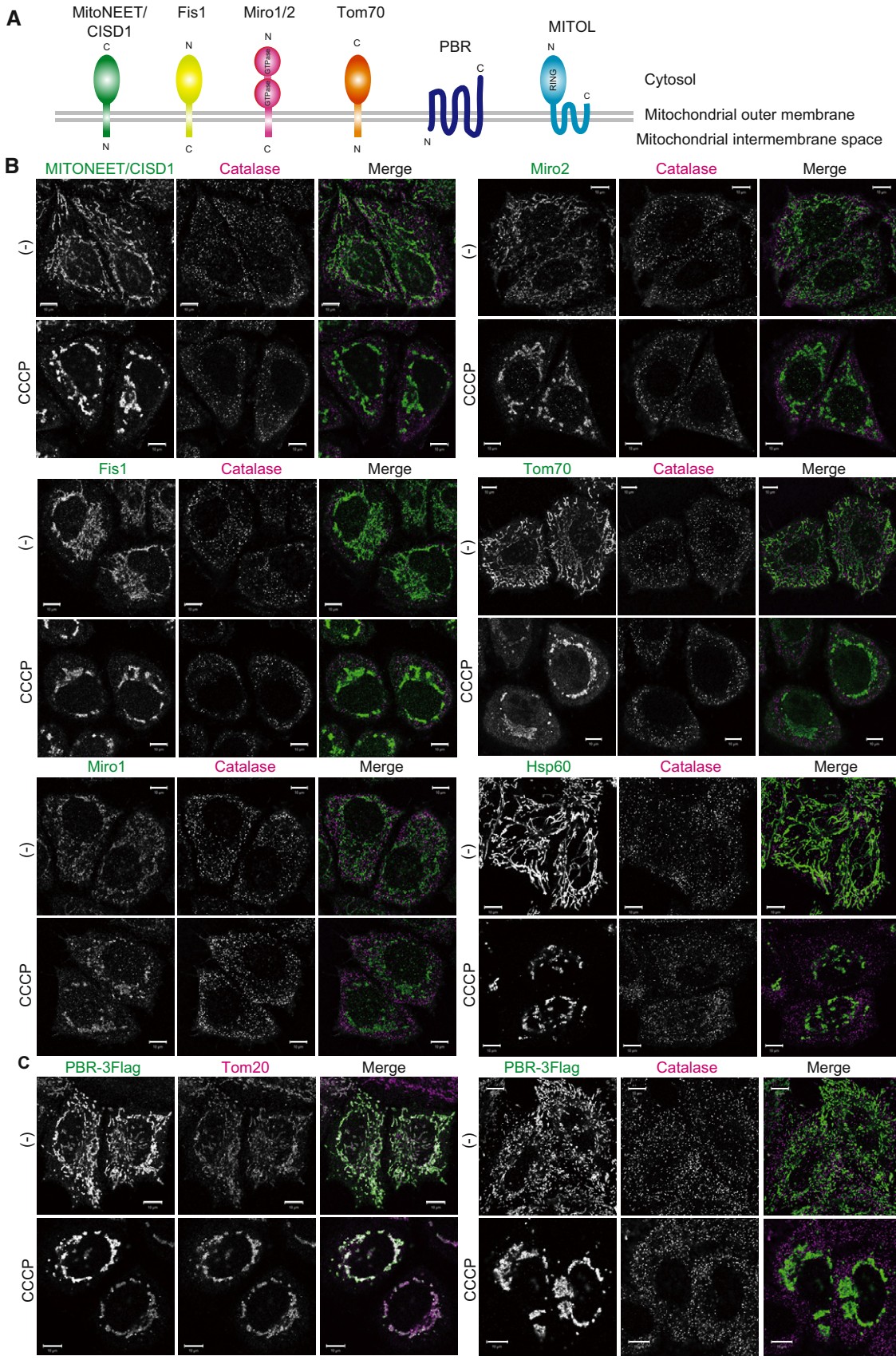

**Figure 4.**

CCCP treatment for 3 h, indicating that mitochondrial proteins other than MITOL do not relocate to peroxisomes.

As MITOL has four transmembrane domains, we also examined whether translocation to peroxisomes is a common phenomenon of mitochondrial proteins with multiple transmembrane domains. PBR (peripheral benzodiazepine receptor) is a multi-spanning outer membrane protein with five transmembrane domains that imports cholesterol to the inner membrane (Fig 4A) [45]. Overexpressed PBR-3Flag localized on mitochondria but did not undergo CCCP-induced translocation to peroxisomes (Fig 4C). Based on these immunocytochemical observations, the redistribution from mitochondria to peroxisomes is specific to MITOL.

**MITOL is targeted to mitochondria via Tom70 and is then delivered to peroxisomes**

Next, we sought to elucidate the mechanism underlying the peroxisomal targeting of MITOL. Does newly synthesized MITOL have the ability to directly integrate into the peroxisomal membrane when it fails to target to mitochondria? To examine this, siRNAs targeting several subunits of the mitochondrial outer membrane translocator (*Tom70*, *Tom20*, *Tom40*, and *Sam50*) were transfected into HeLa cells twice within a 24-h interval. At 72 h post-transfection, the cells were re-transfected with 3Flag-MITOL and Su9-GFP (the targeting sequence of the FoF1 ATPase subunit 9; a model precursor protein that targets to the matrix). Cells were then subjected to immunostaining 16 h post-plasmid transfection. Tom70, Tom20, Tom40, and Sam50 were all efficiently knocked down (Fig 5A). As expected, Su9-GFP (positive control for canonical Tom20/Tom40-dependent transport) was not imported into the mitochondria, but rather localized to the cytosol and nucleus when *Tom20* and *Tom40* were knocked down (Fig 5B). In these *Tom20*- and *Tom40*-knockdown cells, however, 3Flag-MITOL localized to mitochondria. Although *Sam50* siRNA affected cell viability, 3Flag-MITOL was still localized to mitochondria. In contrast, when *Tom70* was knocked down, 3Flag-MITOL did not localize to mitochondria, instead it was dispersed in the cytosol (Fig 5B). Consistent with this, PBR integration into the mitochondrial outer membrane was previously reported to be Tom70-dependent [45]. For statistical analysis, the number of HeLa cells exhibiting cytosolic localization of Su9-GFP or 3Flag-MITOL in each siRNA experiment was determined; a finding that confirmed the earlier results (Fig 5C). The immunocytochemistry-based cytosolic localization signal of MITOL in Tom70-knockdown cells was weak (Fig 5B), an effect that we surmise was the result of limited epitope accessibility under the immunocytochemical conditions when MITOL is released into the cytosol. However, to demonstrate cytosolic localization of MITOL biochemically, we also tried subcellular fractionation. Although a significant percentage of MITOL detaches from mitochondria during the fractionation process, a sufficient amount of MITOL was retained in the mitochondria-enriched fraction in control cells (Fig 5D, lanes 2 and 3). In contrast, almost all of the MITOL was collected in the cytosolic fraction in si*Tom70*-treated cells with no detectable signal in the mitochondria-enriched fraction (Fig 5D, lanes 5 and 6). This result further underscores that MITOL is dispersed to the cytosol when cellular levels of Tom70 are decreased. Importantly, MITOL in the *Tom70*-knockdown cells did not co-localize with catalase, indicating that the MITOL precursor, which was not targeted to the mitochondria, was unable to integrate into the peroxisomal membranes. This means that translocation of MITOL from mitochondria to peroxisomes occurs via a Parkin-mediated mitochondrial quality control pathway.

**MITOL is retained on mitochondria in peroxisome-null cells**

To examine mitophagy-induced MITOL localization in the absence of peroxisomes, we genetically generated peroxisome-depleted cells. Pex19 is essential for forming and maintaining peroxisomes [46]. To generate peroxisome-null cell lines, we used the CRISPR/Cas9 gene editing system to knock out *PEX19* (Appendix Fig S4A). We first confirmed peroxisomal loss by verifying that the catalase signal in the *PEX19* knockout HCT116 cell line was localized to the cytosol as reported previously (Fig 6A)[46,47]. Wild-type and *PEX19* knockout HCT116 cells stably expressing HA-Parkin and 3Flag-MITOL were then immunostained with anti-Flag, anti-catalase, anti-Hsp60, and anti-Tom20 antibodies. In wild-type HCT116 cells, MITOL translocation from mitochondria to peroxisomes proceeded as expected in response to mitochondrial depolarization (Fig 6B and C). In *PEX19* knockout cells, however, MITOL remained localized on mitochondria after valinomycin treatment (Fig 6B and C, and Appendix Fig S4B).

**Peroxins mediate MITOL translocation**

The above results prompted us to further analyze the molecular basis of MITOL redistribution to peroxisomes. In *PEX19* knockout

**Figure 5. MITOL is targeted to mitochondria via Tom70 and is then delivered to peroxisomes.**

A  siRNA-based knockdown of endogenous Tom70, Tom20, Tom40, and Sam50. HeLa cells treated with the corresponding siRNAs and immunoblotted with the indicated antibodies. AIF, apoptosis-inducing factor.

B  HeLa cells were immunostained with anti-Flag and anti-catalase antibodies after treatment of each siRNA. *Tom20*- and *Tom40*-siRNA treatment inhibited import of Su9-GFP (the targeting sequence of the FoF1 ATPase subunit 9) into the mitochondria, leading to an accumulation of Su9-GFP precursor proteins in the cytosol and cell nucleus. 3Flag-MITOL still localized to mitochondria in *Tom20*, *Tom40*, and *Sam50* siRNA-treated cells. In contrast, 3Flag-MITOL dispersed into the cytosol when *Tom70* was knocked down. Scale bars, 10 μm.

C  Statistical analysis of the MITOL subcellular localization following 15 μM CCCP treatment for 3 h in cells treated with *control*, *Tom70*, *Tom20*, *Tom40*, or *Sam50* siRNAs. Su9-GFP was not imported into the mitochondria, but rather localized to the cytosol and nucleus following *Tom20* and *Tom40* knockdown. The number of HeLa cells with cytosol-localized Su9-GFP or 3Flag-MITOL in each siRNA experiment was determined. Data represent the mean ± s.e.m. from > 100 cells in three biological replicates. (In case of *Tom40*, and *Sam50* siRNAs, data represent the mean ± s.e.m. from > 60 cells in three biological replicates.) Statistical significance was calculated using a one-tailed Welch's *t*-test.

D  HeLa cells treated with siControl or si*Tom70* were fractionated into cytosolic and mitochondrial fractions and then immunoblotted with anti-lactate dehydrogenase (LDH), anti-Tom20, and anti-Flag antibodies. LDH was used as a cytosolic marker. Although some MITOL detached from mitochondria during fractionation, sufficient amounts of MITOL were collected in the mitochondria-enriched fraction in control cells. In contrast, almost all of the MITOL was collected in the cytosolic fraction in si*Tom70*-treated cells.

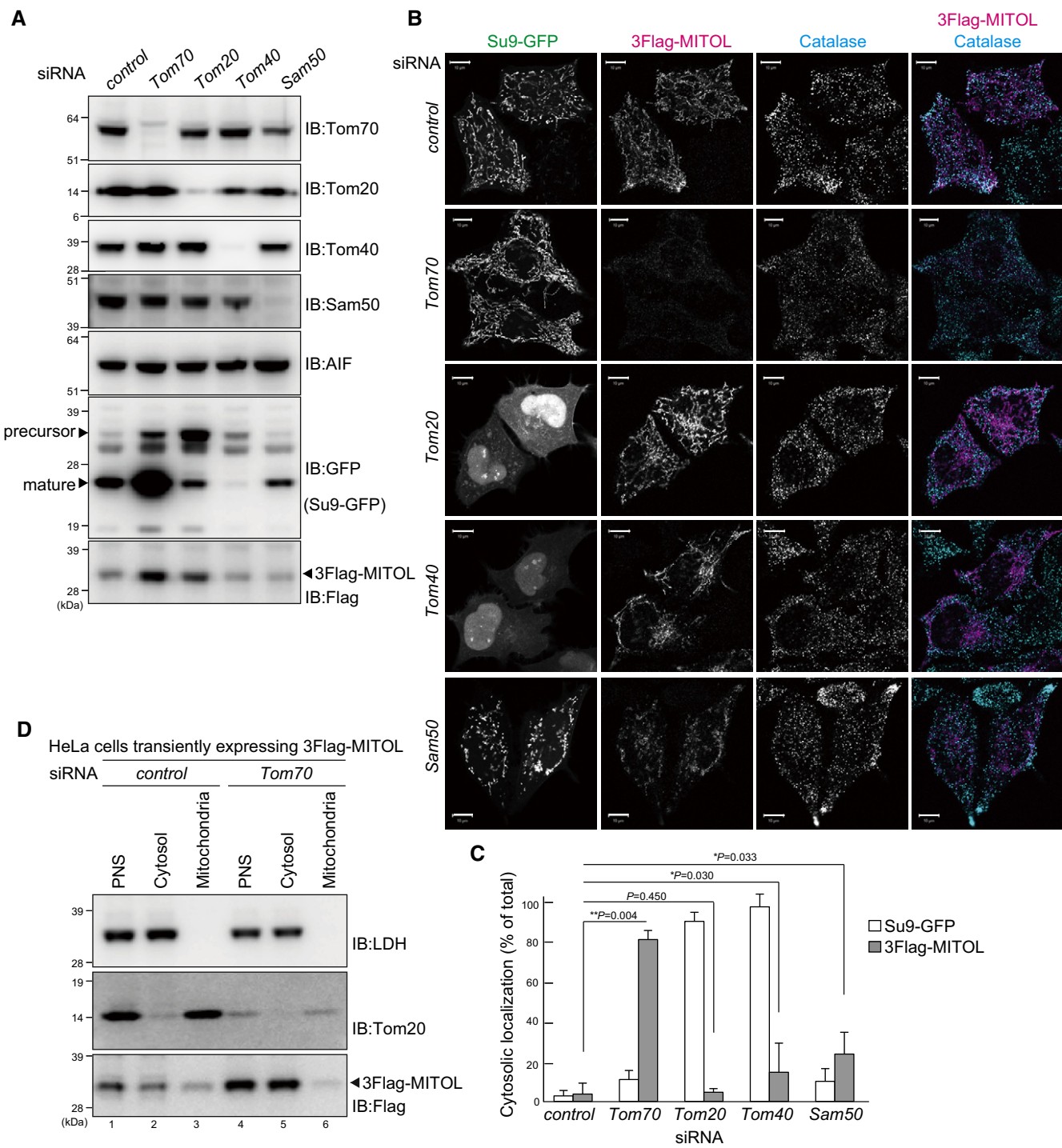

**Figure 5.**

cells, peroxisomes were lost and thus the catalase-positive dot-like signal disappeared (Fig 6A). Under these conditions, even if MITOL is retained on mitochondria (Fig 6C), it is difficult to interpret whether Pex19 is important for MITOL translocation to peroxisomes or whether the mitochondrial localization of MITOL is attributable to the loss of the final destination peroxisomes. To better address this question, we performed knockdown experiments, which resulted in peroxisomal retention and an observable catalase signal (Fig 7C). In

mammalian cells, the targeting of newly synthesized peroxisomal membrane proteins utilizes either a Pex19/Pex3 (class I pathway) or Pex19/Pex16 (class II pathway) [48–50]. Pex3 on the peroxisomal membrane functions as a receptor for cytosolic Pex19, which associates with a newly synthesized substrate (class I pathway). On the other hand, targeting of newly synthesized Pex3 to peroxisomes requires Pex19 and Pex16 (class II pathway). To understand which pathway mediates MITOL translocation, we examined MITOL

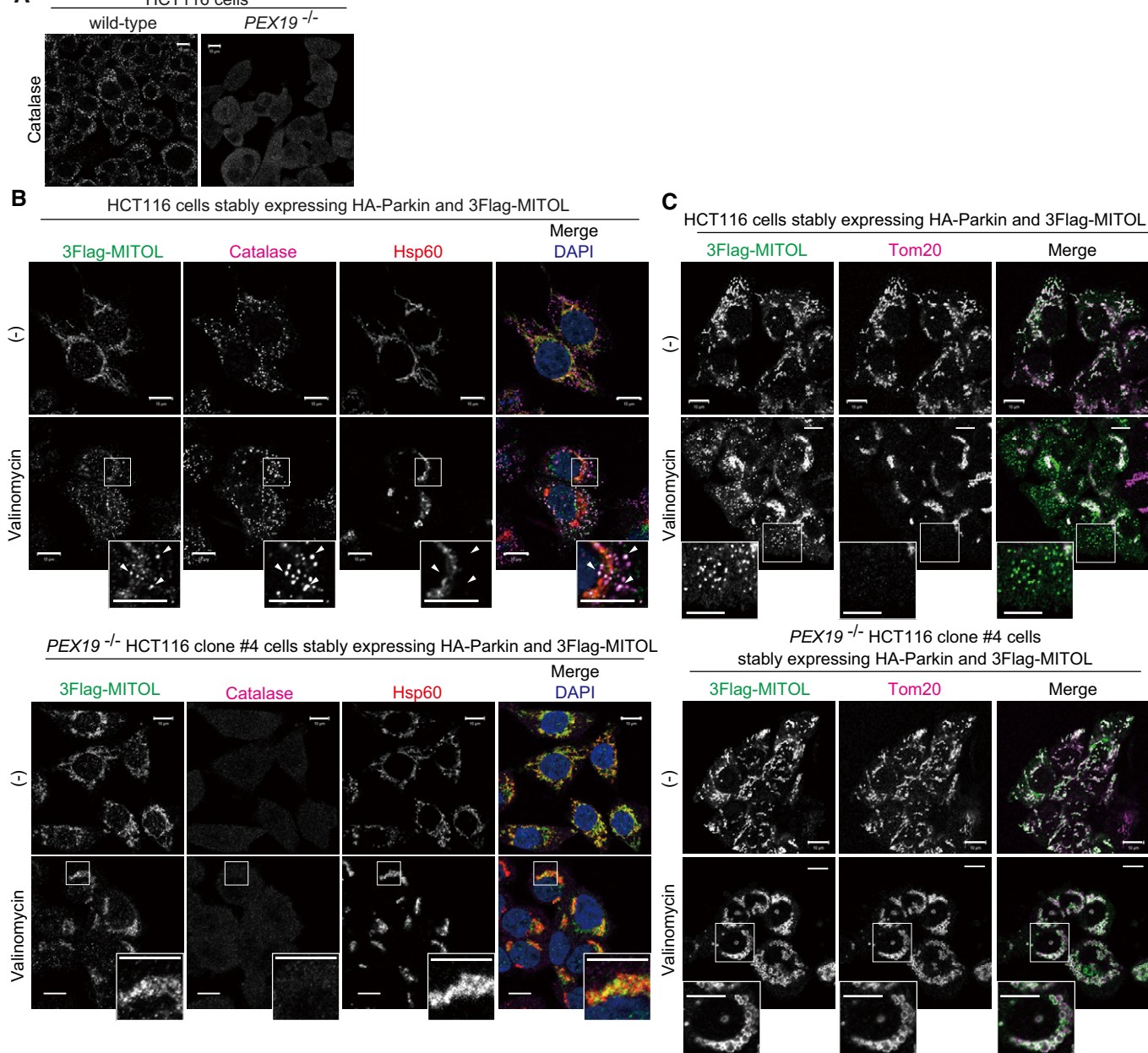

**Figure 6.** **MITOL retained on mitochondria in peroxisome-null cells.**

A   CRISPR/Cas9 gene editing was used to generate peroxisome-null cell lines by knocking out *PEX19*. In wild-type HCT116 cells, catalase localized to peroxisomes (dot-like structures), whereas in *PEX19*$^{-/-}$ cells catalase was diffusely localized throughout the cytosol, indicating loss of peroxisomes. Scale bars, 10 μm.

B, C   Wild-type and *PEX19*$^{-/-}$ HCT116 cells stably expressing HA-Parkin and 3Flag-MITOL were treated with 10 μM valinomycin for 3 h, and then subjected to immunocytochemistry with anti-Flag, anti-catalase, and anti-Hsp60 (B), or anti-Flag and anti-Tom20 (C) antibodies. In wild-type HCT116 cells, MITOL translocated to peroxisomes, whereas MITOL remained associated with mitochondria in *PEX19*$^{-/-}$ cells. Arrowheads indicate representative examples of 3Flag-MITOL co-localization with catalase rather than Hsp60 in wild-type HCT116 cells.

Data information: Higher magnification images of the boxed regions are shown in the small panel. Scale bars, 10 μm.

localization in cells transfected with *PEX3*, *PEX16*, or *PEX19* siRNAs. We confirmed that Pex3, Pex16, and Pex19 were reduced in the corresponding siRNA-treated cells (Fig 7A), and MITOL amounts were not affected in *PEX3-*, *PEX16-*, and *PEX19*-knockdown cells (Fig 7B). siRNA-mediated knockdown of *PEX3*, *PEX16*, or *PEX19* had

no effect on MITOL mitochondrial integration under steady-state conditions. Importantly, however, knockdown of *PEX3*, *PEX16*, or *PEX19* impaired translocation of MITOL to peroxisomes following CCCP treatment (Fig 7C). Quantification analysis of co-localization between 3Flag-MITOL and catalase as Pearson correlation

coefficients revealed that the peroxisomal translocation of MITOL in control siRNA cells (Fig 7D, columns 1 and 2) was decreased in si*PEX3*- and si*PEX16*-treated cells with *PEX3* knockdown having the

most pronounced inhibitory effects (Fig 7D, columns 3 and 4). Indeed, in si*PEX3*-treated cells, no significant difference ($P = 0.16$) in the co-localization of 3Flag-MITOL and catalase was observed

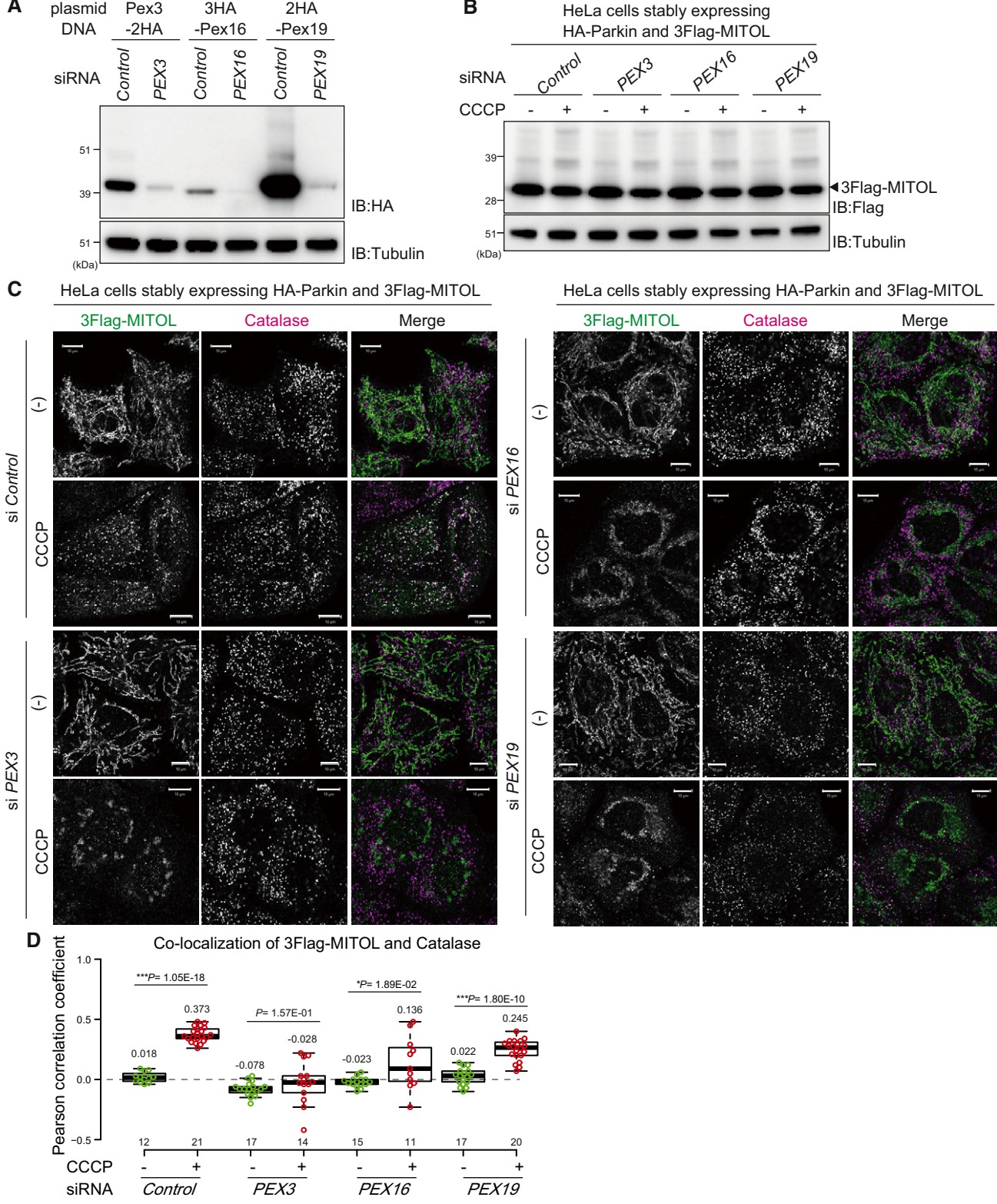

**Figure 7.**

**Figure 7. MITOL translocation to peroxisomes is mediated by peroxins.**

A    siRNA-based knockdown of Pex3, Pex16, and Pex19. HeLa cell lysates treated with the indicated plasmids and siRNAs were immunoblotted with the indicated antibodies.

B    A comparable amount of MITOL was detected in *control*, *PEX3*-, *PEX16*-, and *PEX19*-knockdown cells. HeLa cell lysates stably expressing HA-Parkin and 3Flag-MITOL treated with the corresponding siRNAs were immunoblotted with the indicated antibodies.

C    MITOL translocation from the mitochondria to peroxisomes is highly dependent on Pex3. HeLa cells stably expressing 3Flag-MITOL and HA-Parkin were transfected with control, *PEX3*, *PEX16*, or *PEX19* siRNA, treated with 15 μM CCCP for 3 h, and then subjected to immunocytochemistry with anti-Flag and anti-catalase antibodies. Scale bars, 10 μm.

D    Correlation statistics for the localization of 3Flag-MITOL and catalase in cells transfected with the indicated siRNAs. Dots indicate individual Pearson correlation coefficient data points. In the box-plots, the center lines indicate the medians, the box limits indicate the 25th and 75th percentiles as determined in the R software package, and the whiskers extend 1.5 times the interquartile range from the 25th and 75th percentiles. Means and the number of samples are shown on the box and *X*-axis, respectively. Statistical significance was calculated using a one-tailed Welch's *t*-test.

between CCCP-treated and untreated conditions. In *PEX19*-silenced cells, the co-localization between MITOL and catalase was slightly impaired, but co-localization of 3Flag-MITOL and catalase was significantly increased in CCCP-treated cells (Fig 7D, columns 7 and 8). These results indicate that MITOL translocation from mitochondria to peroxisomes depends on Pex3 activity.

## MITOL is extracted from mitochondria in a p97/VCP ATPase-dependent manner

Since MITOL is a multi-spanning protein, we hypothesized that MITOL is selectively extracted from the mitochondrial outer membrane following mitophagy stimulation. To address this, we focused on p97/VCP. p97/VCP is a well-conserved abundant hexameric ring-shaped AAA$^+$-ATPase that, with differing cofactors, functions in multiple cellular events such as proteasome-dependent protein degradation, membrane fusion, trafficking, autophagy, and genomic DNA surveillance [51,52]. Since p97/VCP is required for proteasomal degradation of the mitochondrial outer membrane proteins mitofusin1 and mitofusin2 following Parkin-mediated ubiquitylation [22], p97/VCP might also play a role in the mitochondrial extraction of MITOL. We first knocked down *p97/VCP* and examined the subcellular localization of MITOL. In control cells, CCCP-dependent co-localization of MITOL and catalase was observed, whereas in *p97/VCP* knockdown cells MITOL predominantly overlapped with Hsp60 (Fig 8A and B). Because of cell viability issues, however, complete p97/VCP depletion was not achieved. Therefore, to obtain more robust data, we used p97/VCP ATPase mutants and

an allosteric chemical inhibitor. Overexpression of p97/VCP E305Q/E578Q (p97QQ), which is known to have a dominant-negative effect [53], inhibited MITOL redistribution from mitochondria to peroxisomes, while overexpression of wild-type p97/VCP did not suppress MITOL redistribution (Fig 8C). Furthermore, in the presence of the specific p97/VCP inhibitor NMS-873, mitochondrial depolarization-induced MITOL translocation was blocked (Fig 8D). Quantification analysis confirmed that the translocation of MITOL to peroxisomes was completely inhibited by the specific p97/VCP inhibitor NMS-873 (Fig 8E). These results revealed that MITOL is extracted from mitochondria in a p97/VCP ATPase-dependent manner.

## Ubiquitylation of C-terminal K268 is required for MITOL relocation to peroxisomes

We have shown that the translocation of MITOL to peroxisomes depends on Parkin, a decrease in mitochondrial membrane potential, and p97/VCP. We next examined whether Parkin-mediated ubiquitylation of MITOL is essential for translocation. To detect MITOL ubiquitylation, total cell lysates from HeLa cells stably expressing HA-Parkin and 3Flag-MITOL after CCCP treatment were immunoblotted with an anti-Flag antibody. Initially, a ubiquitylation-derived molecular weight shift of MITOL was not observed clearly (Fig EV3A, lanes 1 and 2). However, it is possible that the ubiquitylation of MITOL that occurs during mitochondrial localization is removed following translocation to peroxisomes. We thus added NMS-873 (p97/VCP inhibitor used in Fig 8D) to retain MITOL on the depolarized mitochondria, and observed a ubiquitylation-like

**Figure 8. MITOL is extracted from mitochondria in a p97/VCP ATPase-dependent manner.**

A    MITOL did not merge with Hsp60 in control siRNA-treated cells, whereas most MITOL co-localized with Hsp60 in *p97/VCP* knockdown cells in response to mitophagy stimuli. HeLa cells stably expressing 3Flag-MITOL and HA-Parkin were transfected with *control* or *p97/VCP* siRNA, treated with 15 μM CCCP for 3 h, and then subjected to immunocytochemistry with anti-Flag, anti-catalase, and anti-Hsp60 antibodies. Scale bars, 10 μm.

B    HeLa cell lysates treated with *control* or *p97/VCP* siRNA were immunoblotted with anti-p97/VCP and anti-tubulin antibodies.

C    Overexpression of a p97/VCP ATP hydrolysis-defective mutant, E305Q/E578Q (p97QQ), blocked MITOL redistribution from mitochondria to peroxisomes, while overexpression of wild-type p97/VCP had no effect on MITOL redistribution. HeLa cells stably expressing GFP-Parkin were transfected with MITOL-HA and Flag-p97/VCP wild-type or p97QQ, treated with 15 μM CCCP for 3 h, and then subjected to immunocytochemistry with anti-HA and anti-catalase antibodies. Higher magnification images of the boxed regions are shown in the small panel. Scale bars, 10 μm.

D    NMS-873, a specific inhibitor of p97/VCP, prevented MITOL translocation following CCCP treatment. HeLa cells stably expressing HA-Parkin were transfected with 3Flag-MITOL, treated with 15 μM CCCP in the presence or absence of 10 μM NMS-873 for 3 h, and then subjected to immunostaining with anti-Flag, anti-catalase, and anti-HA antibodies. Higher magnification images of the boxed regions are shown in the small panel. Scale bars, 10 μm. Arrowheads in (C) and (D) indicate representative examples of MITOL–peroxisome co-localization that was only observed in the presence of a functional VCP.

E    Correlation statistics for the localization of 3Flag-MITOL and catalase in the presence of NMS-873. Dots indicate individual Pearson correlation coefficient data points. In the box-plots, the center lines indicate the medians, the box limits indicate the 25th and 75th percentiles as determined in the R software package, and the whiskers extend 1.5 times the interquartile range from the 25th and 75th percentiles. Means and the number of samples are shown on the box and *X*-axis, respectively. Statistical significance was calculated using a one-tailed Welch's *t*-test.

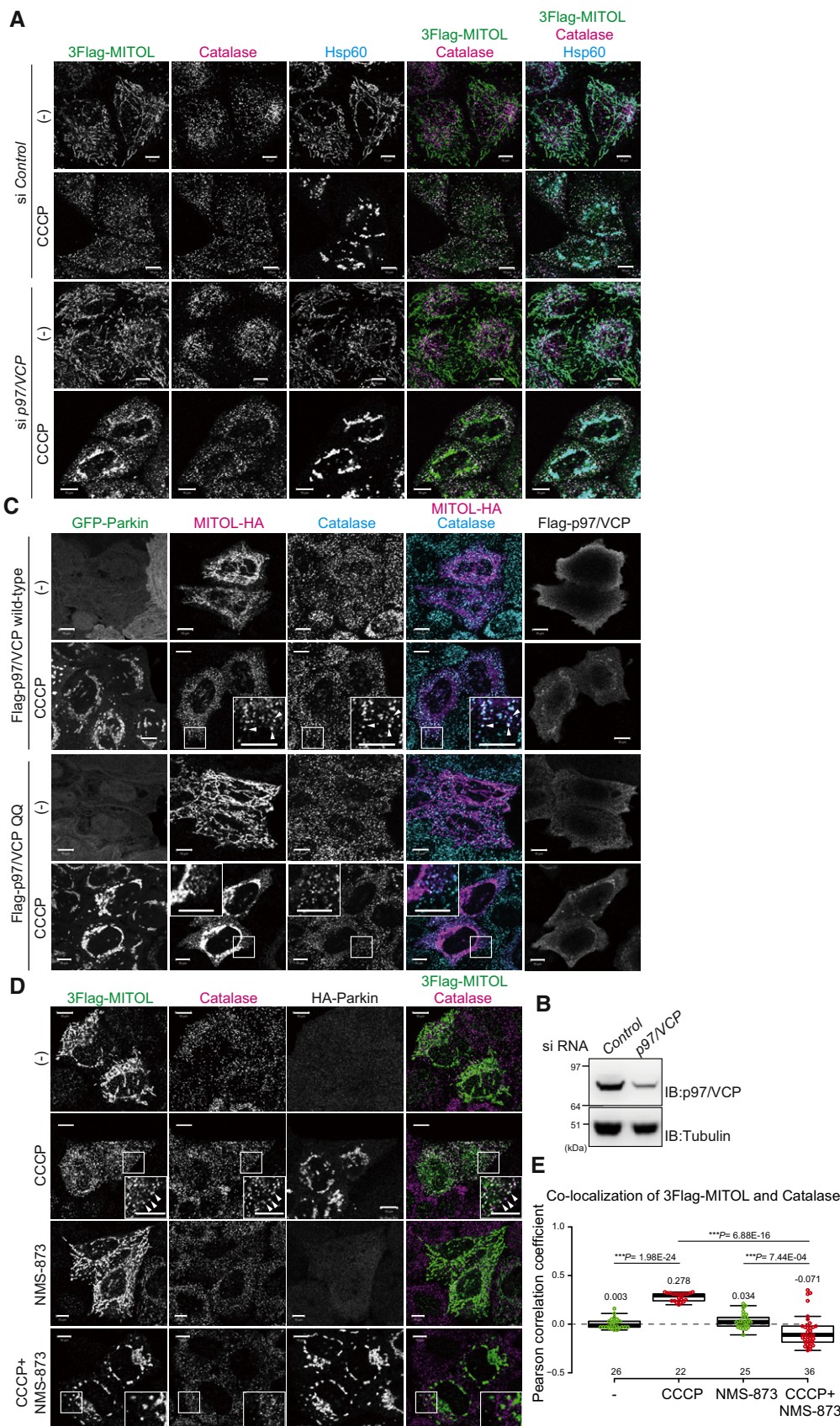

**Figure 8.**

molecular weight shift of 3Flag-MITOL (Fig EV3A, lanes 3 and 4). HeLa cells stably expressing 3Flag-MITOL were immunoprecipitated with an anti-Flag antibody and then immunoblotted with anti-Flag and anti-ubiquitin antibodies. Although MITOL was ubiquitylated following CCCP and NMS-873 treatment in Parkin-expressing cells (Fig 9A, lanes 3 and 4), no ubiquitylation signal was observed in wild-type HeLa cells lacking endogenous Parkin under the same experimental conditions (Fig 9A, lanes 1 and 2). This result confirms that Parkin ubiquitylates MITOL in response to a decrease in the mitochondrial membrane potential.

As indicated previously, clear ubiquitylation was observed following CCCP and NMS-873 treatment (Fig EV3), suggesting that the MITOL ubiquitylation that is mediated on depolarized mitochondria is removed on peroxisomes. It is thus of interest whether MITOL needs to be de-ubiquitylated for peroxisomal localization. Given that the de-ubiquitylating enzyme USP30 localizes to both mitochondria and peroxisomes and is involved in Parkin-mediated mitophagy [54–57], we examined whether USP30 also contributes to MITOL translocation. The ubiquitylation status of MITOL in USP30 knockout HeLa cells (Appendix Fig S5, lanes 4–9) was almost identical to that in wild-type HeLa cells (Appendix Fig S5, lanes 1–3). In addition, following CCCP treatment for 1.5 and 3 h, MITOL translocated equally from mitochondria to peroxisomes in both wild-type and USP30 knockout HeLa cell (Appendix Fig S6). These results suggest that at least USP30-catalyzed de-ubiquitylation is not involved in MITOL translocation.

We next tried to determine the ubiquitylation site of MITOL. Using an LC-MS/MS-based analysis, we identified K40, K54, and K268 as ubiquitin modified sites in the MITOL sequence following valinomycin treatment (Fig EV3B). To examine whether these ubiquitylations are required for MITOL translocation, MITOL K268 was mutated to Ala. The K268A mutant targeted to mitochondria under steady-state conditions, but translocation to peroxisomes upon CCCP treatment was disturbed (Fig 9C). Because the K-to-A mutation not only inhibits ubiquitylation but also changes the charge and bulkiness of the amino acid side chain, we also examined the effect of a K268R mutation that maintains both properties, but that hampers ubiquitylation. Immunocytochemical analysis showed that neither the K268A nor the K268R mutant translocated to peroxisomes (Fig 9C), a conclusion that was supported by quantitative

analyses (Fig 9E). These results suggest that ubiquitylation of K268 is important for Parkin-induced redistribution of MITOL to peroxisomes. Since mass spectrometry analysis revealed that K40 and K54 in the RING domain were also ubiquitylated in response to mitophagy, we made similar mutations to K40 and K54. Translocation of MITOL to peroxisomes was slightly impeded by the K54A and K54R mutations (Fig EV4), although the effect was weaker than that seen with the K268A and K268R mutations. The K40A and K40R mutations had little effect on peroxisomal redistribution of MITOL following CCCP treatment (Fig EV4). A combination of the K54 and K268 mutations failed to generate results that differed significantly from the K268 mutation alone (Fig 9D). While clear ubiquitylation of wild-type MITOL was observed following CCCP and NMS-873 treatment (Fig 9B, lanes 1 and 2), immunoprecipitation analysis revealed that the ubiquitylation ladder was not formed on the MITOL K268A and K54A/K268A mutants (Fig 9B, lanes 3–6). The Lys residue at this position (268 in humans) is well conserved from mammals to zebra fish, suggesting its physiological importance (Fig EV3D). LC-MS/MS-based analysis showed that the abundance ratio of K268 ubiquitylation in valinomycin-treated samples was about 28 times higher than in non-treated control samples (Figs 9F and EV3C); this ratio was less pronounced at K40 and K54. Taken together, these data indicate that the peroxisomal targeting of MITOL from damaged mitochondria requires Parkin-mediated ubiquitylation of K268 as a post-translational modification.

We next investigated whether the translocation-defective mutant of MITOL represses mitophagy progression or not. To quantify mitophagy by flow cytometry, we utilized HCT116 cells stably expressing both Parkin and the mitochondria-localized pH-dependent fluorescent protein mt-Keima, which has a short excitation wavelength under neutral conditions (e.g., in mitochondria) and undergoes a shift to longer excitation wavelengths under acidic conditions (e.g., in lysosomes) [58]. Consequently, mitophagy progression can be monitored as a proportion of cells in which mt-Keima has undergone an acidification-specific fluorescence change. We then compared the mitophagy activity of MITOL knockout HCT116 cells complemented with wild-type MITOL (which can translocate to peroxisomes) and MITOL knockout cells complemented with the K268A MITOL mutant (which is defective in peroxisomal

**Figure 9. Ubiquitylation of K268 in the MITOL C-terminus is required for relocation to peroxisomes.**

A  MITOL was ubiquitylated only in the presence of Parkin when the mitochondrial membrane was decreased. After 15 μM CCCP treatment for 3 h, HeLa cells stably expressing HA-Parkin and 3Flag-MITOL were immunoprecipitated with anti-Flag magnetic beads, and then immunoblotted with the indicated antibodies. Red bars indicate ubiquitylation; the black arrowhead indicates 3Flag-MITOL.

B  The characteristic ubiquitylation ladder was observed for wild-type MITOL, but was absent in the K268A and K54A/K268A mutants. Black arrowhead indicates 3Flag-MITOL.

C  The MITOL K268A and K268R mutants were targeted to mitochondria under steady-state conditions, whereas peroxisomal localization following CCCP treatment was considerably disrupted. HeLa cells stably expressing HA-Parkin were transfected with 3Flag-MITOL wild-type, K268A, or K268R mutants, treated with 15 μM CCCP for 3 h, and then subjected to immunostaining with anti-Flag and anti-catalase antibodies. Scale bars, 10 μm.

D  The double K54/K268 mutation did not change the MITOL localization pattern. The subcellular localization of the MITOL K268A or K268R mutants was not drastically changed by inclusion of the K54A or K54R mutations. Scale bars, 10 μm.

E  Correlation statistics for the localization of 3Flag-MITOL wild-type, K268A, or K268R mutants with catalase. Dots indicate individual Pearson correlation coefficient data points. In the box-plots, the center lines indicate the medians, the box limits indicate the 25th and 75th percentiles as determined in the R software package, and the whiskers extend 1.5 times the interquartile range from the 25th and 75th percentiles. Means and the number of samples are shown on the box and X-axis, respectively. Statistical significance was calculated using a one-tailed Welch's t-test.

F  The fold change in ubiquitylation of MITOL K268, K40, and K54 in valinomycin-treated samples versus untreated samples. After 3 h of valinomycin treatment, $PEX19^{-/-}$ HCT116 cells stably expressing HA-Parkin and 3Flag-MITOL were immunoprecipitated with anti-Flag magnetic beads, and then subjected to LC-MS/MS analysis for label-free quantification of ubiquitylated peptides. Error bars represent the mean ± s.e.m. in three biological replicates.

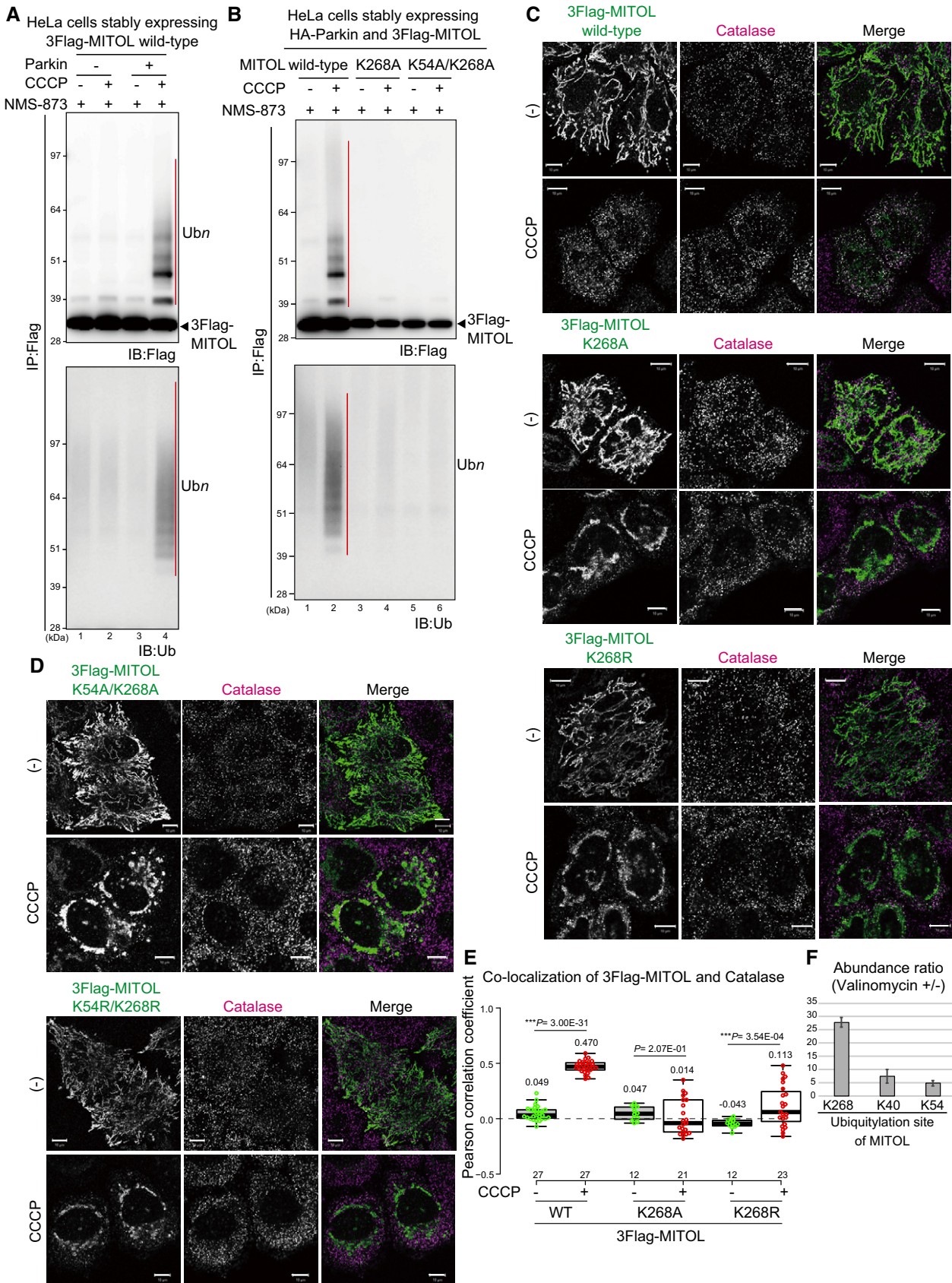

**Figure 9.**

translocation). The mitophagic flux in ΔMITOL cells complemented with MITOL K268A was equivalent to that in ΔMITOL cells complemented with wild-type MITOL, suggesting that the retention of MITOL on depolarized mitochondria does not inhibit mitophagy progression itself (Fig EV5A and B).

**Endogenous MITOL translocates to peroxisomes**

Since the redistribution of MITOL was observed under overexpressed conditions, we next sought to confirm whether endogenous MITOL also displayed peroxisomal localization in response to mitochondrial depolarization. However, specific antibodies suitable for immunocytochemical detection of endogenous MITOL are not available. To address this issue, we established a MITOL-3Flag knock-in (KI) cell line. Neomycin- or hygromycin-resistant markers including *3xFlag* gene cassettes and the short homologous arms were inserted upstream of the *MITOL* stop codon by CRISPR/Cas9-based genome editing ([59] and see Materials and Methods for details). G418 and hygromycin B-resistant single clones were isolated and the appropriate 3Flag-tag insertion was verified by immunoblotting with an anti-Flag antibody (Fig 10A). We observed MITOL subcellular localization in MITOL-3Flag KI HCT116 cells stably expressing HA-Parkin using anti-Flag, anti-catalase, and anti-Hsp60 antibodies. 3Flag-tagged endogenous MITOL co-localized with Hsp60 under steady-state conditions, whereas valinomycin treatment induced translocation of 3Flag-tagged endogenous MITOL from mitochondria to peroxisomes (Fig 10B) similar to that observed with overexpressed MITOL. Quantitative analysis confirmed that co-localization of endogenous MITOL and catalase increased significantly in response to mitophagy stimulation, although a small percentage of 3Flag-tagged endogenous MITOL localized on peroxisomes prior to valinomycin treatment (Fig 10C). The collected data indicate that endogenous MITOL also redistributes from mitochondria to peroxisomes in response to mitochondrial depolarization.

In Fig 8D, we showed that translocation of exogenous MITOL to peroxisomes was blocked by the VCP inhibitor, NMS-873. We next examined whether cells harboring 3Flag-tagged endogenous MITOL would yield similar results. When MITOL-3Flag KI HCT116 cells

stably expressing HA-Parkin were treated with valinomycin for 3 h, 3Flag-tagged endogenous MITOL overlapped with PMP70 (peroxisomal membrane protein 70). In contrast, the combined 3Flag-tagged endogenous MITOL did not redistribute to peroxisomes following the administration of NMS-873 and valinomycin (Fig 10D and E). As shown in Fig 9A and B, exogenous MITOL is ubiquitylated prior to peroxisomal translocation. We thus sought to determine whether endogenous MITOL was ubiquitylated as well. MITOL-3Flag KI HCT116 cells stably expressing HA-Parkin were treated with valinomycin and NMS-873 for 3 h, immunoprecipitated using anti-Flag magnetic beads, and immunoblotted using anti-Flag and anti-ubiquitin antibodies. The immunoblots indicate that 3Flag-tagged endogenous MITOL is also ubiquitylated in response to a loss in the mitochondrial membrane potential (Fig 10F). These data confirm that key aspects of the MITOL translocation pathway (i.e., VCP- and ubiquitin-dependent translocation from depolarized mitochondria to peroxisomes) are relevant for both overexpressed and endogenous MITOL.

**MITOL levels in peroxisome-enriched cellular subfractions are increased following mitochondrial depolarization**

To quantify the subcellular localization of MITOL, high-speed fractionation of cultured cells was performed. Although initially challenging (similar densities between mitochondria and peroxisomes and methodology-based detachment of MITOL), useful fractions were obtained by adding NMS-873 (VCP inhibitor) to the lysis buffer and subjecting the resultant lysates to a multi-step centrifugation process. The post-nuclear supernatant (PNS) collected from HCT116 cells stably expressing HA-Parkin and 3Flag-MITOL was centrifuged at 3,000 $g$ to obtain a "3,000 $g$ pellet" fraction with the resulting supernatant further centrifuged at 100,000 $g$ to yield a "100,000 $g$ pellet" fraction. Mitochondria were recovered in the 3,000 $g$ pellet, whereas the 100,000 $g$ pellet was almost free of mitochondria (Fig 11A, Tom20 signal of lanes 2 and 3). Peroxisomes were collected in both fractions but were enriched in the 100,000 $g$ pellet (Fig 11A, PMP70 signal of lanes 2 and 3). Using these two fractions, we compared the recovery ratio of 3Flag-MITOL. Before valinomycin

---

**Figure 10. Endogenous MITOL translocates to peroxisomes.**

A   To generate MITOL-3Flag knock-in (KI) HCT116 cell lines, *3xFlag* gene cassettes were inserted upstream of the *MITOL* stop codon using CRISPR/Cas9-based gene editing. Insertion of the 3Flag-tag was verified by immunoblotting with an anti-Flag antibody. Asterisk indicates a cross-reacting band.
B   The MITOL subcellular localization was observed in MITOL-3Flag KI HCT116 cells stably expressing HA-Parkin with anti-Flag, anti-catalase, and anti-Hsp60 antibodies. Endogenous MITOL (detectable with an anti-Flag antibody) overlapped with Hsp60 under steady-state conditions, whereas 3 h of valinomycin (10 μM) treatment induced translocation of endogenous MITOL from mitochondria to peroxisomes. Higher magnification images of the boxed regions are shown in the bottom panel. Scale bars, 10 μm.
C   Correlation statistics for the localization of endogenous MITOL-3Flag with catalase. Dots indicate individual Pearson correlation coefficient data points. In the box-plots, the center lines show the medians, box limits indicate the 25th and 75th percentiles as determined by the R software package, and whiskers extend 1.5 times the interquartile range from the 25th and 75th percentiles. Means and the number of samples are shown on the box and the *X*-axis, respectively. Statistical significance was calculated using a one-tailed Welch's *t*-test.
D   The extraction of endogenous MITOL from depolarized mitochondria is blocked by the p97/VCP inhibitor NMS-873. Higher magnification images of the boxed regions are shown in the lower panel. Scale bars, 10 μm.
E   Correlation statistics for the localization of endogenous MITOL and PMP70 in the presence of NMS-873. Dots indicate individual Pearson correlation coefficient data points. In the box-plots, the center lines show the medians, box limits indicate the 25th and 75th percentiles as determined by the R software package, and whiskers extend 1.5 times the interquartile range from the 25th and 75th percentiles. Means and the number of samples are shown on the box and the *X*-axis, respectively. Statistical significance was calculated using a one-tailed Welch's *t*-test.
F   Endogenous MITOL is ubiquitylated following valinomycin treatment. After MITOL-3Flag KI HCT116 cells stably expressing HA-Parkin were treated with valinomycin for 3 h, the collected cell lysates were immunoprecipitated with anti-Flag magnetic beads. The immunoprecipitates were blotted using anti-Flag and anti-ubiquitin antibodies. Red bars indicate ubiquitylation; the black arrowhead indicates MITOL-3Flag.

---

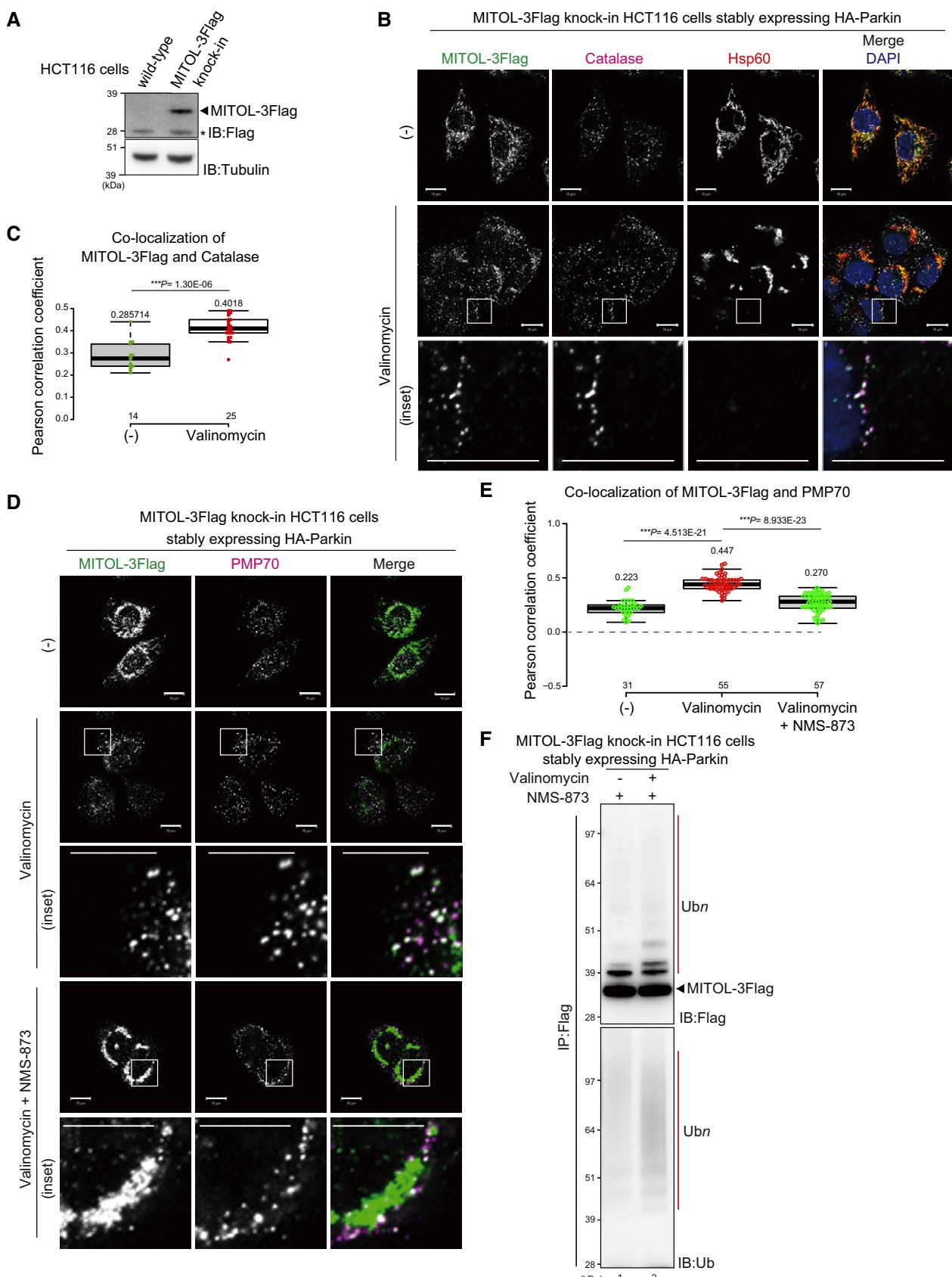

**Figure 10.**

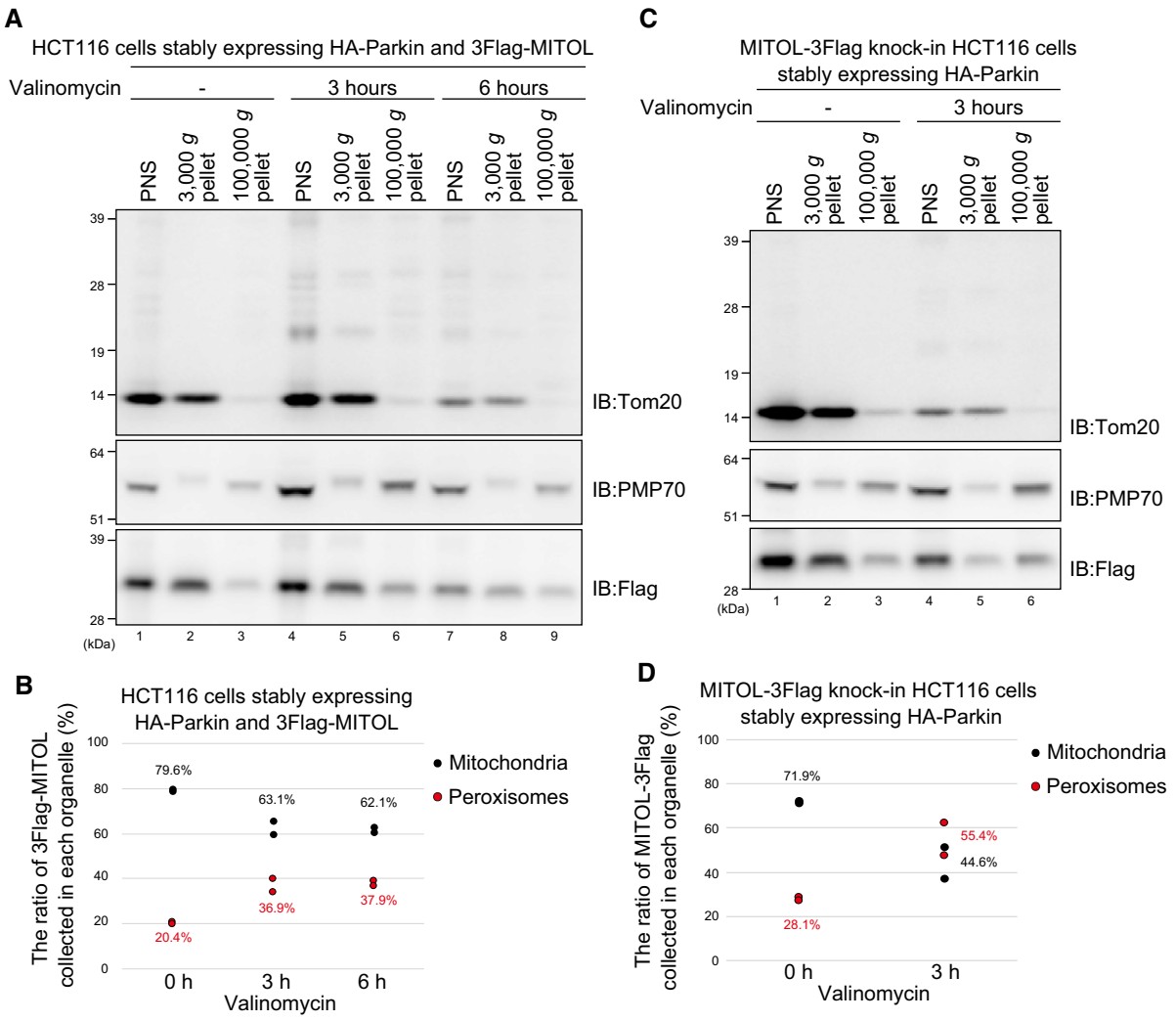

**Figure 11. Subcellular fractionation confirms the redistribution MITOL from depolarized mitochondria to peroxisomes.**

A    Distribution of exogenous 3Flag-MITOL in the mitochondria-rich or peroxisome-rich fraction following cellular fractionation. HCT116 cells stably expressing HA-Parkin and 3Flag-MITOL were subjected to fractionation and detected using anti-Tom20, anti-PMP70, and anti-Flag antibodies. Tom20 and PMP70 were used as mitochondrial and peroxisomal markers, respectively. The 3,000 *g* and 100,000 *g* pellets represent the mitochondria-rich and peroxisome-rich fractions.

B    The recovery ratio of 3Flag-MITOL between the mitochondria-enriched (3,000 *g* pellet) and peroxisome-enriched fraction (100,000 *g* pellet) following valinomycin treatment for the indicated times. The ratio of peroxisome-localized 3Flag-MITOL to mitochondria-localized 3Flag-MITOL increased with valinomycin treatment for 3 h. Graphic data represent results of two biological replicates. In scatter plot, dots indicate individual data points. Black dots indicate the ratio of 3Flag-MITOL collected in the mitochondria-enriched fractions, and red dots are the ratio of 3Flag-MITOL collected in the peroxisome-enriched fractions. Mean values are also shown.

C, D    Distribution of endogenous MITOL in the mitochondria-rich or peroxisome-rich fraction following cellular fractionation. The distribution of endogenous MITOL was examined as in (A) and (B) using MITOL-3Flag knock-in HCT116 cells. Treatment of cells with valinomycin for 3 h reduced the amount of endogenous MITOL in the mitochondria-enriched fraction, but concomitantly increased endogenous MITOL in the peroxisome-enriched fraction. Graphic data represent results of two biological replicates. In scatter plot, dots indicate individual data points. Black dots indicate the ratio of MITOL-3Flag collected in mitochondria-enriched fractions, and red dots are the ratio of MITOL-3Flag collected in the peroxisome-enriched fractions. Mean values are also shown.

treatment, a large proportion (ca. 80%) of the 3Flag-MITOL was collected in the 3,000 *g* pellet (i.e., mitochondria-enriched fraction) and a minor proportion (ca. 20%) in the 100,000 *g* pellet (i.e., peroxisome-enriched fraction; Fig 11B). In contrast, the ratio of 3Flag-MITOL recovered in the peroxisome-rich 100,000 *g* pellet increased from 20.4% to 36.9% following valinomycin treatment for 3 h (Fig 11B). This result supports our model that MITOL redistributes from damaged mitochondria to peroxisomes.

To quantify the proportion of endogenous MITOL in mitochondria/peroxisomes, we repeated the cellular fractionation using MITOL-3Flag knock-in HCT116 cells stably expressing HA-Parkin. As in the overexpressed MITOL analysis (Fig 11A and B), we compared the ratio of endogenous MITOL recovered in the 3,000 *g* and 100,000 g pellets following valinomycin treatment. Endogenous MITOL was largely (ca. 72%) detected in the mitochondria-enriched 3,000 *g* pellet (Fig 11C, lane 2) with less (ca. 28%) recovered in the

peroxisome-enriched 100,000 *g* pellet prior to valinomycin treatment (Fig 11D). Treatment of cells with valinomycin for 3 h caused the proportion of endogenous MITOL localized on mitochondria to decrease from 72% to 45% (Fig 11C, lane 5), but to increase from 28% to 55% in the peroxisome-enriched 100,000 *g* pellet (Fig 11C, lane 6). These results confirm our immunocytochemical data indicating that endogenous MITOL also redistributes to peroxisomes following valinomycin treatment.

### MITOL lacking the eight most C-terminal amino acids (MITOLΔC8) causes peroxisome expansion following CCCP treatment

Then what is the functional relevance of MITOL translocation to peroxisomes? It is unlikely that MITOL is transported to peroxisomes because it is cytotoxic when retained on damaged mitochondria. If so, it should be sufficient for MITOL to be degraded like other Parkin substrates. We surmise that translocation of MITOL is not to eliminate MITOL, but rather to assist MITOL function on peroxisomes. During systematic deletion analyses of MITOL, we happened to find that a MITOL mutant lacking the eight most C-terminal amino acids (MITOLΔC8) caused peroxisome expansion after CCCP treatment in the presence of Parkin (Fig 12A). To quantify the effect of MITOLΔC8 on peroxisomal size and abundance, HeLa cells expressing HA-Parkin and wild-type or ΔC8 MITOL were treated with 15 μM CCCP for 3 h, and then the number of peroxisomes was counted as PMP70-positive dots per area (100 μm$^2$). The number of peroxisomes in cells expressing MITOLΔC8 was significantly reduced relative to cells expressing wild-type MITOL (Fig 12C). In addition, we compared the approximate size of peroxisomes (determined by the number of pixels occupied by one peroxisome) and found a drastic expansion of peroxisomes after CCCP treatment in cells expressing MITOLΔC8 (Fig 12D). To examine whether the E3 activity of MITOL is required for this expansion, we generated two mutants, one in which Cys65/Cys68 within the RING domain was changed to Ser (CS) and another in which the conserved Zn-binding His43 was replaced with Trp (H43W). Both of the E3-inactive MITOL mutants with the C-terminal eight amino acid deletion (CS/ΔC8 and H43W/ΔC8) localized to peroxisomes following CCCP treatment but did not trigger peroxisome expansion

(Fig 12B; higher background peroxisomal signals in the absence of CCCP treatment are derived from CS and H43W mutants, as described in Fig 3B). The C-terminal eight amino acid deletion might convert MITOL to a constitutive-active form via de-repression. Following ubiquitin immunostaining, a signal that overlapped with MITOLΔC8 was observed on enlarged peroxisomes as well as on Parkin-ubiquitylated damaged mitochondria (Fig 13A). This peroxisomal ubiquitylation signal was absent in cells expressing the E3-inactive MITOL mutant (Fig 13B). These results indicate that peroxisomes are highly ubiquitylated by translocated MITOLΔC8 following CCCP treatment, suggesting that MITOL potentially regulates both the abundance and size of peroxisomes.

## Discussion

PINK1 and Parkin direct mitochondria with reduced membrane potentials toward degradation. During this process, a large number of the outer mitochondrial membrane (OMM) proteins are ubiquitylated. Here, we show for the first time that MITOL migrates from damaged mitochondria to peroxisomes depending on Parkin-catalyzed ubiquitylation. We have elucidated in detail the molecular mechanism underlying the translocation of MITOL to peroxisomes, and found that (i) newly synthesized MITOL is integrated into the OMM via Tom70, (ii) mitophagy stimulation triggers Parkin-dependent ubiquitylation of MITOL K268 that is essential for translocation, and (iii) p97/VCP and Pex3/16 assist in the translocation of K268-ubiquitylated MITOL from damaged mitochondria to peroxisomes. Since Tom70 knockdown did not result in peroxisome localization of MITOL, it seems that mitochondrial localization is a prerequisite for MITOL translocation (Fig 5). Conversely, when peroxisomal translocation was blocked by Pex19 depletion (Fig 6), MITOL was retained on mitochondria. Therefore, the peroxisome is not necessary for either mitochondrial targeting or localization of MITOL.

Although atypical, there are examples of a limited number of OMM proteins that escape Parkin-mediated degradation on mitochondria, and translocate to the other organelle. FKBP38 and Bcl-2 translocate from damaged mitochondria to the ER during Parkin-mediated mitophagy [60]. In their study of the basis

---

**Figure 12.  MITOL lacking C-terminal 8 amino acids (MITOLΔC8) causes expansion of peroxisomes following CCCP treatment.**

A   HeLa cells expressing HA-Parkin and wild-type 3Flag-MITOL or 3Flag-MITOL lacking C-terminal 8 amino acids (ΔC8) were treated with 15 μM CCCP for 3 h, and then subjected to immunocytochemistry with anti-Flag and anti-PMP70 antibodies. Expanded peroxisomes were observed in MITOLΔC8-expressing cells. Higher magnification images of the boxed regions are shown in the small panel. Scale bars, 10 μm.

B   3Flag-MITOLΔC8 with the E3-inactive Cys65Ser/Cys68Ser (CS) and H43W mutations were transfected into HeLa cells stably expressing HA-Parkin, treated with 15 μM CCCP, and then subjected to immunocytochemistry with anti-Flag and anti-PMP70 antibodies. After 3 h of CCCP treatment, both the CS and H43W mutants localized on peroxisomes but expansion was not observed. Scale bars, 10 μm.

C   HeLa cells expressing HA-Parkin and MITOL wild-type or ΔC8 were treated with 15 μM CCCP for 3 h, and the number of peroxisomes was counted as PMP70-positive dots per 100 μm$^2$. In the box-plots, dots indicate individual data points, the center lines show the medians, box limits indicate the 25$^{th}$ and 75$^{th}$ percentiles as determined by the R software package, and whiskers extend 1.5 times the interquartile range from the 25$^{th}$ and 75$^{th}$ percentiles. Means and the number of samples are shown on the box and the *X*-axis, respectively. Statistical significance was calculated using a one-tailed Welch's *t*-test. The abundance of peroxisomes in cells expressing MITOLΔC8 was significantly decreased as compared with cells expressing wild-type MITOL.

D   MITOLΔC8 causes a drastic expansion of peroxisomes following CCCP treatment. HeLa cells expressing HA-Parkin and wild-type or ΔC8 MITOL were treated with 15 μM CCCP for 3 h, and the approximate size of peroxisomes was determined as the number of pixels occupied by one peroxisome. The PMP70-positive pixels per 100 μm$^2$ were divided by the number of peroxisomes in the same area. In the box-plots, dots indicate individual data points, the center lines show the medians, box limits indicate the 25$^{th}$ and 75$^{th}$ percentiles as determined by the R software package, and whiskers extend 1.5 times the interquartile range from the 25$^{th}$ and 75$^{th}$ percentiles. Means and the number of samples are shown on the box and the *X*-axis, respectively. Statistical significance was calculated using a one-tailed Welch's *t*-test.

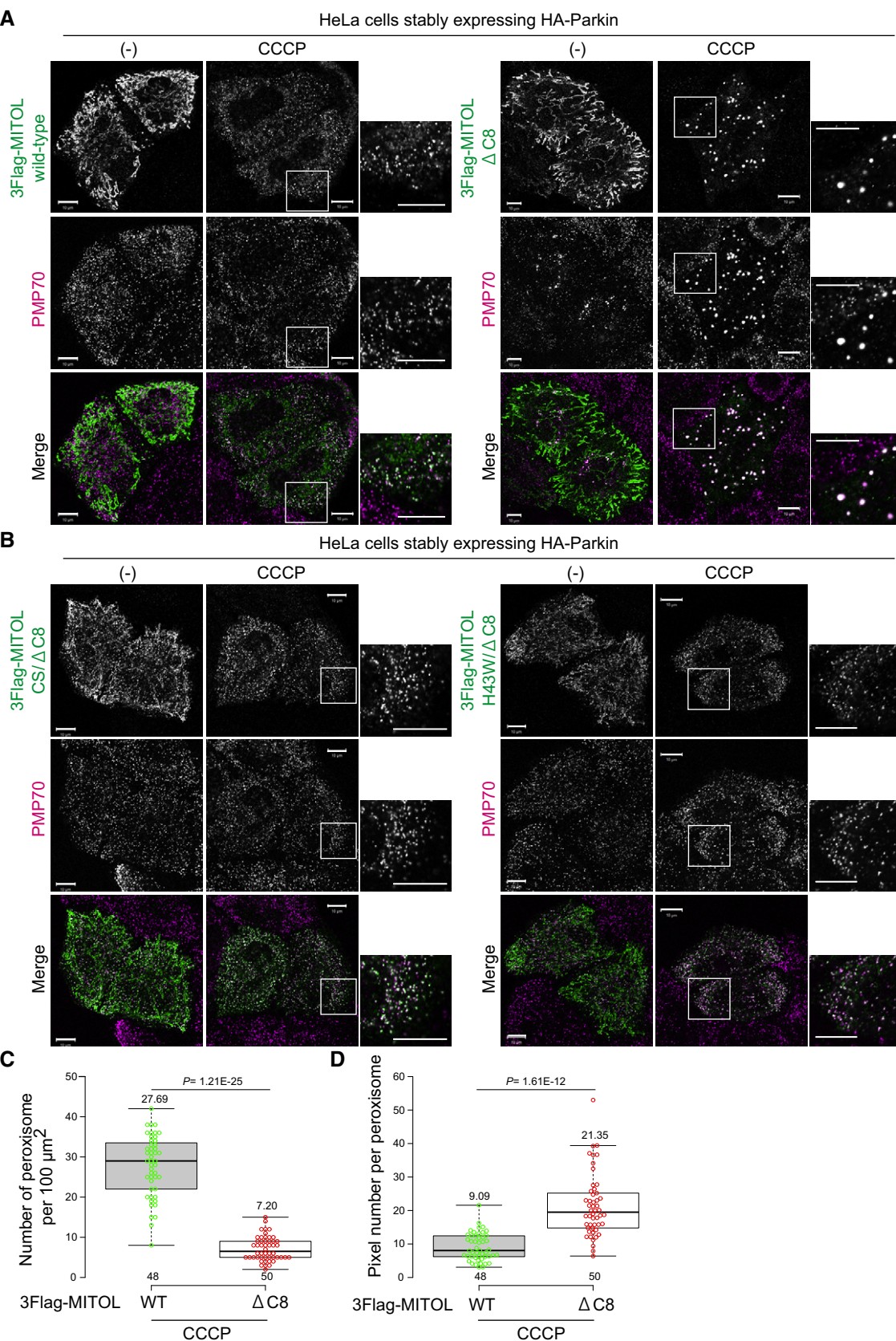

**Figure 12.**

underlying the mitochondria retention of Bcl-XL and Omp25 (neither targets to the ER despite topologies comparable with FKBP38 and Bcl-2) during mitophagy, Saita *et al* found that the number of basic

amino acids in the C-terminus determined the protein fates. Mutational analyses of the FKBP38, Bcl-2, and Bcl-XL revealed that one basic amino acid (K or R) in the C-terminal region directed

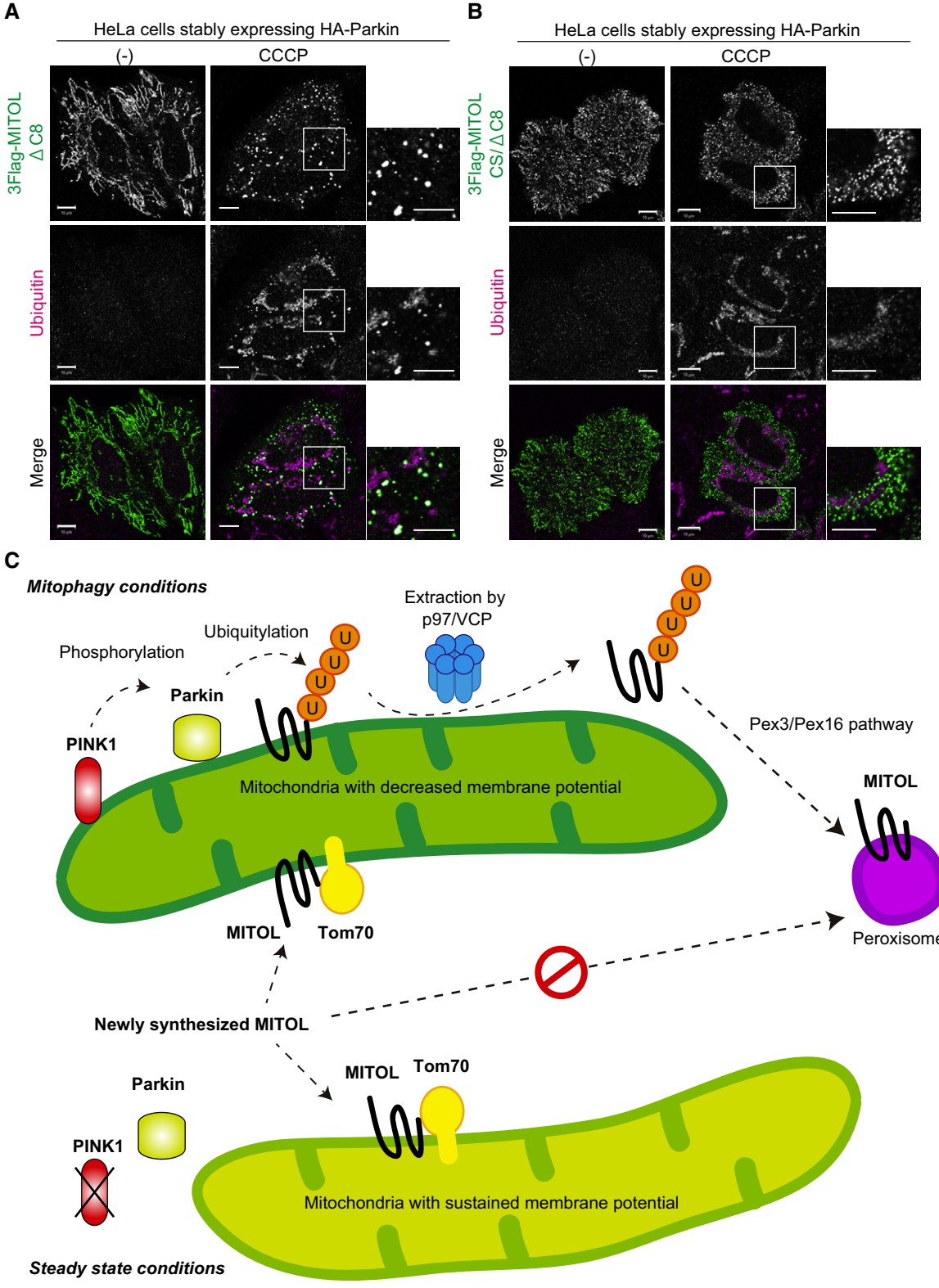

**Figure 13.**

**Figure 13. MITOLΔC8 ubiquitylates expanded peroxisomes following CCCP treatment.**

A, B HeLa cells expressing HA-Parkin and MITOLΔC8 (A) or MITOLΔC8 lacking E3 activity (B) were treated with 15 μM CCCP for 3 h, and then subjected to immunocytochemistry with anti-Flag and anti-ubiquitin antibodies. Expanded peroxisomes were ubiquitylated upon E3 activity of MITOL. Higher magnification images of the boxed regions are shown in the small panel. Scale bars, 10 μm.

C The schematic model of PINK1/Parkin-mediated MITOL redistribution. MITOL is targeted to mitochondria with sustained membrane potential via Tom70. In response to mitophagy stimuli, Parkin-catalyzed ubiquitylation of MITOL causes its extraction from damaged mitochondria in a p97/VCP-dependent manner, and then, MITOL translocates to peroxisomes via Pex3/Pex16 pathway. As ubiquitylation of MITOL is rarely observed in the absence of NMS-873, MITOL inserted into the peroxisome membrane is shown here in the non-ubiquitylated form.

translocation to the ER, whereas two or more basic amino acids in the C-terminal region resulted in mitochondrial retention and subsequent degradation in response to mitophagy.

In this report, we showed that the MITOL C-terminal residue K268 is essential for escape from damaged mitochondria (Fig 9). Since K268 is a basic amino acid in the MITOL C-terminus, the molecular mechanism underlying MITOL translocation could be similar to that of FKBP38 and Bcl-2 [60]. However, multiple points suggest that the mechanisms are completely different. First, FKBP38 and Bcl-2 are tail-anchored proteins, and it is unknown whether the C-terminal region, which determines translocation from mitochondria to the ER, faces the cytoplasmic or luminal side. The MITOL C-terminus, in contrast, is oriented toward the cytoplasmic side as evidenced by Parkin-mediated ubiquitylation of K268 (Fig 9). Second, although the E3 activity of Parkin is required for FKBP38 to translocate from depolarized mitochondria to the ER, it is unknown whether the C-terminal regions of FKBP38 undergo ubiquitylation similar to MITOL. Third, in the case of FKBP38 and Bcl-2, both K and R residues in the C-terminus can direct the proteins from the mitochondria to the ER, whereas in the case of MITOL, the K268R mutation hampered translocation from mitochondria to peroxisomes (Fig 9). For MITOL, the more important factor is whether the C-terminal K undergoes ubiquitylation rather than the presence of the basic residue itself.

Regarding mitochondria-to-peroxisome transport, mitochondria-derived vesicles (MDVs) have been reported. The OMM RING finger protein MAPL/MuL1/MULAN was incorporated into MDVs via a Vps35-dependent mechanism and transported to peroxisomes, but this process was independent of PINK1 and Parkin [61,62]. Moreover, only 10–20% of the total peroxisomes were MAPL positive, suggesting MAPL translocates to specific subpopulations of peroxisomes [63]. In contrast, MITOL-positive small dots predominantly coincided with all peroxisomes following mitophagy stimulation. MAPL-positive MDVs were generated from the curvature of the OMM; however, it is unknown whether the transit of MAPL-positive MDVs to peroxisomes requires VCP. Given that VCP is involved in protein extraction rather than vesicular formation in general, our results (Fig 8) suggest that not vesicular formation but selective MITOL extraction from the OMM is involved. The molecular mechanisms underlying MITOL redistribution from damaged mitochondria to peroxisomes might be completely different than those of MAPL.

Ubiquitin determines subcellular localization of various substrate proteins. For example, ubiquitylation accelerates a number of protein transport events including the following: from the plasma membrane to early endosomes in endocytosis, from the surface of late endosomes to the multivesicular body, from cytosol/plasma membrane to the virus particles during virus budding, from the ER to cytosol in ER-associated degradation, and from various organelles to autophagosomes in selective autophagy. Here we found that

ubiquitylation functions as a new signal for mitochondria-to-peroxisome transport. By what mechanism does MITOL translocate to peroxisomes in response to ubiquitylation? Various other ubiquitylated OMM proteins such as MitoNEET/CISD1, Fis1, Miro1/2, Tom20, and Tom70 do not translocate to peroxisomes during Parkin-mediated mitophagy (Fig 4), indicating that the peroxisomal targeting of MITOL is not solely attributable to ubiquitylation. We think the most reasonable hypothesis is that MITOL has both peroxisomal localization and mitochondrial localization sequences, but that the latter targeting signal is dominant. Dysfunction of the mitochondrial localization sequence by C-terminal ubiquitylation in response to Parkin-mediated mitophagy triggers the usually latent peroxisomal localization sequence and promotes MITOL translocation to the peroxisome. We have found that VCP is essential for MITOL localization to peroxisomes—this VCP involvement is a unique aspect of MITOL translocation that has not been reported for either FKBP38 or Bcl-2. Ubiquitylation plays an essential role in VCP-based recognition [64]. Taken together, when Parkin ubiquitylates MITOL K268, the ubiquitin assists in MITOL mitochondrial release via VCP-based recognition, and at the same time this ubiquitin suppresses the mitochondrial localization signal, which allows the peroxisome localization signal to manifest.

On the other hand, this translocation event raises new questions regarding its purpose. OMM proteins ubiquitylated by Parkin are recognized by the proteasome for prompt degradation, or they are recognized by autophagy adaptor/receptor proteins such as OPTN and NDP52 for degradation via the lysosome [6–9,65]. Thus, OMM proteins ubiquitylated by Parkin are essentially destined for degradation. If MITOL is unnecessary or cytotoxic on damaged mitochondria, it should be sufficient for MITOL to be targeted for degradation like other OMM proteins. However, MITOL was not degraded following Parkin-catalyzed ubiquitylation; instead translocated to peroxisomes. These results suggest that translocation of MITOL is not to eliminate MITOL, but rather to assist in retention of MITOL on peroxisomes. Although the functional relevance of ubiquitin-dependent MITOL translocation to peroxisomes has not been unveiled completely, we obtained an important clue to the significance. As shown in Fig 12, in the presence of Parkin, the MITOL mutant lacking the eight most C-terminal amino acids (MITOLΔC8) induced both peroxisome expansion and a reduction in peroxisomal abundance following CCCP treatment. Ubiquitin was accumulated on expanded peroxisomes in MITOLΔC8-expressing cells, whereas E3-inactive MITOLΔC8 mutants failed to lead ubiquitylation and expansion of peroxisomes (Figs 12 and 13). Mitochondria and peroxisomes share some metabolic pathways, and to maintain cellular homeostasis their distribution and abundance are highly coordinated [45,66–68]. As such, it is possible that MITOL plays a critical role in balancing the two organelles when mitochondria are damaged. More detailed explorations of this potential homeostatic mechanism warrant future study.

In total, this study demonstrates a novel function for Parkin-catalyzed ubiquitylation in the translocation of a mitochondrial protein to peroxisomes (Fig 13C) and provides new significance regarding ubiquitylation of damaged mitochondrial proteins.

## Materials and Methods

### Plasmids and antibodies

To construct plasmids for transient expression of MITOL-GFP, MITOL-HA, MITOL-3Flag, and 3Flag-MITOL, the *MITOL* coding sequence was amplified from a HeLa cDNA library and inserted into the HindIII/KpnI sites of pEGFP-N1 (Clontech), pcDNA (Invitrogen), and pcDNA3.1 (Invitrogen) to generate C-terminal GFP, HA, and 3Flag tags, respectively. For stable expression of MITOL, the *MITOL* coding sequence was inserted into the BamHI/EcoRI sites of the pMXs-Puro retroviral expression vector (CELL BIOLABS, INC.) harboring the 3Flag-tag sequence. The MITOL C65S/C68S (CS), H43W, K268A/R, K54A/R, or K40A/R mutations were introduced by primer-based PCR mutagenesis. The PBR-HA plasmid was kindly gifted by Dr. Endo (Kyoto Sangyo University). To prepare the PBR-3Flag plasmid, the DNA fragment encoding *PBR* was inserted into BamHI/EcoRI sites of the pcDNA3.1 vector containing the 3Flag sequence. For transient expression of PMP34-FusionRed, the *FusionRed* coding sequence (Evrogen) was inserted into the pcDNA vector to generate the pcDNA-FusionRed plasmid. Then, the *PMP34* coding sequence was inserted into the HindIII/KpnI sites of the pcDNA-FusionRed plasmid. pcDNAZeo/RnPEX3-HA2, pcDNAZeo/HA3-HsPEX16, and pcDNAZeo/HA2-HsPEX19 were generated as described previously [69,70,71]. The pCMV2/Flag-p97/VCP was described previously [72] and E305Q/E578Q (p97QQ) plasmid was gifted by Dr. Kakizuka (Kyoto University). The GFP-Parkin, Flag-Parkin, and Su9-GFP plasmids were generated as described previously [4,73]. For doxycycline-inducible expression, the *3FLAG-MITOL* coding sequence was inserted into the *Sal*I/*Bam*HI sites of the pTRE3G-IRES vector (MBL) to generate pTRE3G-3Flag-MITOL plasmid.

The following antibodies were used for immunoblotting: anti-Flag (Cat. # PM020, MBL or FLA-1, Cat. # M185-3L, MBL, 1:1,000), anti-Tom70 (a kind gift from Dr. Otera, Kyushu University, 1:1,000), anti-Tom20 (FL145, Cat. # sc-136211, RRID:AB_2207538, Santa Cruz, 1:500–1:1,000), anti-Tom40 (a kind gift from Dr. Oka, Rikkyo University, 1:1,500), anti-Sam50 (EPR8718, Cat. # ab133709, Abcam, 1:500), anti-AIF (E-1, Cat. # sc-13116, RRID:AB_626654, Santa Cruz, 1:1,000), anti-GFP (Cat. # ab6556, RRID:AB_305564, Abcam, 1:2,000), anti-p97/VCP (Cat. # ab11433, RRID:AB_298039, Abcam, 1:1,000), anti-Tubulin (Cat. # ab6160, RRID:AB_305328, Abcam, 1:1,000), anti-LDH (Cat. # ab2101, RRID:AB_302839, Abcam, 1:1,000), anti-PMP70 (Cat. # SAB4200181, RRID: AB_10639362, SIGMA, 1:1,000), and anti-Ubiquitin (P4D1, Cat. # sc-8017, RRID:AB_628423, Santa Cruz, 1:1,000), anti-mitofusin2 (Cat. # ab56889, RRID:AB_2142629, Abcam, 1:500), anti-MTCO2 antibody (12C4F12, Cat. # ab110258, Abcam, 1:500).

### The following antibodies were used for immunocytochemistry:

anti-HA (TANA2, MBL, 1:1,000), anti-Flag (Cat. # PM020, MBL or FLA-1, Cat. # M185-3L, MBL, 1:1,000), anti-Catalase (1A1, Cat.

# LF-MA0003, RRID:AB_1611839, AbFrontier, 1:200), anti-Pex14 (Cat. # ABC142, Millipore, 1:200), anti-PMP70 (Cat. # SAB4200181, RRID: AB_10639362, SIGMA, 1:200), anti-Sec61β (Cat. # 07-205, RRID: AB_11212145, Upstate, 1:3,000), anti-LAMP1 (H4A3, Cat. # sc-20011, RRID:AB_626853, Santa Cruz, 1:300), anti-Tom20 (FL-145, Cat. # sc-136211, RRID:AB_2207538, Santa Cruz, 1:1,000), anti-Hsp60 (Cat. # N-20, Santa Cruz, 1:250), anti-MitoNEET/CISD1 (Cat. # 16006-1-AP, RRID:AB_2080268, ProteinTech, 1:250), anti-Fis1 (a kind gift from Dr. Oka, Rikkyo University, 1:250), anti-Miro1 (Cat. # HPA016087, RRID: AB_1079813, SIGMA, 1:200), anti-Miro2 (Cat. # 11237-1-AP, RRID: AB_2179539, ProteinTech, 1:200), anti-Drp1 (8/DLP1, Cat. # 611113, BD Transduction Laboratories, 1:200), anti-mitofusin1 (3C9, Cat. # H00055669-M04, RRID:AB_581724, Abnova, 1:200), anti-Tom70 (a kind gift from Dr. Otera, Kyushu University, 1:200), and anti-Ubiquitin (P4D1, Cat. # sc-8017, RRID:AB_628423, Santa Cruz, 1:100).

### Cell culture and transfections

HeLa cells and SH-SY5Y cells were cultured at 37°C with 5% $CO_2$ in Dulbecco's modified Eagle's medium (DMEM; Gibco) containing 1× nonessential amino acids (Gibco), 1× sodium pyruvate (Gibco), 1× penicillin–streptomycin–glutamine (Gibco), and 10% fetal bovine serum. HCT116 cells were cultured in McCoy's 5A medium (Gibco) supplemented with 1× nonessential amino acids (Gibco), 10% fetal bovine serum, and 1× GlutaMAX (Gibco). *PINK1* knockout HeLa cells and *USP30* knockout HeLa cells were prepared as previously reported [41,55]. To generate HeLa cells stably expressing HA-Parkin or GFP-Parkin, HeLa cells transiently expressing mCAT1 (murine cationic amino acid transporter-1) were infected with recombinant retroviruses harboring HA-Parkin or GFP-Parkin. Recombinant retroviruses were produced using PLAT-E cells as described previously [12].

HeLa cells and HCT116 cells stably expressing 3Flag-MITOL and HA-Parkin were established by recombinant retrovirus infection. Vector particles were produced in HEK293T cells by co-transfection with Gag-Pol, VSV-G, and the retrovirus plasmids using Lipofectamine LTX Reagent (Thermo Fisher Scientific) [40]. After 12 h of transfection, the culture medium was replaced with fresh medium. The cells were further cultivated for 24 h. Collected viral supernatants were then used to infect HeLa cells or HCT116 cells using 8 μg/ml polybrene (SIGMA). Plasmid transfections were performed using FuGENE6 (Promega) or polyethylenimine (Polyscience Inc.) according to the manufacturer's instructions.

To depolarize mitochondria, cells were treated with 15 μM carbonyl cyanide *m*-chlorophenylhydrazone (CCCP; Wako) or 10 μM valinomycin (SIGMA) for 3 h unless otherwise specified. To inhibit proteasome activity, 10 μM MG132 (SIGMA) was used, and to inhibit p97/VCP activity, 10 μM NMS-873 (SIGMA) was used. To prevent SH-SY5Y cell death in response to the valinomycin treatment, an apoptosis inhibitor, Z-VAD-FMK (Peptide Institute), was included in the incubation.

### Dox-inducible expression system

The pTRE3G-3Flag-MITOL and pCMV-Tet3G plasmids were co-transfected into HeLa cells stably expressing HA-Parkin. After 24 h, doxycycline (100 ng/ml; Clontech) was added to cells for 3 h. The cells were then repeatedly washed with fresh media, and treated with 15 μM CCCP for 3 h to induce mitochondrial depolarization.

## Subcellular fractionation

HCT116 cells stably expressing HA-Parkin and 3Flag-MITOL, or MITOL-3Flag knock-in HCT116 cells stably expressing HA-Parkin, were harvested in homogenization buffer [20 mM HEPES–NaOH, pH 7.5, 0.25 M sucrose, 1 mM DTT, 2 mM EGTA, and protease inhibitor cocktail EDTA-free (Roche)]. Cells were homogenized using four strokes of a Potter-Elvehjem homogenizer and then centrifuged at 800 $g$ for 5 min to obtain a post-nuclear supernatant (PNS). The PNS fraction was further separated into supernatant and pellet fractions by centrifugation at 3,000 $g$ for 10 min. The 3,000 $g$ supernatant fraction was then centrifuged at 100,000 $g$ for 30 min to yield a 100,000 $g$ pellet fraction. Each fraction was suspended in the same volume as the PNS fraction, and identical sample volumes were loaded on gels for immunoblotting. For separation of cytosolic and mitochondria-enriched fractions in si*Tom70*-treated cells, the cells were harvested in a modified homogenization buffer [10 mM HEPES–NaOH, pH 7.5, 0.25 M sucrose, protease inhibitor cocktail, EDTA-free (Roche), and 10 μM NMS-873] and homogenized by passing through a 27-gauge needle seven times. After centrifugation at 1,000 $g$ for 10 min, a PNS was collected. The PNS was further centrifuged at 20,400 $g$ for 10 min to obtain cytosolic and mitochondria-enriched fractions. Each band was quantified using ImageQuant TL (GE Healthcare).

## RNA interference

Non-targeting control siRNA and siRNA oligos for p97/VCP were described previously. The siRNA oligonucleotide sequences are as follows: control, 5′-CGUUAAUCGCGUAUAAUACGCGUAT-3′; p97/VCP, 5′-CGGGAGAGGCGCGCGCCAUUTT-3′ [74,75]. For siRNA knockdown of Pex3, Pex16, and Pex19, Stealth siRNA (HSS112408, HSS145149, and HSS108913, respectively) were purchased from Invitrogen. The siRNA oligonucleotide sequences are as follows: Pex3, 5′-GGGAGGAUCUGAAGAUAAUAAGUUU-3′; Pex16, 5′-GGAUCCUACGGAAGGAGCUUCGGAA-3′; Pex19, 5′-AGAAUGGUUGCAGAGUCAUCGGGAA-3′. siRNAs were transfected into cells using Lipofectamine RNAiMAX (Invitrogen) according to the manufacturer's instruction. After 24 h of transfection, the medium was replaced with fresh medium and the cells were grown for another 24 h. For co-transfection of siRNA and plasmid DNA, Lipofectamine 2000 (Invitrogen) was used according to the manufacturer's instruction. After 6 h of transfection, the medium was replaced with fresh medium and the cells were grown for another 42 h. For knockdown of *Tom40* and *Sam50*, siGENOME Human siRNA SMARTpools were purchased from Thermo Scientific. The target sequences are as follows: Tom40, 5′-GCUGGGAAAUACACAUUGA-3′, 5′-GAUAGCAA CUGGAUCGUGG-3′, 5′-UCUCAACGCUCAGGUCAUU-3′, 5′-GCAAGA ACAAGUUUCAGUG-3′; Sam50, 5′-CAAAUGGGUUAGACGUUAC-3′, 5′-CGAAGGAGACUACCUAGGU-3′, 5′-UAACUGAAUUGAGGAGAU U-3′, 5′-GAACAAGCAACUCAUAUUU-3′. For knockdown of *Tom20*, and *Tom70*, FlexiTube siRNAs (SI00301959 and SI00301973, respectively) were purchased from QIAGEN. The target sequences were as follows: Tom20/TOMM20, 5′-AAAGTTACCTGACCTTAAAGA-3′; Tom70/TOMM70A, 5′-AGACAAATAAGAAGGAATGTT-3′. siRNAs were transfected to HeLa cells twice within a 24-h interval. After 72 h of initial transfection, cells were transfected with Su9-GFP and

3Flag-MITOL plasmids using FuGENE6 (Promega). After 16 h of plasmid transfection, cells were fixed with 4% paraformaldehyde and subjected to immunostaining and subcellular fractionation.

## CRISPR/Cas9-based generation of a *PEX19* knockout cell line

*PEX19*$^{-/-}$ HCT116 cells were established by CRISPR/Cas9-based genome editing with an antibiotic-selection strategy. The gRNA target sequence for the region of exon 1 in the *PEX19* gene (5′-GT GTCGGGGCCGAAGCGGACAGG-3′) was selected using an online CRISPR design tool (CRISPRdirect). Two DNA oligonucleotides, hPex19-ex1-1-CRISPR-F (5′-TGTATGAGACCAC GTGTCGGGGCCGA AGCGGAC-3′) and hPex19-ex1-1-CRISPR-R (5′-AAAC GTC CGC TTC GGC CCC GAC AC GTGGTCTCA-3′) were annealed and introduced into a linearized pEF1-hspCas9-H1-gRNA vector (Cas9 SmartNuclease™; System Biosciences, LLC) according to the manufacturer's protocol. The DNA fragment was verified by DNA sequencing. Neomycin- and hygromycin-resistant marker plasmids were constructed as follows. The neomycin-resistant gene (*NeoR*), including loxP sites along with the appropriate promoter and terminator, was PCR-amplified from pMK286 [59] using BamHI-NeoR-F (5′-ggc cGG ATC Cct aat taa cta gAT AAC TTC GTA TAA TGT ATG CTA TAC GAA GTT ATc tga ggc gga aag aa-3′) and NeoR-BamHI-R (5′-GGC Cgg atc cAT AAC TTC GTA TAG CAT ACA TTA TAC GAA GTT Ata acg acc caa cac cg-3′) primers. The hygromycin-resistant gene (*HygroR*), including loxP sites with appropriate promoter and terminator, was PCR-amplified from pMK287 [59] using BamHI-HygroR-F (5′-ggc cGG ATC Cct aat taa cta gAT AAC TTC GTA TAA T-3′) and HygroR-BamHI-R (5′-ggc cGG ATC Cta gtg aac ctc ttc g-3′) primers. The amplified *NeoR* and *HygroR* fragments were digested with BamHI and ligated into the corresponding site of pBluescript II SK(-) to make pBSK/NeoR and pBSK/HygroR, respectively. The 247 bp of the 5′ and 3′ homology arms of the *PEX19* exon 1 region, which lacks the gRNA target sequence but has a BamHI site in the middle (total 500 bp), was synthesized and cloned into a pUC57-Amp vector (GENEWIZ) to make pUC57-Amp/*PEX19*-ex1-donor. *NeoR* and *HygroR* resistant markers extracted by BamHI digestion from pBSK/NeoR and pBSK/HygroR were then inserted into the BamHI site of pUC-Amp/*PEX19*-ex1-donor. The resultant NeoR and HygroR donor plasmids containing the *PEX19* exon 1 homology arm were transfected into HCT116 cells with the gRNA plasmid using FuGENE6 (Promega). Cells were grown in McCoy's 5A media containing 700 μg/ml G418 (G8168; SIGMA) and 100 μg/ml hygromycin B (10687-10, Invitrogen). *PEX19*$^{-/-}$ single clones were screened by PCR using genomic DNA to verify neomycin-resistant and hygromycin-resistant gene insertion. To verify insertion of the resistant markers, the following primers were used; hPex19-ex1-1-check-F2 (5′-CGCCAGGTAATTTGGGAAGTTG-3′), hPex19-ex1-1-check-R2 (5′-CCTGCCCGTCCCTAATATCTC-3′). The absence of peroxisomes in *PEX19*$^{-/-}$ cells was further confirmed by immunostaining with anti-Catalase and anti-Pex14 antibodies.

## Generation of a *MITOL-3Flag* knock-in cell line

*MITOL-3Flag* knock-in HCT116 cells were established using CRISPR/Cas9-based genome editing with an antibiotic-selection strategy. The gRNA target sequence for the region of exon 6 in the *MITOL* gene (5′-AGAAATATAATAAAGCACTTAGG-3′) was selected

using an online CRISPR design tool (CRISPRdirect). Two DNA oligonucleotides, hMITOL-ex6-3-CRISPR-KI-F (5′-TGTATGAGACCA C AGAAATATAATAAAGCACTT-3′) and hMITOL-ex6-3-CRISPR-KI-R (5′-AAAC AAG TGC TTT ATT ATA TTT CT GTGGTCTCA-3′) were annealed and introduced into a linearized pEF1-hspCas9-H1-gRNA vector (Cas9 SmartNuclease™; System Biosciences, LLC) according to the manufacturer's protocol. The DNA fragment was verified by DNA sequencing. Neomycin- and hygromycin-resistant marker plasmids were constructed as follows. The neomycin-resistant gene (*NeoR*), including 3Flag was amplified by PCR from pMK283 [59] using pMK283,284_BamHI(F) (5′-AAA gga tcc ggt gca ggc gcc AAG GAA-3′) and pMK277,283_BamHI(R) (5′-CCC GGA TCC AAC GAC CCA ACA CCG TGC-3′) primers. The hygromycin-resistant gene (*HygroR*), including 3Flag was amplified by PCR from pMK284 [59] using pMK283,284_BamHI(F) (5′-AAA gga tcc ggt gca ggc gcc AAG GAA-3′) and pMK278,284_BamHI(R) (5′-CCC GGA TCC TAG TGA ACC TCT TCG AGG-3′) primers. The 247 bp of the 5′ and 3′ homology arms of the *MITOL* exon 6 region, which lacks the gRNA target sequence but has a BamHI site in the middle (total 500 bp), was synthesized and cloned into a pUC57-Amp vector (GENEWIZ) to make pUC57-Amp/*MITOL*-ex6-donor. The PCR-amplified *3Flag-NeoR* and *3Flag-HygroR* resistant markers were extracted by BamHI digestion and then inserted into the BamHI site of pUC-Amp/*MITOL*-KI-ex6-donor. The resultant 3Flag-NeoR and 3Flag-HygroR donor plasmids containing the *MITOL* exon 6 homology arm were transfected into HCT116 cells with the gRNA plasmid using FuGENE6 (Promega). Cells were grown in McCoy's 5A media containing 700 μg/ml G418 (G8168, SIGMA) and 100 μg/ml hygromycin B (10687-10, Invitrogen). After 48 h, cells were expanded with neomycin and hygromycin-containing McCoy's media. MITOL-3Flag knock-in HCT116 cell single clones were screened by immunoblotting using an anti-Flag antibody.

## Immunoblotting

HeLa cells were solubilized with TNE-N$^+$ buffer [20 mM Tris–HCl, pH 8.0, 150 mM NaCl, 1 mM EDTA, 1% NP-40, and protease inhibitor cocktail complete EDTA-free (Roche) in the presence of 1 mM N-ethylmaleimide (NEM)]. After removing insoluble debris by centrifugation, the supernatant was collected to obtain total cell lysates. The total protein concentration of the lysates was determined using a BCA Protein Assay Kit or a 660 nm Protein Assay Reagent (Pierce). SDS–PAGE sample buffer was added to lysates, and samples were boiled at 98°C for 5 min. Proteins were separated on 4–12% Bis-Tris SDS–PAGE gels (NuPAGE; Invitrogen) in MOPS buffer. Proteins were transferred to PVDF membranes (Merck), blocked with 5% skim milk/TBST, and incubated with primary antibodies. Membranes were then incubated with horseradish peroxidase-conjugated goat anti-mouse, goat anti-rabbit, and donkey anti-rat antibodies (Cat. # 315-035-048, #111-035-144, and #712-035-153, Jackson ImmunoResearch Inc.) as secondary antibodies. Images were obtained using an ImageQuant LAS 4000 (GE Healthcare).

## Immunoprecipitation

HeLa cells grown in a 10-cm dish were solubilized with TNE-N$^+$ buffer. After centrifugation, the supernatant was incubated with

anti-DDDDK-tag mAb Magnetic beads (M185-11; MBL) for 3 h at 4°C. The agarose was washed three times with TNE-N$^+$ buffer, and proteins were extracted by adding SDS–PAGE sample buffer.

## Immunocytochemistry

HeLa cells were fixed with 4% paraformaldehyde (Wako), permeabilized with 0.15% or 0.5% Triton X-100, and incubated with primary antibodies followed by 1:2,000 secondary antibodies [Alexa Fluor 488-, 568-, or 647-conjugated goat or donkey anti-mouse, anti-rabbit, or anti-goat IgG antibody (Invitrogen)]. Microscopy images were acquired at room temperature on a laser-scanning microscope with Plan Apochromat 63×/1.4 Oil (LSM710 or LSM780; Carl Zeiss). Image sizes were adjusted with Photoshop software (Adobe). For quantitative co-localization and statistical analysis, the co-localization of MITOL with other proteins was calculated using Zen software (Carl Zeiss) as a Pearson correlation coefficient. The number of peroxisomes was counted as PMP70-positive dots per area (100 μm$^2$) in randomly selected cells. After the images were converted into threshold images, Zen software (Carl Zeiss) was used to calculate the PMP70-positive pixels per 100 μm$^2$, which was then divided by the number of peroxisomes in the same area to yield an approximate size for peroxisomes. In the associated box-plots of the data, the dots indicate individual Pearson correlation coefficient data points, the center lines indicate the medians, the box limits indicate the 25th and 75th percentiles as determined by the R software package, and whiskers extend 1.5 times the interquartile range from the 25th and 75th percentiles. Outliers are represented by dots outside of the whiskers, mean values are shown on the boxes, and the number of samples is indicated above the X-axis. Statistical significance was calculated using a one-tailed Welch's *t*-test.

## LC-MS/MS-based identification of MITOL ubiquitylation sites

After valinomycin treatment for 3 h, *PEX19*$^{-/-}$ HCT116 cells stably expressing 3Flag-MITOL and HA-Parkin grown in a 10-cm dish were treated with 0.1% paraformaldehyde at room temperature for 10 min. Then, cells were treated with 300 mM glycine at room temperature for 3 min. After cells were washed with PBS three times, the collected cell suspensions were centrifuged at 2,000 *g* for 5 min. RIPA buffer [50 mM Tris–HCl, 150 mM NaCl, 0.1% (w/v) SDS, 0.5% (w/v) sodium deoxycholate, and 1% (v/v) Triton X-100] supplemented with protease inhibitor cocktail complete EDTA-free (Roche) was added to the pellet. After pipetting, cells were lysed on ice for 10 min. After centrifugation at 14,000 *g* for 10 min, the collected supernatants were incubated with anti-DDDDK-tag mAb Magnetic beads (Cat. # M185-11, MBL) for 1.5 h at 4°C with rotation. The beads were washed three times with RIPA buffer and twice with 50 mM ammonium bicarbonate. Proteins on the beads were digested by adding 200 ng trypsin (MS grade, Pierce) for 16 h at 37°C. The digests were reduced, alkylated, acidified and desalted using GL-Tip SDB (GL Sciences). The eluates were evaporated in a SpeedVac concentrator and dissolved in 0.1% trifluoroacetic acid.

LC-MS/MS analysis of the resultant peptides was performed on an EASY-nLC 1200 UHPLC connected to a Q Exactive Plus mass spectrometer equipped with a nanoelectrospray ion source (Thermo Fisher Scientific). The peptides were separated on a 75 μm inner diameter × 150 mm C18 reversed-phase column (Nikkyo Technos)

with a linear 4–28% acetonitrile (ACN) gradient for 0–100 min followed by an increase to 80% ACN for 10 min. The mass spectrometer was operated in a data-dependent acquisition mode with a top 10 MS/MS method. MS1 spectra were measured with a resolution of 70,000, an automatic gain control (AGC) target of $1 \times 10^6$ and a mass range from 350 to 1,500 $m/z$. HCD MS/MS spectra were acquired at a resolution of 17,500, an AGC target of $5 \times 10^4$, an isolation window of 2.0 $m/z$, a maximum injection time of 60 ms and a normalized collision energy of 27. Dynamic exclusion was set to 10 s. Raw data were directly analyzed against the SwissProt database restricted to *H. sapiens* using Proteome Discoverer version 2.2 (Thermo Fisher Scientific) with Mascot search engine version 2.5 (Matrix Science) for identification and label-free precursor ion quantification. The search parameters were as follows: (i) trypsin as an enzyme with up to two missed cleavages; (ii) precursor mass tolerance of 10 ppm; (iii) fragment mass tolerance of 0.02 Da; (iv) carbamidomethylation of cysteine as a fixed modification; and (v) acetylation of the protein N-terminus, oxidation of methionine and di-glycine modification of lysine as variable modifications. Peptides were filtered at a false-discovery rate of 1% using the percolator node. Normalization was performed such that the total sum of abundance values for each sample over all peptides was the same.

## FACS-based mitophagy assay

HCT116 cells stably expressing YFP-Parkin, mt-mKeima, and 3Flag-MITOL were grown in a 6-well plate and treated with 10 µM oligomycin (Cat. # 495455; Calbiochem) and 4 µM antimycin A (Cat. # A8674; SIGMA) for 0, 3, or 6 h. Cells were then resuspended in sorting buffer (phosphate buffer with 2.5% FBS). Analysis was performed using FACSDiva software on a BD LSRFortessa X-20 cell sorter. mKeima was measured using dual-excitation ratiometric pH measurements at 405 (pH 7) and 561 (pH 4) nm lasers with 610/20 nm emission filters. For each sample, 10,000 cells of YFP/mt-Keima double positive cells were collected.

**Expanded View** for this article is available online.

## Acknowledgements

We thank Dr. Otera (Kyushu University) for the anti-Tom70, Dr. Oka (Rikkyo University) for the anti-Tom40 and anti-Fis1, Dr. Endo (Kyoto Sangyo University) for PBR-HA plasmid, and Dr. Kakizuka (Kyoto University) for p97/VCP plasmid. *ATG5* KO HeLa cells, pBMNz(B)/YFP-Parkin plasmid, and pRetroQ/mt-Keima plasmid were kindly gifted by Dr. Chunxin Wang and Dr. Richard Youle (NIH). This work was supported by JSPS KAKENHI Grant Number JP18H02443, the Chieko Iwanaga Fund for Parkinson's Disease Research, and Joint Usage and Joint Research Programs, the Institute of Advanced Medical Sciences, Tokushima University (to N.M.); by JSPS KAKENHI Grant Number JP26000014 (to K.T.); by JSPS KAKENHI Grant Numbers JP15K19037 and JP18K14708 (to F.K.); by MEXT/JSPS KAKENHI Grant Numbers JP18H05500, JP16K18545, and JP18K06237 (to K.Y.); by JSPS KAKENHI Grant Number JP17K08635 (to H.K.); by the Takeda Science Foundation (to K.T. and N.M.); by the Ohsumi Frontier Science Foundation (to Y.K.); and by MEXT/JSPS KAKENHI Grant Numbers JP26116007, JP15K21743, and JP17H03675 (to Y.F.). We are grateful to Dr. Okatsu (The University of Tokyo), Dr. Okumoto (Kyushu University), and Dr. Tanegashima (Tokyo Metropolitan Institute of Medical Science) for valuable comments and technical supports.

## Author contributions

Conceptualization: FK, KY, NM, KT; methodology: FK, KY, HK; investigation: FK, KY, MK, HK; writing—original draft: FK; Review and editing: FK, KY, HK, YK, YF, NM, KT; funding acquisition: FK, KY, HK, YK, YF, NM, KT; and supervision: FK, KY, NM, KT.

## Conflict of interest

The authors declare that they have no conflict of interest.

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
