## [Review Process File · EMBO Reports]

Parkin-mediated ubiquitylation redistributes MITOL/March5 from mitochondria to peroxisomes

Fumika Koyano, Koji Yamano, Hidetaka Kosako, Yoko Kimura, Mayumi Kimura, Yukio Fujiki, Keiji Tanaka, and Noriyuki Matsuda

Review timeline:

Submission date:	15th Jan 2019
Editorial Decision:	30th Jan 2019
Revision received:	28th Jun 2019
Editorial Decision:	30th Jul 2019
Revision received:	30th Aug 2019
Accepted:	11th Sep 2019

Editor: Martina Rembold / Deniz Senyilmaz Tiebe

Transaction Report:

1st Editorial Decision

30th Jan 2019

Thank you for the submission of your research manuscript to our journal. We have now received the full set of referee reports that is copied below.

After reading the referee reports it becomes clear that the findings are potentially interesting but that a lot of work will be required to substantiate them. It will be necessary to quantify the data, to support the imaging-based data with subcellular fractionation, to provide data on endogenous proteins (Parkin, MITOL), to discriminate the effect of mitochondrial depolarization clearly from mitophagy and to provide missing controls. In addition, further data on the biochemistry of peroxisomal MITOL should be provided. Moreover, we note that the manuscript does not provide any data on the functional relevance of peroxisomal MITOL. We think that some evidence for the physiological relevance of MITOL translocation is required for publication in EMBO reports. Referee 3 suggested assessing the effect on mitophagy and on peroxisome membrane permeability.

Given the potential interest of your findings I would like to give you the possibility to revise your manuscript for EMBO reports. Please address all referee concerns as outlined above and in their reports and please also add some data on the functional relevance of MITOL translocation. Please address all referee concerns in a complete point-by-point response. Acceptance of the manuscript will depend on a positive outcome of a second round of review. It is EMBO reports policy to allow a single round of revision only and acceptance or rejection of the manuscript will therefore depend on the completeness of your responses included in the next, final version of the manuscript.

I realize that quite some work will be required to substantiate and extend the current data set, in particular the addition of functional relevance might be demanding. I have therefore taken the liberty to discuss your study and the referee reports with my colleague Andrea Leibfried, executive editor of our new open access sister journal Life Science Alliance (<http://www.life-science-alliance.org/>). Life Science Alliance is launched as a partnership between EMBO Press, Rockefeller Press, and Cold Spring Harbor Laboratory Press, and publishes work that is of high value to the respective communities across all areas in the life sciences. I am happy to say that Andrea would like to

publish your work pending less demanding revision. Andrea would expect a point-by-point response to all concerns raised and accordingly changes to the manuscript text. The requests for adding quantifications/statistics, providing improved imaging data, analyzing MITOL localization after CHX and CCCP treatment, adding missing controls, and confirming MITOL localization via fractionation analysis should get addressed. Adding an analysis at endogenous protein levels, analyzing MITOL topology at peroxisomes and its ubiquitylation status as well as addressing point 7 of rev#1 is NOT NEEDED for publication in Life Science Alliance.

Should you decide to embark on a revision for EMBO reports, then please submit the revised manuscript within three months of a request for revision. But please do not hesitate to contact us if a 3-months time frame is not sufficient for the revisions so that we can discuss the revisions further.

REFEREE REPORTS:

Referee #1:

In this manuscript Koyano and colleagues reported that MITOL/March5, a mitochondrial ubiquitin ligase, translocates from mitochondria to peroxisomes after mitophagy stimulation with membrane depolarizing agents. Using confocal microscopy and immunofluorescence, authors observed that MITOL specifically relocates to peroxisomes in a Parkin-dependent manner. They also found that this new MITOL translocation is mediated by the ATPase p97/VCP and peroxins such as Pex3. Finally, by mass spectrometry analysis, they found that ubiquitination of K268 by Parkin is required for MITOL translocation. Overall, this is an interesting manuscript and highlights not only the cross talk between mitochondria and peroxisomes, but also that Parkin may have functions outside of simply marking mitochondria for mitophagy. The whys and wherefores of this translocation are unknown and as such this work is somewhat phenomenological; however, my feeling is that this should not preclude publication as there is a lot of interest in this area. I also have other concerns as detailed below.

Major points:

1. The vast majority of conclusions are based on fluorescence microscopy colocalization interpretations, but quantitative data are completely absent. To support their conclusions and to show experimental variation, authors should perform quantitative colocalization and statistical analyses.
2. Related to the above point, MITOL may translocate to peroxisomes, but is this every one? What is the proportion of MITOL-positive peroxisomes after mitophagy induction and could recruitment be specific to a subpopulation of peroxisomes?
3. I appreciate that the vast majority of the field overexpresses Parkin to visualise mitophagy, despite this being somewhat artefactual it may not even be required for MITOL translocation. There are many cell lines that express easily detectable levels of (and activatable) endogenous Parkin (such as SH-SY5Y) and the authors should determine if MITOL translocation also occurs in an endogenous Parkin-dependent manner in these.
4. Figure 2, authors should analyse MITOL localisation after CHX and CCCP treatment. Is MITOL still localized in peroxisomes? This will support the notion that pre-existing MITOL moved to peroxisomes following mitochondrial depolarisation.
5. In Figure 2B, a positive control is missing (e.g. mitochondrial protein known to be degraded by the proteasome and mitophagy - such as mitofusin1/2). Also, loading controls are missing in panel B as well as C.
6. At multiple instances, the authors refer to the translocation as mitophagy-induced, when actually it is depolarisation-induced. While mitophagy and depolarisation are linked they are not the same and there is no mitophagy monitoring in the manuscript to distinguish between the two. Does actual mitophagy influence, or be influenced by, translocation? For example, is translocation of MITOL required for mitophagy progress i.e. can mitophagy still occur with ubiquitination-deficient MITOL?
7. The requirement of peroxisomes for mitochondrial MITOL removal is very interesting and suggests peroxisomes are involved in the translocation process (and not just the end destination). Can the authors monitor the translocation by live cell microscopy as this might give more of an insight, for example mitochondrial-peroxisome contact sites may be critical for this translocation?
8. In the Discussion, the authors should at least discuss/speculate a small bit about the biological function of mitochondria-to-peroxisome MITOL translocation after depolarisation.

Minor points:

9. For better visualisation of colocalization between proteins, some higher magnification views would help the reader.
10. Figure 6, results from western blot analysis don't match with immunofluorescence images. In Panel A, in Tom70-silenced cells, MITOL level seems increased compared to control. Yet in Panel B, MITOL level seems decreased in Tom70-silenced cells. Authors should clarify this point. In addition, in Panel C it is not clear how the authors quantify the % of cells with the indicated protein in cytosol. Indeed Su9-GFP seems to be enriched in the nucleus in Tom20- and Tom40- silenced cells. Some explanation would help.
11. Some experiments are done with CCCP and some with valinomycin - is there any specific reason for a lack of consistency? Are the authors sure that they both have the same effect?

Referee #2:

In this study, Koyano and colleagues propose that the mitochondrial E3 ubiquitin ligase MITOL/March5 translocates to peroxisomes during mitophagy. They further suggest that PINK1, peroxins and E3 activity of Parkin, but not MITOL E3 activity, are essential for VCP-dependent extraction of MITOL from mitochondria. Overall this is an interesting study, which could highlight a new important role for ubiquitin in the redistribution of MITOL (and possibly other proteins) from depolarised mitochondria to peroxisomes.

However, I have two major concerns, which in my opinion, preclude this study from being published in EMBO Reports (at least in the current form):

- There is quantification/statistical analysis throughout the manuscript (except for Figure 6C). A quantification should be included, at the very least in ALL the imaging experiments (especially as a vast majority of the conclusions are based on immunofluorescence). In the current version, it is impossible to assess the robustness of the findings.
- The study is largely based on over-expression. They authors have generated a MITOL-3Flag KI cell line. When possible, why not performing the experiments in this cell line?

Specific comments:

- In Figure 1, the authors should perform subcellular fractionations to confirm MITOL peroxisomal localisation following CCCP or valinomycin treatment. Moreover, the authors should perform a time course of CCCP.
- In Figure 2 B and C, there is no loading control. In Figure 2B, the authors suggest that MITOL is not degraded via mitophagy, as the MITOL levels don't change drastically after CCCP treatment. In order to draw such conclusion, the authors should extend the CCCP treatment (currently 3hrs only), and blot for positive mitophagy markers. Moreover, these experiments should be performed not only in whole cell lysates, but also in subcellular fractions.
- In Figure 5B, it seems that a consistent fraction of MITOL is still located at the mitochondria after CCCP treatment. The authors should provide better resolution pictures, and of course quantify the proportion of MITOL at the mitochondria/peroxisome following DMSO/CCCP or valinomycin treatment.
- In Figure 6C, have stats been performed? In Figure 6A, the authors should perform sub-cellular fractionations to ascertain that MITOL is in the cytosolic fraction of the TOM70 siRNA cells.
- In Figure 8A, to evaluate the KD, the authors over-express the different Pex and then KD. If not antibody is available, the authors should evaluate the KD of endogenous PEX by qPCR. The authors conclude that "the most pronounced inhibitory effects on MITOL translocation were observed following PEX3 KD" and that "the role of Pex3 is particularly significant". Again, how can they make such statements with no quantification?
- In Figure 9, have the authors considered using patient cells from VCP mutations carriers?
- In Figure 10, the authors claim that MITOL ubiquitination is mediated by Parkin, but have they performed the same experiments in WT Hela cells?

Referee #3:

This is an intriguing paper by Koyano et al. describing the relocation of a mitochondrial outer membrane E3 ligase, called MITOL/March5, from the outer membrane of damaged mitochondria to peroxisomes via a process that depends on the ubiquitylation of MITOL at K268, the activity of the mitochondrial E3 ligase Parkin but not that of MITOL itself, the peroxins Pex3, Pex16 and Pex19 on peroxisomes and the AAA-ATPase p97/VCP, which is believed to extract MITOL from the mitochondrial outer membrane.

Although the authors describe well several steps of this relocation process, the principal missing component is the physiological relevance and the generality of this process, which raises many important questions regarding whether this represents a significant advance or an oddity. Additionally, the mechanism of extraction of a protein from mitochondria and reinsertion into peroxisomes raises many key mechanistic questions that are unanswered. Consequently, this paper is too premature at present in terms of significance and mechanisms for publication in EMBO Reports.

I enclose some comments to help the authors improve their manuscript.

What is the physiological role of MITOL and how is that affected by its relocation to peroxisomes? For example, does it affect peroxisome membrane permeability (Hosoi et al., 2017; PMC5350511) or mitophagy?

When MITOL is relocated to peroxisomes, this should also be shown biochemically. Also, is MITOL an integral membrane protein in peroxisomes and does its topology resemble that found in mitochondria? What is its ubiquitylation status?

Is this a new pathway of peroxisomal membrane protein (PMP) targeting to peroxisomes, in which case it is very poorly characterized? Does MITOL have a membrane peroxisomal targeting signal (mPTS)? It appears not to because in the absence of mitochondrial targeting via Tom70, MITOL is cytosolic and not peroxisomal. What is the role of ubiquitylation in the activation of the mPTS?

Does MITOL need to be deubiquitylated prior to its relocation to peroxisomes?

Are any other mitochondrial proteins that MITOL associates with (e.g. Drp1) relocated with MITOL?

If mitophagy is blocked, downstream of Parkin, for example by blocking autophagy, does MITOL still relocate to peroxisomes?

In peroxin-deficient cells, is any MITOL relocated to the ER, via which several PMPs traffic to peroxisomes.

Is mono- or poly-ubiquitylation at K268 involved?

1st Revision - authors' response

28th Jun 2019

We would like to thank the three Reviewers for their comments as their suggestions allowed us to obtain new results and identify areas in the manuscript that needed greater clarification. We have performed additional experiments and expanded the dataset to address specific Reviewer concerns. In addition, Mayumi Kimura was added as a co-author for her contributions during the revision.

The point-by-point responses to the Reviewer comments are listed below.

Reviewer #1:

In this manuscript Koyano and colleagues reported that MITOL/March5, a mitochondrial ubiquitin ligase, translocates from mitochondria to peroxisomes after mitophagy stimulation with membrane depolarizing agents. Using confocal microscopy and immunofluorescence, authors observed that

MITOL specifically relocates to peroxisomes in a Parkin-dependent manner. They also found that this new MITOL translocation is mediated by the ATPase p97/VCP and peroxins such as Pex3. Finally, by mass spectrometry analysis, they found that ubiquitination of K268 by Parkin is required for MITOL translocation. Overall, this is an interesting manuscript and highlights not only the cross talk between mitochondria and peroxisomes, but also that Parkin may have functions outside of simply marking mitochondria for mitophagy. The whys and wherefores of this translocation are unknown and as such this work is somewhat phenomenological; however, my feeling is that this should not preclude publication as there is a lot of interest in this area. I also have other concerns as detailed below.

Our reply:

We thank the Reviewer for the positive evaluation of our manuscript. In response to the Reviewer's comments, we have incorporated the results of a number of additional experiments (e.g. new Fig. 1, 2, 3, 6, 7, 8, 9, and 10) as elaborated below. I believe that these new results effectively address the Reviewer's concerns.

Major points:

1. The vast majority of conclusions are based on fluorescence microscopy colocalization interpretations, but quantitative data are completely absent. To support their conclusions and to show experimental variation, authors should perform quantitative colocalization and statistical analyses.

Our reply:

We thank the reviewer for this constructive comment. We agree that quantitative colocalization and statistical analyses should be performed and thus used Zen software (Carl Zeiss) to calculate colocalization between MITOL and other proteins as Pearson correlation coefficients (note - we cannot calculate colocalization between MITOL and peroxisomes as *PEX19*^{-/-} cells have no peroxisomes in Fig. 6). These data have been incorporated into the revised manuscript as new Fig. 1E, 1H, 2B, 3C, 3D, 7D, 8E, 9E, 10C, and 10E.

2. Related to the above point, MITOL may translocate to peroxisomes, but is this every one? What is the proportion of MITOL-positive peroxisomes after mitophagy induction and could recruitment be specific to a subpopulation of peroxisomes?

Our reply:

As shown in the new Fig. 1 and Fig. EV1D, our data indicate that MITOL translocated to all peroxisomes and did not translocate to specific subpopulations of peroxisomes. Regarding the proportion of MITOL-positive peroxisomes, the Pearson correlation coefficients (determined using Zen software; Carl Zeiss) were about 0.4 between MITOL-HA and catalase (Fig. 1E) or MITOL-GFP and PMP34-FusionRed (Fig. 1H). Under sufficient colocalization conditions, Pearson correlation coefficients are typically 0.5 – 0.6, confirming that MITOL localizes on a significant proportion of peroxisomes. Because the resolution of peroxisomes (viewed as very small dots scattered in the cytoplasm) in our original figures made it difficult to discern colocalization between MITOL and peroxisomes, we have included higher magnification images in almost all of the figures in the revised manuscript (also in our response to comment #9 of Reviewer #1).

3. I appreciate that the vast majority of the field overexpresses Parkin to visualise mitophagy, despite this being somewhat artefactual it may not even be required for MITOL translocation. There are many cell lines that express easily detectable levels of (and activatable) endogenous Parkin (such as SH-SY5Y) and the authors should determine if MITOL translocation also occurs in an endogenous Parkin-dependent manner in these.

Our reply:

As suggested by the Reviewer, we utilized the SH-SY5Y human neuroblastoma cell line to determine if endogenous levels of Parkin are able to cause the transition of MITOL from damaged mitochondria to peroxisomes. SH-SY5Y cells transiently expressing 3Flag-MITOL were treated with valinomycin + ZVAD-FMK for 3 or 6 hours, then analyzed by immunofluorescence (note - ZVAD-FMK is an apoptosis inhibitor that was included in the incubation because SH-SY5Y cells are more sensitive to uncoupler treatment and detach easily). As shown below, MITOL once again redistributed to peroxisomes (*Figure 1 for Reviewer #1, below*). However, the frequency of MITOL

redistribution in SH-SY5Y cells appeared to be lower than that observed when overexpressed in HeLa cells. We think this difference arises because of the atypical physiological conditions. Typically, only a minor population of mitochondria are damaged, and thus the physiological levels of Parkin are sufficient to manage them. However, uncoupler treatments with valinomycin or CCCP cause depolarization of all mitochondria, a cellular event that is unlikely to happen under physiological conditions, and as a result the endogenous levels of Parkin are overwhelmed. Given that the endogenous levels of Parkin in SY-SH5Y cell lines are much lower than overexpression conditions, the endogenous Parkin might be insufficient to assist in relocation of all MITOL molecules from damaged mitochondria to peroxisomes. However, as shown in *Figure 1 for Reviewer #1 below*, we do observe peroxisomal localization of MITOL without exogenous Parkin expression.

Figure 1 for Reviewer #1

4. Figure 2, authors should analyse MITOL localisation after CHX and CCCP treatment. Is MITOL still localized in peroxisomes? This will support the notion that pre-existing MITOL moved to peroxisomes following mitochondrial depolarisation.

Our reply:

We thank the reviewer for this keen criticism. Thanks to this comment, we realized the following problem about CHX-treatment experiments.

Parkin translocation to depolarized mitochondria depends on newly-synthesized PINK1 that has accumulated on the outer mitochondrial membrane when PINK1 import into matrix has been inhibited. Therefore, we cannot use CHX as it blocks new PINK1 synthesis and consequently also blocks Parkin translocation (see also Fig. 7 of Narendra et al., PLoS Biol 2010, 8, e1000298). Because cells were treated with both CCCP and CHX in Fig. 2C of the original manuscript, corresponding data were deleted from the revision manuscript.

Instead of using CHX, we utilized a dox-inducible system to test if pre-existing MITOL on the mitochondria moves to peroxisomes directly. HeLa cells stably expressing HA-Parkin were transiently transfected with pTRE3G-3Flag-MITOL and pCMV-Tet3G plasmids. Following doxycycline treatment for 3 hours to induce expression of MITOL, we observed that MITOL was localized to mitochondria (*Figure 2 for Reviewer #1, below*). After repeated wash out of doxycycline to terminate MITOL synthesis, the cells were treated with CCCP for more than 3 hours.

Under these conditions, we found that a major portion of MITOL had merged with Pex14 (peroxisomal membrane protein), suggesting strongly that pre-existing MITOL on mitochondria had moved to the peroxisomes following CCCP treatment. Additional data that showed MITOL could not localize to peroxisomes when Tom70 was knocked-down (new Fig. 5) also support this conclusion (i.e. newly synthesized MITOL in Tom70 knocked-down cells accumulated in the cytoplasm and did not localize on peroxisomes). These important new findings have been incorporated into the revised manuscript as new Fig. 2E.

Figure 2 for Reviewer #1

5. In Figure 2B, a positive control is missing (e.g. mitochondrial protein known to be degraded by the proteasome and mitophagy - such as mitofusin1/2). Also, loading controls are missing in panel B as well as C.

Our reply:

We are grateful for this comment as Reviewers #1 and #2 identified the same issue in Fig. 2 (see also our response to comment #4 for Reviewer #2). Regarding positive control, Reviewer #2 also requested to blot for positive mitophagy markers. Moreover, Reviewer #2 requested to perform experiments not only in whole cell lysates, but also in subcellular fractions. Together with this comment #4 of Reviewer #2, we compared MITOL with Mitofusin2 (Mfn2). As shown in *Figure 3A for Reviewer #1* below, extended valinomycin treatment combined with fractionation analysis for 6 hours confirmed that the total cellular amount of MITOL had not been dramatically decreased, whereas the positive control Mitofusin2 (MFN2) underwent rapid degradation within 3 hours of Valinomycin treatment.

Regarding loading control, tubulin blotting have been incorporated in the revised manuscript as new Fig. 2C (also shown below as *Figure 3B for Reviewer #1*, below). Moreover, in this image, we used an anti-ubiquitin antibody to confirm Parkin-mediated mitophagy (compare lanes 2 and 5 in *Figure 3B for Reviewer #1*) and also show that the proteasomal inhibitor MG132 blocked degradation of ubiquitylated proteins (lanes 3 and 6 in *Figure 3B for Reviewer #1*). These important new findings have been incorporated into the revised manuscript as new Fig. 2C and 2D.

Figure 3 for Reviewer #1

A

**B**
6-1. At multiple instances, the authors refer to the translocation as mitophagy-induced, when actually it is depolarisation-induced. While mitophagy and depolarisation are linked they are not the same and there is no mitophagy monitoring in the manuscript to distinguish between the two.

Our reply:

We agree with the Reviewer's comment and have changed the text for example from "MITOL redistributes from mitochondria to peroxisomes during mitophagy." to "MITOL redistributes from mitochondria to peroxisomes in response to mitochondrial depolarization in a Parkin-dependent manner (or mitophagy stimulation)."

6-2. Does actual mitophagy influence, or be influenced by, translocation? For example, is translocation of MITOL required for mitophagy progress i.e. can mitophagy still occur with ubiquitination-deficient MITOL?

Our reply:

We thank the Reviewer for this insightful comment as it inspired us to use flowcytometry as a means of investigating whether the translocation-defective mutant of MITOL represses mitophagy progression or not. To quantify mitophagy, we utilized HCT116 cells stably expressing both Parkin and the mitochondria-localized pH-dependent fluorescent protein mt-Keima, which has a short excitation wavelength under neutral conditions (e.g., in mitochondria) and undergoes a shift to

longer excitation wavelengths under acidic conditions (e.g., in lysosomes) [Katayama et al. (2011) Chem Biol. 18, pp1042-52]. Consequently, mitophagy progression can be monitored as a proportion of cells in which mt-Keima undergoes an acidification-specific fluorescence change.

To examine the effect of Parkin-mediated ubiquitylation and peroxisomal trans-localization of MITOL on mitophagy, we compared mitophagy activity between *MITOL* knockout HCT116 cells complemented with wild-type MITOL (which can translocate to peroxisomes) and *MITOL* knockout cells complemented with the K268A MITOL mutant (which is defective in peroxisomal translocation). The mitophagic flux in Δ *MITOL* cells complemented with MITOL K268A was equivalent to that in Δ *MITOL* cells complemented with wild-type MITOL, suggesting that the retention of MITOL on depolarized mitochondria does not inhibit mitophagy progression (*Figure 4 for Reviewer #1, below*).

Regarding the Reviewer's first question "Does actual mitophagy influence translocation?", we ask the Reviewer to see our rebuttal to comment #6 of Reviewer #3 below to avoid repetition.

7. The requirement of peroxisomes for mitochondrial MITOL removal is very interesting and suggests peroxisomes are involved in the translocation process (and not just the end destination). Can the authors monitor the translocation by live cell microscopy as this might give more of an insight, for example mitochondrial-peroxisome contact sites may be critical for this translocation?

Our reply:

Based on this comment, we examined if mitochondrial-peroxisome contact sites are critical for MITOL translocation, and if peroxisomes are a midpoint in the translocation process rather than the end destination. Following extended CCCP treatment, HeLa cells stably expressing HA-Parkin and 3Flag-MITOL were stained with the mitochondrial marker Tom20 (new Fig. EV1F: because Tom20 was substantially degraded via mitophagy, Tom20-positive cells were selected for imaging at 12h and 24h CCCP treatment), the ER marker Sec61 β (new Fig. EV1E), and the peroxisome marker Pex14 (new Fig. EV1D). MITOL merged with Pex14 at a peak of 3 hours post-CCCP treatment with a reduction in the signal observed over time. Although MITOL colocalized with Pex14, no colocalization was observed for either Tom20 or Sec61 β .

Following the extended CCCP treatment period (for > 3h), Parkin-catalyzed ubiquitylation resulted in the perinuclear clustering of depolarized mitochondria (Okatsu et al., Genes Cells 2010, 15, 887-900; also shown in panel two of new Fig. EV1F). If MITOL localized at mitochondria-peroxisome contact sites as suggested, it should be concentrated around the perinuclear region. However, this immunofluorescence pattern was not observed, suggesting MITOL localization at mitochondria-peroxisome contact sites does not occur.

8. In the Discussion, the authors should at least discuss/speculate a small bit about the biological function of mitochondria-to-peroxisome MITOL translocation after depolarization.

Our reply:

In response to comment #1 of Reviewer #3, we obtained an important clue to the functional relevance of mitochondria-to-peroxisome MITOL translocation. Based on newly obtained results, we have referred our hypothesis about the biological function of MITOL translocation in the Discussion (see also our response to comment #1 for Reviewer #3).

Minor points:

9. For better visualisation of colocalization between proteins, some higher magnification views would help the reader.

Our reply:

We thank the reviewer for this constructive comment and have included higher magnification images in Figs. 1, 2, 3, 6, 8, and 10.

10-1. Figure 6, results from western blot analysis don't match with immunofluorescence images. In Panel A, in Tom70-silenced cells, MITOL level seems increased compared to control. Yet in Panel B, MITOL level seems decreased in Tom70-silenced cells. Authors should clarify this point.

Our reply:

We appreciate this keen criticism. In general, immunocytochemical signals are intense when the target protein localizes on/in the specific organelle, whereas the signals are weak when proteins are dispersed in the cytosol. Moreover, because the target protein is not completely denatured in immunocytochemistry, it is possible that the epitope is masked by the specific conformation, and in this case the anti-flag antibody was inaccessible to the cytosol-released form of MITOL. In both cases, the immunoblotting signal more accurately reflects the MITOL amount. Nevertheless, based on the Reviewer's comment, we performed subcellular fractionation of *TOM70* knocked-down cells. Although a significant portion of MITOL became detached from the mitochondria during fractionation, a sufficient amount was retained in the mitochondria-enriched fraction of control cells (lanes 2 and 3 in Figure 5 for Reviewer #1, below). In contrast, almost all of the MITOL was collected in the cytosolic fraction of si*Tom70*-treated cells with no signal observed in the mitochondria-enriched fraction (lanes 5 and 6 in Figure 5 for Reviewer #1, below). This result is consistent with the immunocytochemical data that showed dispersion of MITOL into the cytosol following a reduction in Tom70. The fractionation data have been incorporated into the revised manuscript as new Fig. 5D.

10-2. In addition, in Panel C it is not clear how the authors quantify the % of cells with the indicated protein in cytosol. Indeed Su9-GFP seems to be enriched in the nucleus in Tom20- and Tom40- silenced cells. Some explanation would help.

Our reply:

In the original Fig. 6C (Fig. 5C in the revised manuscript), in each siRNA experiment HeLa cells exhibiting cytosolic localization of Su9-GFP or 3Flag-MITOL were counted. As Reviewer #2 commented, Su9-GFP accumulated in the cytosol and nucleus of *TOM20*- and *TOM40*-silenced cells. Although we do not know the reason for this distribution, the results are consistent with published data (Otera et al. 2007, JCB 179, 1355-63). We have added the following sentence to the new manuscript “Su9-GFP was not imported into the mitochondria, but rather localized to the cytosol and nucleus when *Tom20* and *Tom40* were knocked down.”

11. Some experiments are done with CCCP and some with valinomycin - is there any specific reason for a lack of consistency? Are the authors sure that they both have the same effect?

Our reply:

Because HCT116 cells are prone to shedding, a milder uncoupler, valinomycin, was used in all experiments with the HCT116 cells. In HeLa cells, we first used CCCP to observe MITOL translocation, but to confirm that MITOL translocation is not a CCCP-specific event, valinomycin was also used. In both cases, we confirmed that treatment with CCCP or valinomycin induced MITOL redistribution to peroxisomes as shown in Fig. 1 and Fig. EV1.

Reviewer #2:

In this study, Koyano and colleagues propose that the mitochondrial E3 ubiquitin ligase MITOL/March5 translocates to peroxisomes during mitophagy. They further suggest that PINK1, peroxins and E3 activity of Parkin, but not MITOL E3 activity, are essential for VCP-dependent extraction of MITOL from mitochondria. Overall this is an interesting study, which could highlight a new important role for ubiquitin in the redistribution of MITOL (and possibly other proteins) from depolarised mitochondria to peroxisomes.

However, I have two major concerns, which in my opinion, preclude this study from being published in EMBO Reports (at least in the current form):

We appreciate the Reviewer's positive evaluation that our work highlights a new role for ubiquitin in the redistribution of a mitochondrial protein to peroxisomes. We agree that the original study had a number of weaknesses as indicated by the Reviewer and have performed a number of additional experiments in response to those comments.

I believe that the new results effectively address all of the Reviewer's comments, including the two major concerns. The specific revision details are described in the point-by-point responses below.

1.- There is quantification/statistical analysis throughout the manuscript (except for Figure 6C). A quantification should be included, at the very least in ALL the imaging experiments (especially as a vast majority of the conclusions are based on immunofluorescence). In the current version, it is impossible to assess the robustness of the findings.

Our reply:

We thank the Reviewer for this constructive comment. We used Zen software (Carl Zeiss) to calculate colocalization between MITOL and other proteins as Pearson correlation coefficients in all of the immunocytochemical experiments. These quantitative results have been incorporated in the revised manuscript and are included in new Fig.1E, 1H, 2B, 3C, 3D, 7D, 8E, 9E, 10C, and 10E. (note - *PEX10*^{-/-} cells have no peroxisomes and thus we cannot calculate colocalization between MITOL and peroxisomes in Fig. 6).

2. - The study is largely based on over-expression. They authors have generated a MITOL-3Flag KI cell line. When possible, why not performing the experiments in this cell line?

Our reply:

I appreciate this constructive suggestion. We agree with the idea that the key experimental results using MITOL-overexpressing cells should be confirmed using MITOL-3Flag knock-in cells. In the original Fig. 9, we showed that translocation of exogenous MITOL to peroxisomes was inhibited by a VCP inhibitor (NMS-873). We thus examined whether similar results could be obtained using the MITOL-3Flag knock-in cell line. When MITOL-3Flag knock-in HCT116 cells stably expressing HA-Parkin were treated with valinomycin for 3 hours, the 3Flag-tagged endogenous MITOL

overlapped with PMP70 (peroxisomal membrane protein 70) (*Figure 1A for Reviewer #2, below*) indicating endogenous MITOL localized to peroxisomes following dissipation of the mitochondrial membrane potential (this result is consistent with Fig. 5 in the original manuscript). In contrast, the 3Flag-tagged endogenous MITOL failed to redistribute to peroxisomes following the combined administration of NMS-873 and valinomycin (*Figure 1A for Reviewer #2, below*).

As shown in the original Fig. 10, exogenous MITOL undergoes ubiquitylation prior to peroxisomal translocation. We thus examined if endogenous MITOL likewise undergoes ubiquitylation. MITOL-3Flag knock-in HCT116 cells stably expressing HA-Parkin were treated with valinomycin and NMS-873 for 3 hours, immunoprecipitated using anti-Flag magnetic beads, and immunoblotted using anti-Flag and anti-ubiquitin antibodies. The immunoblots indicate that 3Flag-tagged endogenous MITOL was also ubiquitylated in response to a loss in the mitochondrial membrane potential (*Figure 1B for Reviewer #2, below*). We have thus confirmed that key findings (i.e VCP- and Ubiquitin-dependent translocation of MITOL from depolarized mitochondria) reported in the original manuscript hold true for both over-expressed and endogenous MITOL. Given the importance of the aforementioned results, these new data have been incorporated into the revised manuscript as new Fig. 10.

Figure 1 for Reviewer #2

Specific comments:

3. - In Figure 1, the authors should perform subcellular fractionations to confirm MITOL peroxisomal localization following CCCP or valinomycin treatment. Moreover, the authors should perform a time course of CCCP.

Our reply:

We thank the Reviewer for this critical comment. We consequently sought to use subcellular fractionation to quantify MITOL localization. Technical difficulties, however, needed to be overcome as the density of mitochondria is close to that of peroxisomes in cultured cells, and the fractionation process caused MITOL to detach from the mitochondrial membrane. To overcome these challenges, we added a VCP inhibitor (NMS-873) to the lysis buffer during lysate collection. The lysates were then subjected to a multi-step centrifugation process. The PNS (post-nuclear supernatant) fraction collected from HCT116 cells stably expressing HA-Parkin and 3Flag-MITOL was centrifuged at 3,000 g to obtain a 3,000 g pellet. The resulting supernatant was further centrifuged at 100,000 g to yield a 100,000 g pellet. Mitochondria were recovered in the 3,000 g pellet, whereas the 100,000 g pellet was almost free from mitochondria. In contrast, although peroxisomes were collected in both fractions, they were enriched in the 100,000 g pellet (*Figure 2A for Reviewer #2*, below). We then compared the recovery ratio of 3Flag-MITOL in the 3,000 g and 100,000 g pellets. Prior to valinomycin treatment, a large proportion (ca. 80%) of 3Flag-MITOL was collected in the 3,000 g pellet and only a small proportion (ca. 20%) in the 100,000 g pellet. Conversely, the ratio of 3Flag-MITOL recovered in the peroxisome-rich 100,000 g fraction increased from 20.4% to 36.9% following valinomycin treatment for 3 hours (*Figure 2B for Reviewer #2*, below). This result supports our model that MITOL redistributes from damaged mitochondria to peroxisomes. These fractionation data have been added to the revised manuscript as new Fig. 11.

Figure 2 for Reviewer #2

A

4. - In Figure 2 B and C, there is no loading control. In Figure 2B, the authors suggest that MITOL is not degraded via mitophagy, as the MITOL levels don't change drastically after CCCP treatment. In order to draw such conclusion, the authors should extend the CCCP treatment (currently 3hrs only), and blot for positive mitophagy markers. Moreover, these experiments should be performed not only in whole cell lysates, but also in subcellular fractions.

Our reply:

Reviewers #1 and #2 had the same comment for Fig. 2B and 2C (also see our response to comment #5 of Reviewer #1). A loading control using an anti-tubulin antibody has been incorporated in the new Fig. 2C in the revised manuscript. Moreover, according to this comment, we extended the uncoupler treatment period for 6 h and analyzed the amount of MITOL and Mfn2 (positive degradation marker requested from Reviewer #1) in the subcellular fractionation. We confirmed that the total cellular amount of MITOL had not been dramatically decreased, whereas MFN2 underwent rapid degradation within 3 hours of valinomycin treatment especially in mitochondria-rich fractions (new Fig 2D).

5. - In Figure 5B, it seems that a consistent fraction of MITOL is still located at the mitochondria after CCCP treatment. The authors should provide better resolution pictures, and of course quantify the proportion of MITOL at the mitochondria/peroxisome following DMSO/CCCP or valinomycin treatment.

Our reply:

As indicated, high resolution pictures have been added to the corresponding Figure (Fig. 5B in the original manuscript and now Fig. 10B in revision). Moreover, based on the Reviewer's comment,

subcellular fractionation was done to quantify the proportion of endogenous MITOL that localized to the mitochondria and peroxisomes following mitophagy stimulation (Fig. 11C in revision). Similar to overexpressed MITOL (please see our response to Reviewer #2 comment #3 above), we compared the ratio of endogenous MITOL recovered in the 3,000 g pellet and the 100,000 g pellet following valinomycin treatment. Endogenous MITOL in the 3Flag knock-in cells was largely (ca. 72%) detected in the mitochondria-enriched 3,000 g pellet (lane 2 of *Figure 3A for Reviewer #2*, below) with a substantially lower amount (ca 28%) recovered in the peroxisome-enriched 100,000 g pellet prior to valinomycin treatment (quantified data are also shown as *Figure 3B for Reviewer #2*, below). Valinomycin treatment for 3 hours decreased the proportion of endogenous MITOL localized to mitochondria from 72% to 45% (lane 5 of *Figure 3A for Reviewer #2*, below). At the same time, a significant amount of MITOL (ca. 55%) was collected in the peroxisome-enriched 100,000 g pellet (lane 6 of *Figure 3A for Reviewer #2*, below). These results confirm the immunocytochemical data, which showed that endogenous MITOL also redistributes to peroxisomes following valinomycin treatment. Given the importance of these fractionation experiments, the data have been incorporated into the revised manuscript as new Fig. 10 and Fig. 11.

Figure 3 for Reviewer #2

6-1. - In Figure 6C, have stats been performed?

Our reply:

Based on the Reviewer's comment, statistical analysis results for each condition in Fig. 6C have been incorporated into the revised manuscript (new Fig. 5C).

6-2. In Figure 6A, the authors should perform sub-cellular fractionations to ascertain that MITOL is in the cytosolic fraction of the TOM70 siRNA cells.

Our reply:

We thank Reviewer for the comment – again both Reviewer #1 and #2 identified the same point for further clarification. In general, immunocytochemical signals are intense when the target protein localizes on/in the specific organelle, whereas the signals are weak when proteins are dispersed in the cytosol. Moreover, because the target protein is not completely denatured in immunocytochemistry, it is possible that the epitope is masked by the specific conformation, and in this case the anti-flag antibody was inaccessible to the cytosol-released form of MITOL. In both cases, the immunoblotting signal more accurately reflects the MITOL amount. However, for completeness we also performed subcellular fractionation of *TOM70* knocked-down cells. Although a significant portion of MITOL detached from the mitochondria during fractionation, a sufficient amount was retained in the mitochondria-enriched fraction of control cells (*lanes 2 and 3 in Figure 7 for Reviewer #1*). In contrast, almost all of the MITOL was collected in the cytosolic fraction of *siTom70*-treated cells with no signal observed in the mitochondria-enriched fraction (*lanes 5 and 6 in Figure 5 for Reviewer #1*). This result is consistent with the immunocytochemical data that showed dispersion of MITOL into the cytosol following a reduction in Tom70. The fractionation data have been incorporated into the revised manuscript as new Fig. 5D.

7. - In Figure 8A, to evaluate the KD, the authors over-express the different Pex and then KD. If not antibody is available, the authors should evaluate the KD of endogenous PEX by qPCR. The authors conclude that "the most pronounced inhibitory effects on MITOL translocation were observed following PEX3 KD" and that "the role of Pex3 is particularly significant". Again, how can they make such statements with no quantification?

Our reply:

Based on the Reviewer's comment, we purchased antibodies for detecting endogenous Pex3, Pex16, and Pex19 and assessed the degree of knockdown for each. Immunoblotting data confirmed siRNA-mediated reduction in all three endogenous Pex proteins (Pex3, Pex16, and Pex19) (*Figure 4A for Reviewer #2, below*).

To address the Reviewer's concern regarding our conclusion that "the most pronounced inhibitory effects on MITOL translocation were observed following PEX3 KD" was not fully supported by quantification, colocalization between 3Flag-MITOL and catalase was quantified using Pearson correlation coefficients. These results confirmed our previous finding regarding the pronounced inhibitory-effect on MITOL translocation to peroxisomes in *siPEX3*- and *siPEX16*-treated cells, with PEX3 knock-down having the most pronounced inhibitory effects (*Figure 4B for Reviewer #2, below*). Indeed, in *siPEX3*-treated cells, no significant difference ($P=0.16$) in the colocalization of 3Flag-MITOL and catalase was observed between CCCP-treated and untreated conditions. The quantification data have been added into the revised manuscript (new Fig. 7D).

Figure 4 for Reviewer #2

8. - In Figure 9, have the authors considered using patient cells from VCP mutations carriers?

Our reply:

I appreciate the constructive suggestion. However, we have not considered this type of experiment as cells from VCP mutation carriers are unavailable to us. I will remember the suggested experiment should the opportunity arise for a future study.

9. - In Figure 10, the authors claim that MITOL ubiquitination is mediated by Parkin, but have they performed the same experiments in WT HeLa cells?

Our reply:

As per the Reviewer's suggestion, the ubiquitylation of MITOL was examined in wild-type HeLa cells. Although MITOL was ubiquitylated following CCCP and NMS-873 treatment in Parkin-expressing cells as shown in Fig. 10 in the original manuscript, no ubiquitylation signal was observed in wild-type HeLa cells lacking endogenous Parkin under the same experimental conditions (Figure 5 for Reviewer #2, below). This result confirms that Parkin ubiquitylates MITOL in response to mitophagy stimulation. Given the importance of this result, these data have been incorporated into the revised manuscript as new Fig. 9A.

Figure 5 for Reviewer #2
Reviewer #3:

This is an intriguing paper by Koyano et al. describing the relocation of a mitochondrial outer membrane E3 ligase, called MITOL/March5, from the outer membrane of damaged mitochondria to peroxisomes via a process that depends on the ubiquitylation of MITOL at K268, the activity of the mitochondrial E3 ligase Parkin but not that of MITOL itself, the peroxins Pex3, Pex16 and Pex19 on peroxisomes and the AAA-ATPase p97/VCP, which is believed to extract MITOL from the mitochondrial outer membrane.

Although the authors describe well several steps of this relocation process, the principal missing component is the physiological relevance and the generality of this process, which raises many important questions regarding whether this represents a significant advance or an oddity. Additionally, the mechanism of extraction of a protein from mitochondria and reinsertion into peroxisomes raises many key mechanistic questions that are unanswered. Consequently, this paper is too premature at present in terms of significance and mechanisms for publication in EMBO Reports.

Our reply:

We sincerely thank Reviewer #3 for identifying weak points in our manuscript, and for helping to improve the study overall. However, we resolutely disagree with the comment “*the mechanism of extraction of a protein from mitochondria and reinsertion into peroxisomes raises many key mechanistic questions that are unanswered. Consequently, this paper is too premature at present in terms of significance and mechanisms for publication*” as we have revealed the molecular mechanism underlying this new phenomenon of protein extraction from mitochondria and reinsertion into peroxisomes. In this manuscript, we show that Parkin-catalyzed ubiquitylation at Lysine 268, p97/VCP function, and the Pex3/16/19 pathway (in particular Pex3, which has a pronounced effect on peroxisomal reinsertion) underpin MITOL extraction from mitochondria and reinsertion into peroxisomes. Moreover, our paper demonstrates a novel function for ubiquitylation as it accelerates translocation of a substrate from mitochondria to peroxisomes.

Nevertheless, to address this perceived weakness, and to avoid similar perceptions by readers, we performed a number of additional experiments. Specific details are presented in the point-by-point responses to the Reviewer’s comments below.

I enclose some comments to help the authors improve their manuscript.

1. What is the physiological role of MITOL and how is that affected by its relocation to peroxisomes? For example, does it affect peroxisome membrane permeability (Hosoi et al., 2017; PMC5350511) or mitophagy?

Our reply:

I appreciate the constructive suggestion that provided us important data. It is unlikely that retention of MITOL on damaged mitochondria is cytotoxic and thus is transported to peroxisomes (if so, it should be sufficient for MITOL to be degraded like other Parkin substrates). We thus surmise that translocation of MITOL is not to eliminate MITOL, but rather to assist MITOL function on peroxisomes. During systematic deletion analyses of MITOL, we happened to find that MITOL mutant lacking C-terminal 8 amino acids (MITOL Δ C8) causes peroxisomal expansion after CCCP treatment in the presence of Parkin (*Figure 1A for Reviewer #3*, below). To examine whether the E3 activity of MITOL is required for expansion of peroxisomes, we mutated Cys65/Cys68 within the RING-domain to Ser (CS) or the conserved Zn-binding His43 to Trp (H43W). Importantly, these E3-inactive MITOL mutants lacking C-terminal 8 amino acids (CS/ Δ C8 and H43W/ Δ C8) localized to peroxisomes following CCCP treatment but did not expand the peroxisomes (*Figure 1B for Reviewer #3*, below). We speculate that Δ C8 converts MITOL to the constitutive-active form via de-repression. When ubiquitin was immune-stained, ubiquitin signal was observed not only on damaged mitochondria that ubiquitylated by Parkin but also on enlarged peroxisomes that overlapped with MITOL Δ C8 (*Figure 2 for Reviewer #3*, below). Ubiquitylation on expanded peroxisomes were canceled by E3-inactive MITOL mutation (*Figure 2 for Reviewer #3*, below). These results suggest that peroxisomes are highly ubiquitylated by translocated MITOL Δ C8 upon CCCP treatment, suggesting MITOL has a potential to regulate abundance and size of peroxisomes. Mitochondria and peroxisomes share some metabolic pathways, and their distribution and abundance are highly coordinated to maintain cellular homeostasis. Although the functional

relevance of ubiquitin-dependent MITOL translocation to peroxisomes has not been unveiled completely, it is possible that MITOL regulates abundance of peroxisomes when mitochondria are damaged (see Discussion). These data have been incorporated into the revised manuscript as new Figs 12 and 13.

Figure 1 for referee #3

Figure 2 for referee #3

2-1. When MITOL is relocated to peroxisomes, this should also be shown biochemically.

Our reply:

We thank the Reviewer for this comment. Indeed, the other Reviewers expressed the same concern. As shown in new *Figure 11 in the revised manuscript*, we provide biochemical support for MITOL relocation to peroxisomes. Fractionation experiments confirmed that MITOL translocates from depolarized mitochondria to peroxisomes. To avoid repetition, please see our responses to comments #3 and #5 of Reviewer #2.

2-2. Also, is MITOL an integral membrane protein in peroxisomes and does its topology resemble that found in mitochondria? What is its ubiquitylation status?

Our reply:

The topology of MITOL is oriented such that both the N- and C-terminus are directed to the cytoplasm (Fig. 4A). Our MS data support this topology as K40 in the N-terminus hydrophilic portion and K268 in the C-terminus hydrophilic portion were ubiquitylated (note that ubiquitylation never happens in the intermembrane and matrix space in mitochondria: Fig. 9). To examine whether MITOL has the same topology in peroxisomes, immunocytochemistry was performed using a weak permeabilization reagent. When HeLa cells were permeabilized with 25 $\mu\text{g}/\text{mL}$ digitonin, peroxisomal matrix proteins were not stained because the peroxisomal membrane was not sufficiently permeabilized and the antibody was unable to access the peroxisomal lumen (digitonin panel in *Figure 3 for Reviewer #3*, below). Similar immunocytochemistry analyses under weak-permeabilization conditions was done using N-terminally or C-terminally 3Flag-tagged MITOL. After CCCP treatment, peroxisomal localization puncta were detected for both 3Flag-MITOL and MITOL-3Flag (digitonin panel in *Figure 3 for Reviewer #3*, below), and these puncta merged with catalase (peroxisomal matrix protein) when permeabilized with TritonX-100 (Triton panel in *Figure 3 for Reviewer #3*, below). We thus concluded that the topology of MITOL is identical in mitochondria and peroxisomes.

Figure 3 for Reviewer #3

3. Is this a new pathway of peroxisomal membrane protein (PMP) targeting to peroxisomes, in which case it is very poorly characterized? Does MITOL have a membrane peroxisomal targeting signal (mPTS)? It appears not to because in the absence of mitochondrial targeting via Tom70, MITOL is cytosolic and not peroxisomal. What is the role of ubiquitylation in the activation of the mPTS?

[Figures for referees not shown.]

4. Does MITOL need to be deubiquitylated prior to its relocation to peroxisomes?

Our reply:

We thank the Reviewer for this constructive comment. To answer this query, we focused on the deubiquitylating enzyme. Given that USP30 localizes to both mitochondria and peroxisomes and is involved in Parkin-mediated mitophagy (Sato Y et al, Nat Struct Mol Biol. 2017; Marcassa E et al, EMBO Rep. 2018; Riccio V et al, J Cell Biol. 2019, Bingol B et al., Nature 2014), we sought to determine if USP30 also contributes to MITOL translocation. In *USP30* knockout HeLa cells, the translocation of MITOL from mitochondria to peroxisomes was equivalent to that in *USP30*^{+/+} wild-type HeLa cells (Fig EV6). Moreover, the ubiquitylation status of MITOL in *USP30* knockout cells (lanes 1-3 of Fig EV5C) was almost identical to that in *USP30* expressing HeLa cells (lanes 4-9). These results suggest that at least Usp30-catalyzed de-ubiquitylation is not involved in MITOL translocation to peroxisomes.

5. Are any other mitochondrial proteins that MITOL associates with (e.g. Drp1) relocated with MITOL?

Our reply:

I thank the Reviewer for this constructive comment. So far, Drp1, Fis1, Mitofusin (MFN) 1/2, MiD49, and McI1 have been reported to associate with MITOL (Yonashiro R et al, EMBO J. 2006; Nakamura N et al, EMBO Rep. 2006; Park YY et al, J Cell Sci. 2010; Xu S et al, Mol Biol Cell. 2016; Cherok E et al, Mol Biol Cell. 2017). Among these MITOL-associate proteins, we showed in

the original manuscript that Fis1 did not translocate to peroxisomes (original Fig. 4B). To expand on this, we performed immunocytochemistry to see if other MITOL-associated proteins localized to peroxisomes following CCCP treatment. Because antibodies that detect endogenous MFN2, MiD49, and McI1 are not available, we constructed tagged plasmids to observe the subcellular localization of MFN2, MiD49, and McI1. We found that none of the proteins (Drp1, MFN1, MFN2, McI1, and MiD49) was predominantly localized to peroxisomes following CCCP treatment, indicating that other mitochondrial proteins that associate with MITOL do not relocate to peroxisomes. These new data have been incorporated into the revised manuscript as new Fig. EV2.

6. If mitophagy is blocked, downstream of Parkin, for example by blocking autophagy, does MITOL still relocate to peroxisomes?

Our reply:

To examine if mitophagy inhibition affects the peroxisomal translocation of MITOL, we utilized an *ATG5* knockout cell line (*ATG5* is indispensable for autophagy progression) stably expressing Parkin. When we analyzed the peroxisomal localization of MITOL in these cells by immunocytochemistry, we found that the degree of MITOL overlap with PMP70 following mitophagy stimulation was equivalent to that observed in wild-type cells (*Figure 5 for Reviewer #3*, below).

Figure 5 for Reviewer #3

[Figures for referees not shown.]

7. In peroxin-deficient cells, is any MITOL relocated to the ER, via which several PMPs traffic to peroxisomes.

Our reply:

As indicated by the Reviewer, it has been reported that several PMPs are transported to peroxisomes via the ER. If MITOL is also transported from damaged mitochondria to peroxisomes via the ER, we would expect ER accumulation of MITOL in peroxin-deficient cells. We thus examined the ER localization of MITOL. As shown below (*Figure 7A for Reviewer #3*), in depth colocalization analysis between MITOL and Sec61beta (ER marker) in *PEX19* knockout cells revealed no evidence in support of the localization of MITOL to the ER in peroxisome-deficient cells. Statistical analyses indicated a negative correlation between ER localization and valinomycin treatment, an effect that likely arose from the perinuclear clustering of damaged mitochondria (*Figure 7B for Reviewer #3*). These results suggest that MITOL does not move to peroxisomes via the ER.

Figure 7 for Reviewer #3

A

B

8. Is mono- or poly-ubiquitylation at K268 involved?

Our reply:

Stringently speaking, it is difficult to distinguish multiple mono-ubiquitylation of K268 (e.g., multiple monoubiquitylation at K40, K54, and K268) from polyubiquitylation at K268. However, because a single mutation at K268 (e.g. K268A) almost completely abolished ubiquitylation (lane 4 of Fig. 9B) and because MITOL showed a smear-like high-molecular weight modification (lane 2 of Fig. 9B), we surmise that polyubiquitylation of MITOL at K268 is important for translocation.

2nd Editorial Decision

30th Jul 2019

Thank you for the submission of your revised manuscript to EMBO reports. We have now received the full set of referee reports that is copied below.

As you will see, all referees are now positive about the study and support publication after some

remaining concerns have been addressed. Referee 1 suggests adding the reviewer figures to the manuscript since they contain important data. Referee 2 is concerned that the conclusion regarding unchanged MITOL protein levels needs additional experimental support. Referee 3 asks to clarify the observation that MITOL on peroxisomes is not ubiquitylated, while there is clearly ubiquitin on the peroxisomes.

From the editorial side, there are also a few things that we need before we can proceed with the official acceptance of your study.

- 1) Please provide up to five keywords.
- 2) Please use abbreviations for the author names in the Author Contributions section (e.g., FK for Fumika Koyano)
- 3) Please update the references to the numbered format of EMBO reports. The abbreviation 'et al' should be used if more than 10 authors. You can download the respective EndNote file from our Guide to Authors
<https://drive.google.com/file/d/0BxFM9n2IEE5oOHM4d2xEbmpxN2c/view>
- 4) Please note that we can accommodate only up to 5 EV figures. I suggest to provide the most important supplementary information in the form of EV figures and to combine all other supporting information into an Appendix.
The Appendix is a single pdf file that includes a table of content on the first page with page numbers, all figures and their legends. Please follow the nomenclature Appendix Figure Sx throughout the text and also label the Appendix figures according to this nomenclature. For more details please refer to our guide to authors.
- 5) Our data editors from Wiley have already inspected the Figure legends for completeness and accuracy. I have also taken the liberty to make some changes to the Abstract. Please see our suggested changes in the attached Word file.
- 6) We routinely check all figure callouts and noticed that callouts to Fig 10E, Fig 13C and the panels of Fig EV2 are missing in the text.
- 7) Fig EV7 only has one panel, therefore the 'A' label is not required.
- 8) Please provide scale bars in all magnification boxes.
- 9) EMBO reports papers are accompanied online by A) a short (1-2 sentences) summary of the findings and their significance, B) 2-3 bullet points highlighting key results and C) a synopsis image that is 550x200-400 pixels large (width x height). You can either show a model or key data in the synopsis image. Please note that the size is rather small and that text needs to be readable at the final size. Please send us this information along with the revised manuscript.

REFeree REPORTS:

Referee #1:

This is a re-review of the manuscript by Koyano et al., describing the translocation of MITOL from mitochondria to peroxisomes upon mitochondrial depolarisation. On the whole, the authors have done a good job in addressing my concerns. I just have some small minor points remaining.

1) The authors appear to have carried out some experiments for the benefit of the reviewers alone and not the future reader! I strongly suggest the authors add the new figures and brief descriptions in the text (can be as EV/supplemental) as I believe they not only enhance the paper but would also be expected - Figure 1 for reviewer #1 shows that this translocation is not an artefact of exogenous Parkin. This is important. Figure 4 for Reviewer #1 shows that the translocation is not necessary for

mitophagy per se. This is also important.

2) I appreciate that the authors have now added magnified images when looking at co-localization, and I also appreciate that peroxisomes are very small. However, it is still not easy to see overlap. Perhaps, and providing it does not make the image too crowded, the authors could add a few arrows to highlight representative examples of MITOL-Peroxisome co-localization.

Referee #2:

The authors have performed a considerable amount of work and have addressed most of my criticisms (quantification/statistical analysis, experiments with endogenous/3Flag MITOL, subcellular fractionations).

Very few additional comments:

- In New Figure 2D, the authors claim that "we confirm that total cellular amount of MITOL had not been dramatically decreased". However, this sentence is approximative/hazardous as (1) the levels of 3Flag MITOL do indeed seem to decrease in the PNS after 6h treatment and (2) the authors, yet again, don't perform any quantification. In New Figure 2D, the authors use Mfn as a mitophagy marker, however Mfn is known to be extracted from the depolarised mitochondria following ubiquitination, and as such it is not an ideal mitophagy marker. The authors should also use a matrix marker to assess clearance of damaged mitochondria after a longer (than 6 hours) treatment.

Minor comment: in response to my comment 6.2, the authors suggest that I refer to Figure 7 for reviewer 1. I assume it is Figure 5 for reviewer 1?

Referee #3:

The authors have made a very good attempt at answering my questions. My only concern is whether this is a physiological movement of MITOL to peroxisomes that is being described. The authors make a weak statement that the movement of MITOL to peroxisomes might affect peroxisome abundance and size, without any real quantification, especially of size, and that this depends on the E3 activity of MITOL. The phenomenon described is interesting enough and carefully done to warrant publication, but the physiological relevance remains to be rigorously proven.

In Fig. 13C, why is the MITOL that is peroxisome-associated not shown as being ubiquitylated, even though their data show ubiquitin on peroxisomes. Is this ubiquitin not on MITOL? This ambiguity should at least be acknowledged explicitly by a statement in the legend to Fig. 13C such as "Even though our data in A and B indicate the presence of ubiquitin on peroxisomes containing MITOL, we do not know the target of this ubiquitylation, so MITOL inserted into the peroxisome membrane is shown here in the non-ubiquitylated form."

2nd Revision - authors' response

30th Aug 2019

We would like to thank the three Reviewers for giving us the opportunity to improve our manuscript. We have performed a few additional experiments and expanded the dataset to address the Reviewer's remaining concerns. The point-by-point responses to the reviewer comments are listed below.

Referee #1:

This a re-review of the manuscript by Koyano et al., describing the translocation of MITOL from mitochondria to peroxisomes upon mitochondrial depolarisation. On the whole, the authors have done a good job in addressing my concerns. I just have some small minor points remaining.

1)The authors appear to have carried out some experiments for the benefit of the reviewers alone and not the future reader! I strongly suggest the authors add the new figures and brief descriptions in

the text (can be as EV/supplemental) as I believe they not only enhance the paper but would also be expected - Figure 1 for reviewer #1 shows that this translocation is not an artefact of exogenous Parkin. This is important. Figure 4 for Reviewer #1 shows that the translocation is not necessary for mitophagy per se. This is also important.

Our reply:

We thank the Reviewer for the suggestion. We have incorporated *Figure 1 for Reviewer #1* from the previous rebuttal letter into the revised manuscript as new Appendix Figure S6. Similarly, *Figure 4 for Reviewer #1* was added to the revised manuscript as new Figure EV5. Because FACS analysis was repeated during the revision process, the new Figure EV5 is slightly modified from the original *Figure 4 for Reviewer #1* with new statistical analysis. Our conclusion, however, remains the same.

2) I appreciate that the authors have now added magnified images when looking at co-localization, and I also appreciate that peroxisomes are very small. However, it is still not easy to see overlap. Perhaps, and providing it does not make the image too crowded, the authors could add a few arrows to highlight representative examples of MITOL-Peroxisome co-localization.

Our reply:

As per the Reviewer's suggestion, arrowheads were added to the magnified images to enhance the readers' finding of MITOL-peroxisome co-localization.

Referee #2:

The authors have performed a considerable amount of work and have addressed most of my criticisms (quantification/statistical analysis, experiments with endogenous/3Flag MITOL, subcellular fractionations).

Very few additional comments:

- In New Figure 2D, the authors claim that "we confirm that total cellular amount of MITOL had not been dramatically decreased". However, this sentence is approximative/hazardous as (1) the levels of 3Flag MITOL do indeed seem to decrease in the PNS after 6h treatment and (2) the authors, yet again, don't perform any quantification. In New Figure 2D, the authors use Mfn as a mitophagy marker, however Mfn is known to be extracted from the depolarized mitochondria following ubiquitination, and as such it is not an ideal mitophagy marker. The authors should also use a matrix marker to assess clearance of damaged mitochondria after a longer (than 6 hours) treatment.

Our reply:

To address this crucial comment, the protocol for the fractionation experiment depicted in Figure 2D was modified. Previously, sample cells were divided into three groups and treated with valinomycin for 0, 3, and 6 h, and then serially centrifuged at 800 *g* and 3,000 *g* to obtain the post-nuclear supernatant (PNS) and mitochondria-enriched pellet (Mt-rich) fractions, respectively. The 3,000 *g* pellet fraction was suspended in the same volume as the PNS fraction to reflect the protein distribution (see *Figure 1A for Reviewer #2* below) and then equal volumes of both were loaded and immunoblotted. However, because long-term treatment with valinomycin results in significant cell death, the number of viable cells decreases over the treatment period. As a consequence, the protein content of the samples differs over time. It is likely this fluctuation in cellular protein that caused the reduction in 3Flag-MITOL that the Reviewer observed in the previous data. In our new experiment, the chemical apoptosis inhibitor ZVAD-FMK was included to prevent valinomycin-induced cell death. In addition, rather than equalizing input volume, the total amount of protein contained in each PNS sample was determined and normalized across the PNS samples, and then equal volumes of the PNS and 3,000 *g* pellet fraction pair were loaded and blotted using the indicated antibodies (see *Figure 1B for Reviewer #2* below). We then used ImageQuant TL (GE Healthcare) to calculate the relative band intensity across three independent experiments.

Figure 1 for reviewer #2

Lastly, given that MFN is extracted from depolarized mitochondria prior to mitophagy and undergoes proteasomal degradation (Tanaka et al., JCB. 2010), we sought to compare MITOL with a more appropriate marker protein for mitophagy-associated degradation. We thus extended valinomycin treatment to 24 h and used MTCO2 (Cytochrome c oxidase subunit 2) as an indicator of mitophagic degradation. Under these new conditions, we found MFN2 was significantly reduced at 3 h whereas a similar reduction in MTCO2 levels was not apparent until 24 h in both the PNS and Mt-enriched fractions. In contrast, the PNS levels of 3Flag-MITOL at 3 and 6 h were comparable with a slight reduction observed at 24 hours (see Figure 2 for reviewer #2 below). Collectively, we confirmed that the total cellular amount of MITOL does not dramatically decrease in response to mitophagy stimulation even when both MFN (degraded by the proteasome) and MTCO2 (degraded by mitophagy) decrease. These new immunoblotting data have been incorporated in the revised manuscript as new Figures 2D and 2E. We appreciate that the Reviewer's comments lead to an improvement in the quality of our data.

Figure 2 for reviewer #2

Minor comment: in response to my comment 6.2, the authors suggest that I refer to Figure 7 for reviewer 1. I assume it is Figure 5 for reviewer 1?

Our reply:

We thank the reviewer for careful reading. "Figure 5 for reviewer 1" is correct.

Referee #3:

The authors have made a very good attempt at answering my questions. My only concern is whether this is a physiological movement of MITOL to peroxisomes that is being described. The authors make a weak statement that the movement of MITOL to peroxisomes might affect peroxisome abundance and size, without any real quantification, especially of size, and that this depends on the E3 activity of MITOL. The phenomenon described is interesting enough and carefully done to warrant publication, but the physiological relevance remains to be rigorously proven.

Our reply:

We appreciate this constructive comment from Reviewer #3. In response to the Reviewer's comment, we quantitatively analyzed the effects of the MITOL Δ C8 mutant, which lacks eight C-terminal amino acids, on peroxisomal size and abundance. HeLa cells expressing HA-Parkin and MITOL wild-type or the Δ C8 mutant were treated with 15 μ M CCCP for 3 h, and then the number of peroxisomes was determined as PMP70-positive dots per 100 μ m². We confirmed that peroxisome abundance in cells expressing MITOL Δ C8 was significantly decreased compared with cells expressing wild-type MITOL (*left panel in Figure 1 for Reviewer #3, below*). In addition, the approximate size of peroxisomes was determined as the number of pixels occupied by one peroxisome. When PMP70-positive pixels per 100 μ m² was divided by the number of peroxisomes in the same area, we found that MITOL Δ C8 caused a drastic expansion in peroxisomal size following CCCP treatment (*right panel in Figure 1 for Reviewer #3, below*). These data have been incorporated into the revised manuscript as new Fig. 12C and D.

Regarding the physiological relevance of MITOL translocation, although we do not have a clear answer at this time, we agree with its importance and have modified the Discussion text that introduces a potential role. "Mitochondria and peroxisomes share some metabolic pathways, and to maintain cellular homeostasis their distribution and abundance are highly coordinated. As such, it is possible that MITOL plays a critical role in balancing the two organelles when mitochondria are damaged. More detailed explorations of this potential homeostatic mechanism warrant future study".

Figure 1 for reviewer #3

In Fig. 13C, why is the MITOL that is peroxisome-associated not shown as being ubiquitylated, even though their data show ubiquitin on peroxisomes. Is this ubiquitin not on MITOL? This ambiguity should at least be acknowledged explicitly by a statement in the legend to Fig. 13C such as "Even though our data in A and B indicate the presence of ubiquitin on peroxisomes containing MITOL, we do not know the target of this ubiquitylation, so MITOL inserted into the peroxisome membrane is shown here in the non-ubiquitylated form".

Our reply:

As indicated by the Reviewer, ubiquitin accumulates on peroxisomes in cells expressing MITOL $\Delta C8$ following CCCP treatment (Fig 12A). Ubiquitylated bands of wild-type MITOL were clearly observed following CCCP treatment (Fig. EV2A), an indication that MITOL is ubiquitylated on mitochondria prior to peroxisome translocation. However, this ubiquitylation was apparent only in the presence of the p97/VCP inhibitor NMS-873 and was faint in the absence of NMS-873 (Fig. EV2A). This suggests that ubiquitylation of MITOL after translocation to peroxisomes is much lower than before translocation. We, however, have yet to identify the ubiquitylation target of MITOL $\Delta C8$ on peroxisomes. Therefore, the following sentence was added to the legend of Fig. 13C "As ubiquitylation on MITOL is rarely observed in the absence of NMS-873, MITOL inserted into the peroxisome membrane is shown here in the non-ubiquitylated form".

I am very pleased to accept your manuscript for publication in the next available issue of EMBO reports. Thank you for your contribution to our journal.

Corresponding Author Name: Noriyuki Matsuda

Journal Submitted to: EMBO reports

Manuscript Number: EMBOR-2019-47728-T